# LungVis 1.0: an automatic AI-powered 3D imaging ecosystem unveils spatial profiling of nanoparticle delivery and acinar migration of lung macrophages

Lin Yang [1] ✉, Qiongliang Liu[1,11], Pramod Kumar[1], Arunima Sengupta[1], Ali Farnoud[1], Ruolin Shen[2], Darya Trofimova[3,4], Sebastian Ziegler[3,4], Neda Davoudi[5,6], Ali Doryab[1], Ali Önder Yildirim[1], Markus E. Diefenbacher[1,7,8], Herbert B. Schiller[1,9], Daniel Razansky [5,6], Marie Piraud [2], Gerald Burgstaller[1], Wolfgang G. Kreyling [1,10], Fabian Isensee [3,4], Markus Rehberg[1], Tobias Stoeger[1] & Otmar Schmid [1] ✉

Targeted (nano-)drug delivery is essential for treating respiratory diseases, which are often confined to distinct lung regions. However, spatio-temporal profiling of drugs or nanoparticles (NPs) and their interactions with lung macrophages remains unresolved. Here, we present LungVis 1.0, an AI-powered imaging ecosystem that integrates light sheet fluorescence micro-scopy with deep learning-based image analysis pipelines to map NP deposition and dosage holistically and quantitatively across bronchial and alveolar (aci-nar) regions in murine lungs for widely-used bulk-liquid and aerosol-based delivery methods. We demonstrate that bulk-liquid delivery results in patchy NP distribution with elevated bronchial doses, whereas aerosols achieve uni-form deposition reaching distal alveoli. Furthermore, we reveal that lung tissue-resident macrophages (TRMs) are dynamic, actively patrolling and redistributing NPs within alveoli, contesting the conventional paradigm of TRMs as static entities. LungVis 1.0 provides an advanced framework for exploring pulmonary delivery dynamics and deepening insights into TRM-mediated lung immunity.

Pulmonary drug delivery is crucial for treating lung infections and diseases, including administering lung vaccines and gene therapy[1,2]. For instance, respiratory adenoviral vector vaccines or neutralizing antibodies are administered via intranasal aspiration or inhalation to combat aerogenically transmitted viruses like SARS-CoV-2[3–5]. Nanotechnology-enabled inhalation therapies, such as bioadhesive hydrogels[6], SARS-CoV-2 RBD conjugated to lung-derived exosomes[7], polymer- and ionizable lipid-coated mRNA[8,9], aim to neutralize SARS-CoV-2 and enhance gene modulation. These innovative therapies are

tested in rodents or non-human primates through pulmonary delivery, often via bulk liquid application or aerosol inhalation. The investiga-tion of dose and distribution patterns within conducting airways and deep alveolar regions across various delivery routes, however, remains elusive. On the other hand, inhaled particulate matter (PM2.5 ≤ 2.5 μm) and engineered nanoparticles (NPs: 1–100 nm) contribute to human respiratory and cardiovascular diseases and mortality[10–15]. Animal models are often used to study pathophysiological changes, environ-mental lung injuries, and inhalation nanotoxicology through topical

delivery of NPs, chemicals, and tobacco. Yet, the innate immunity and systemic lung responses to the differential distribution patterns of these delivered substances are rarely explored. This gap primarily arises from technological limitations, resulting in a limited understanding of their spatio-temporal NP distribution within the lung.

Delivering drugs to often localized or heterogeneous lung regions is challenging due to variations in spatial deposition influenced by delivery method, lung anatomy, breathing patterns, and aerosol size[16]. Current endeavors focus on optimizing delivery vehicles through material engineering, such as screening ionizable lipids[1], employing biomimetic polymeric NPs[17], and leveraging extracellular vesicles like exosomes[7,18]. Experimental investigation of their local dosage and spatial distribution in animal lungs typically involves imaging techniques. Non-optical imaging techniques including X-ray imaging, planar γ-scintigraphy, positron emission tomography (PET) or single photon emission computed tomography (SPECT), offer quantitative data but lack microscopic detail[19–22]. Optical measurements are predominantly conducted in animals using low-resolution whole-organ or whole-body fluorescence imaging or 2D microscopy[1,7–9,17,18,23], leading to an incomplete understanding of cellular localization and molecular interactions within 3D tissue niches. Although serial block-face cryomicrotome imaging generates 3D lung meshes for visualizing particle deposition, it requires time-intensive data acquisition and laborious manual corrections[24]. Computational models predict site-specific deposition[25], while our previous work integrated X-ray phase contrast and tissue-cleared light sheet fluorescence microscopy (LSFM) for real-time, 3D cellular-resolution mapping of NP distribution in entire murine lungs[20,26]. Given that lung diseases such as asthma or emphysema are often regionally localized to the airways or acinar region[27,28], high spatio-temporal resolution imaging is vital for designing precise, disease-specific treatment strategies for precision inhalation therapy.

Current studies suggest that NPs deposited on airway epithelium are rapidly removed by mucociliary transport, while residual NPs in the acinar region can persist for months to years[29–35]. Biopersistent NPs are likely to passively cross the epithelium and relocate to the lung interstitium. Lung tissue-resident macrophages (TRMs), including alveolar macrophages (AMs) and interstitial macrophages (IMs), play a crucial role in maintaining homeostasis by phagocytosing and removing NPs, endogenous proteins, lipids, and dead cells[31,36]. TRMs are traditionally considered static, even under lipopolysaccharide stimulation and bacterial infection[37]. However, recent evidence suggests that AMs actively patrol the alveolar epithelium in both steady-state and bacterial infection scenarios[38,39]. Also, microparticle-laden AMs have been observed migrating to tracheobronchial (TB) lymph nodes[40]. Nevertheless, the specific roles of AMs and IMs in the regulation of NP redistribution are not yet fully understood. Even passive processes enabling NPs to move and diffuse into interstitial spaces are still considered potentially relevant for prolonged retention.

In this work, we advance the visualization and quantification capability of spatial NP profiling in the murine lung through development of LungVis 1.0, a technological ecosystem integrating tissue-cleared LSFM with artificial intelligence (AI) and deep learning-driven imaging analysis (convolutional neural networks, CNNs)[41] in a precise and automatic manner. The state-of-the-art AI approach, nnU-Net, has shown great promise in semantic segmentation in the biomedical domain[42]. LungVis 1.0 integrates and optimizes data-centric active learning nnU-Net pipelines to overcome multiple LSFM imaging artifacts, achieving accurate label-free segmentation of the complete airway tree. Harnessing this for qualitative and quantitative analysis of bronchial, inter-acinar, and intra-acinar NP distribution patterns in four commonly used preclinical lung delivery routes revealed substantial differences at both macroscopical and microscopical levels. LungVis 1.0 coupled to multimodal analytics uncover the unique role of TRMs

in determining NP cellular fate, potentially paving the way for new strategies in nanomedicine and precision inhalation therapy, particularly for modulating TRM phagocytosis and mobility. The raw and AI-enabled LSFM datasets of 78 murine lungs with AI-driven complete airway segmentations and associated reference annotation as well as the AI model (modified nnU-Net source code) obtained from the LungVis 1.0 ecosystem are accessible via the open-access repositories Zenodo (https://doi.org/10.5281/zenodo.7413818)[43] and GitHub[44], supporting further lung health and disease research.

## Results

### LungVis 1.0: data-centric active learning AI-driven precision segmentation of entire lung airways

Despite the significant interest in pulmonary drug delivery, as evidenced by the ca. 1500 annual publications (since 2019) on this issue in Web of Science Core Collection (Supplementary Fig. 1a), accurate spatial profiling of pulmonary delivered substances (here NPs) throughout the entire lung continues to be a major challenge. To faithfully recapitulate NP distribution and transport in vivo, we developed LungVis 1.0, an AI-driven 3D imaging ecosystem for murine lungs. This integrated methodology includes several key components: optimized tissue-cleared light sheet fluorescence microscopy (LSFM) for data acquisition, AI and deep learning pipelines for whole-airway segmentation, and visualization and quantification of NP spatial deposition (Fig. 1a). Using convolutional neural networks (CNNs), LungVis 1.0 enables automated, precise, and rapid segmentation of the entire lung bronchial tree in non-stained LSFM images, facilitating comprehensive visualization and quantification of NP deposition in airways and acini (Fig. 1a). Specifically, LungVis 1.0 employs a data-centric active learning pipeline combining limited expert annotations with gradually added training samples through manual corrections of model predictions. Initial ground truth (GT) annotations of entire bronchial trees were generated by four lung specialists and underwent quality control via final inspection by the most experienced lung expert. Example reference annotations of the GT in 2D and 3D are shown in Fig. 1b and Supplementary Movie 1. We applied a modified version of nnU-Net[42] to the manually annotated training images (GT) from three lungs and then used human-driven active learning to select and manually correct AI-generated annotations for iteratively retraining the model. This allowed us to quickly expand our initial training set of 3 cases to 21 lung images (Fig. 1c).

Occasionally, some LSFM datasets are challenging to segment due to their wide range of imaging artifacts, such as shadows, blurring, inconsistent illumination, and obscure lung regions, among others. Furthermore, LSFM was carried out using both visible (high autofluorescence, AF1, i.e., excitation/emission (ex/em) = 540 / 590 nm, Supplementary Movie 2) and near-infrared light (low AF2, ex/em = 740/790 nm, Supplementary Movie 3), which further complicates the segmentation task. We thus incorporated dataset-specific data augmentation techniques (e.g., Gaussian blur, local blurring transform) simulating common imaging defects into the nnU-Net framework (Fig. 1c). Our iterative data generation and targeted augmentation approach enabled us to generate a highly robust segmentation method, capable of producing high-quality segmentations of the entire airway structure in previously unannotated images (>60 full lungs), even in the presence of apparent imaging artifacts, as demonstrated by exemplary samples (Fig. 1d, Supplementary Fig. 1b−e, Supplementary Movie 4). The modified nnU-Net also demonstrated remarkable resilience to worst-case scenarios with low AF intensity, out-of-focus, artificial structures, and other imaging errors (Supplementary Fig. 2).

Manually annotating a single 3D LSFM lung image stack, which includes 200-500 2D slices, demands more than 100 h of work (Fig. 1e). Therefore, manually segmenting 78 lung image stacks would be impractical. LungVis 1.0 enables segmentation of a single lung stack

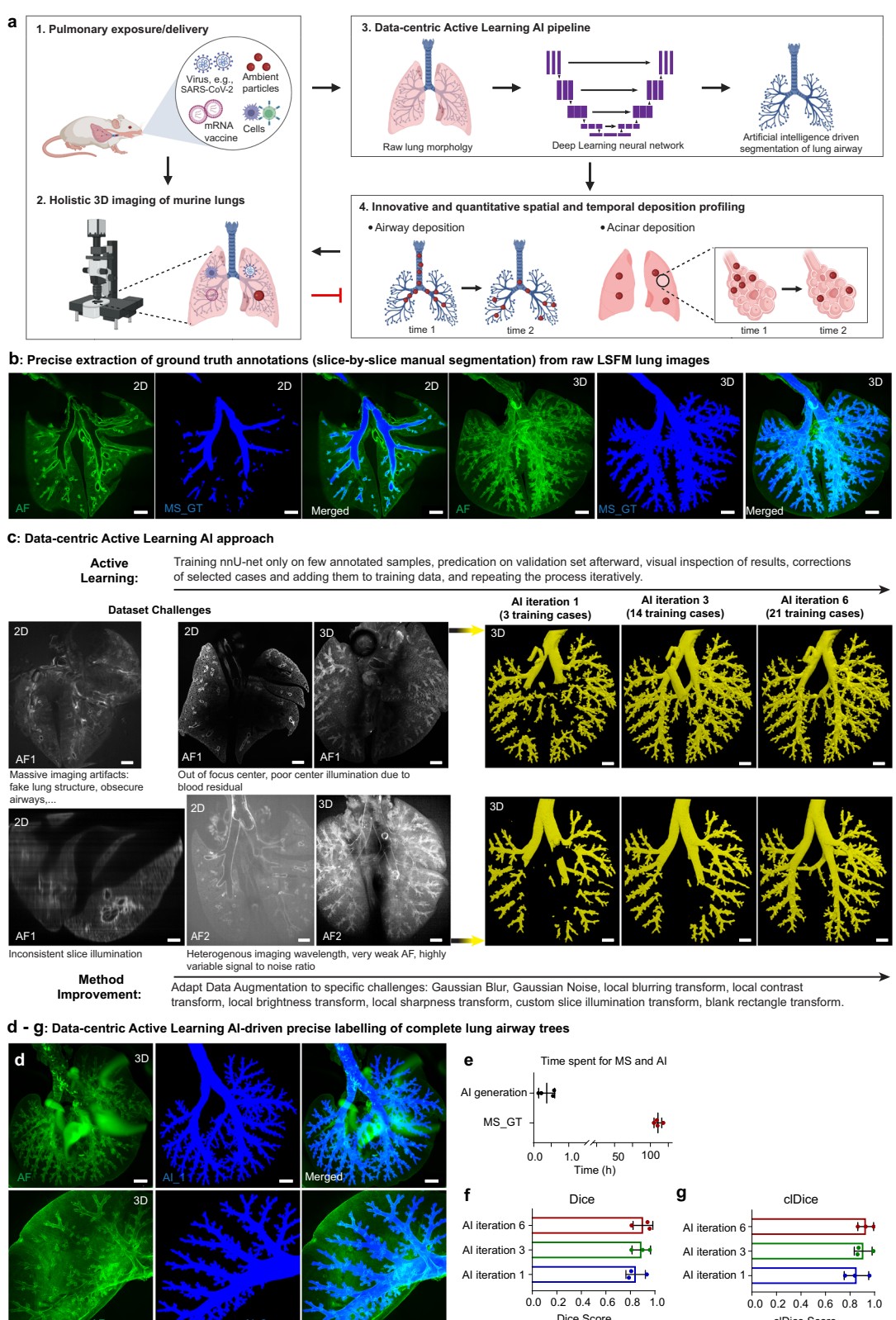

**a**

**1. Pulmonary exposure/delivery**

Virus, e.g., SARS-CoV-2 | Ambient particles | mRNA vaccine | Cells

**2. Holistic 3D imaging of murine lungs**

**3. Data-centric Active Learning AI pipeline**

Raw lung morpholgy | Deep Learning neural network | Artificial intelligence driven segmentation of lung airway

**4. Innovative and quantitative spatial and temporal deposition profiling**

- Airway deposition — time 1, time 2
- Acinar deposition — time 1, time 2

**b**: Precise extraction of ground truth annotations (slice-by-slice manual segmentation) from raw LSFM lung images

2D AF | 2D MS_GT | 2D Merged | 3D AF | 3D MS_GT | 3D Merged

**c**: Data-centric Active Learning AI approach

**Active Learning:** Training nnU-net only on few annotated samples, predication on validation set afterward, visual inspection of results, corrections of selected cases and adding them to training data, and repeating the process iteratively.

**Dataset Challenges**

AI iteration 1 (3 training cases) | AI iteration 3 (14 training cases) | AI iteration 6 (21 training cases)

2D AF1 — Massive imaging artifacts: fake lung structure, obscecure airways,...

2D AF1 / 3D AF1 — Out of focus center, poor center illumination due to blood residual

2D AF1 — Inconsistent slice illumination

2D AF2 / 3D AF2 — Heterogenous imaging wavelength, very weak AF, highly variable signal to noise ratio

**Method Improvement:** Adapt Data Augmentation to specific challenges: Gaussian Blur, Gaussian Noise, local blurring transform, local contrast transform, local brightness transform, local sharpness transform, custom slice illumination transform, blank rectangle transform.

**d - g**: Data-centric Active Learning AI-driven precise labelling of complete lung airway trees

**d**

3D AF | 3D AI_1 | 3D Merged
3D AF | 3D AI_2 | 3D Merged

**e** Time spent for MS and AI

AI generation | MS_GT | Time (h)

**f** Dice

AI iteration 6 | AI iteration 3 | AI iteration 1 | Dice Score

**g** clDice

AI iteration 6 | AI iteration 3 | AI iteration 1 | clDice Score

within an average of 23 mins (Fig. 1e). The AI-generated airway segmentations were quantitatively evaluated using 5-fold Cross Validation during development and a held-out test set. For Cross Validation, the training data was randomly split into five folds and five models were trained on four folds each, with evaluation on the remaining one. The test set consisted of three full lung images not used during development, with Dice Score and centerline Dice Score (clDice) computed for every image and the mean reported. The validation set achieved a Dice Score of 0.92 and a clDice of 0.94, while the test set gradually achieved better performance with a final Dice Score of 0.90 and a clDice of 0.93 in iteration 6 (Fig. 1f, g). These scores suggest that LungVis 1.0 enables automated artifact correction and achieves nearly 100% completeness of the bronchial tree, closely aligning with the ground truth annotation.

**Fig. 1 | Data-centric, active learning, artificial intelligence-driven precise segmentation of complete murine lung airways. a** Schematic illustration of the key features of LungVis 1.0 including 3D LSFM lung imaging and the development of artificial intelligence-driven airway segmentation to resolve the spatial and temporal nanoparticle (NP) deposition profiles in the mouse lung. Illustrations created partially with Biorender.com. **b** Manual extraction of ground truth (MS_GT, lung annotations) from the raw, non-stained LSFM images. 2D and 3D images show the original LSFM lung structure (autofluorescence, AF in green), manually segmented GT (in blue), and merged images. **c** LungVis 1.0 AI pipelines overcome multiple imaging challenges for high-performance airway segmentation. Occasionally, poor image quality arises from imaging shadows, blurring effects, out-of-focus central region (poor illumination in lung center), inconsistent slice illumination, poor and variable signal-to-noise ratio, false gray structures, etc. can be observed in some of LSFM lung images. With the data-centric active learning approach and method improvement, LungVis 1.0 achieved high quality and robust segmentations even in the most challenging cases, as demonstrated for label-free AF lung images in the visible (high AF1 - default AF channel) and near-infrared channel (low AF2). **d** Two exemplary AI segmentations of complete bronchial trees from either a whole lung or a single lung lobe with different imaging errors (i.e., imaging shadow and blurring) are displayed. Representative data from *n* = 78 biological samples. **e** Average time investment for complete airway labeling in lungs via manual *versus* AI segmentation. Data are presented as mean ± SD, *n* = 4 biological replicates. **f–g** The Dice Score and centerline Dice Score were evaluated across three GT lungs in three AI iterations from the test datasets. Data are presented as mean ± SD, *n* = 3 biological replicates. Scale bars:1000 μm. Source data are provided as a Source Data file.

## LungVis 1.0 revealed distinct NP delivery profiles in airways and acini for four pulmonary delivery routes

The lung can be anatomically subdivided in the conducting airways and acinar/alveolar region. The former is dedicated to convective transport of inhaled air from the trachea to the latter region, where the gas-exchange takes place. A consistent terminology is provided to facilitate straightforward comprehension of NP deposition profiles throughout the entire lung enabled with LungVis 1.0 (Supplementary Fig. 3). The conducting airways (bronchial tree) can be segregated into two parts, the central or upper airways (starting from the trachea to large and even smallest airways (bronchioles) in the central/upper region of the lung) and the peripheral or lower airways (the peripheral/lower region of the lung including mainly bronchi and bronchioles). The functional unit of the lung -the acinus- is defined as the distal part of terminal bronchioles starting from the alveolar duct to the most distal alveolar sacs (alveoli)[45]. The acini are located throughout the lung in central, intermediate and peripheral locations. Within an acinus, the proximal acinar region (PAR) and distal acinar region (DAR) are designated to pinpoint the intra-acinar NP distribution features. Albeit essential for precision inhalation therapy, a comprehensive understanding of the drug/NP delivery and distribution patterns throughout the entire lung down to the acini is currently inadequate mainly due to technological limitations.

We utilized LungVis 1.0 to elucidate holistic and cellular-resolution NP distribution profiles in airways and acinar regions immediately following four prevalent pulmonary delivery methods: intranasal (bulk) liquid aspiration (INLA), intratracheal (bulk) liquid instillation (ITLI), ventilator-assisted aerosol delivery (VAAD), and nose-only aerosol inhalation (NOAI) (Fig. 2a). Melamine resin fluorescent (MF) NPs exhibit uniform particle size (monodisperse) and spherical morphology (Fig. 2b). As reported in our previous studies[20,26], MF NPs demonstrate robust fluorescence stability under various chemical and mechanical treatments. Furthermore, in vitro incubation of MF NPs with MH-S cells for up to four days shows that the fluorescence signal remains resilient to phagolysosomal degradation by macrophages, exhibiting neither bleaching nor leaching (Supplementary Fig. 4). These results underscore optical stability of MF NPs for longitudinal biokinetics studies down to cellular resolution in tissue-cleared optically transparent lungs (Fig. 2c).

While raw lung images illustrate general regional differences in NP deposition between liquid-based (INLA) and aerosol-based (NOAI) applications across the entire lung (Supplementary Fig. 5a, b), quantitative characterization of these differences resolved for airways and acini proves difficult. The advanced airway-acinus separation capabilities of LungVis 1.0, however, overcome this challenge by offering comprehensive 3D visualization of NP distribution patterns in lung airways and acini post bulk-liquid and aerosol applications (Fig. 2d–o). The schematic illustrations delineate NP distribution patterns within the airways, as well as across central and peripheral acini (inter-acinar regions), and intra-acinar regions (PAR, DAR), pinpointing distinct deposition differences attributed to liquid- *versus* aerosol-based NP delivery methods as derived from LungVis 1.0 (Fig. 2d, g, j, m). Immediately after bulk liquid application, LungVis 1.0-derived visualization of the airways only highlights a predominant deposition of NPs (as a form of large aggregates) in the central and upper bronchi and bronchioles, as indicated by orange arrowheads in INLA and ITLI images (Fig. 2e, h, Supplementary Movie 1), although a few peripheral bronchioles in the upper lung also receive high NP deposition (blue arrowheads). Conversely, aerosol-based applications (VAAD and NOAI, Fig. 2k, n) revealed a more even NP distribution across both upper/lower and central/peripheral bronchioles, marked by orange and white arrowheads, respectively. In concordance with NP airway deposition, the central and upper acini received the majority of NPs during bulk-liquid delivery. In contrast, the aerosol-based applications exhibited a highly homogeneous NP distribution throughout the entire lung, spanning from the central to peripheral acinar regions (Fig. 2l, o, Supplementary Movie 5). Moreover, aerosol-based methods facilitated NP distribution along the alveolar ducts to deep sacs, achieving a balanced NP presence between proximal and distal acinar regions (PAR and DAR), as depicted in Fig. 2l, o. This is distinctly different from bulk liquid methods, notably INLA, predominantly targeting the PAR, as evidenced in 3D and 2D views (Fig. 2f, i, Supplementary Fig. 5c, d). Although VAAD and NOAI shared similar intra-acinar deposition characteristics, VAAD resulted in a higher NP dosage within PAR as compared to DAR, as highlighted in Supplementary Fig. 5e, f.

The quantitative dosage of NPs in the entire lung was determined through spectrofluorometric analysis of tissue homogenates (Fig. 2a). In comparison, liquid-based administrations, such as INLA (28.1% ± 16.6%) and ITLI (59.2% ± 14.5%), demonstrate substantially higher delivery efficiencies (percentages of pulmonary-deposited dose to invested dose) than aerosol-based routes like VAAD (4.24% ± 1.58%) and NOAI (0.19% ± 0.02%) (Fig. 2p). The higher invested doses for aerosol deliveries, with VAAD at 166.7 μg and NOAI at 475 μg, as compared to liquid applications (ITLI: 25 μg and INLA: 34.7 μg), partially offset these differences, leading to comparable lung delivered doses. Specifically, ITLI and INLA delivered 14.8 ± 3.7 μg and 9.70 ± 5.75 μg, respectively, compared to 4.24 ± 1.58 μg for VAAD and 0.91 ± 0.13 μg for NOAI. The central-to-peripheral deposition (C/P) ratio, which is generally used to indicate holistic homogeneity of aerosol deposition in the lung, yields C/P ratios close unity for perfect peripheral penetration of aerosol deposition in human lungs[22]. Based on our LSFM images, this study confirms peripheral aerosol deposition for VAAD and NOAI as indicated by C/P ratios of 1.37 ± 0.13 and 1.32 ± 0.14, respectively, while bulk liquid application yields preferential deposition in the central part of the lung for ITLI and - even more pronounced - for INLA with C/P ratios of 2.45 ± 0.41 and 3.43 ± 0.35, respectively (Fig. 2q).

In addition to traditional analysis of NP delivery profiles, LungVis 1.0 introduces previously unreported metrics for detailed quantitative

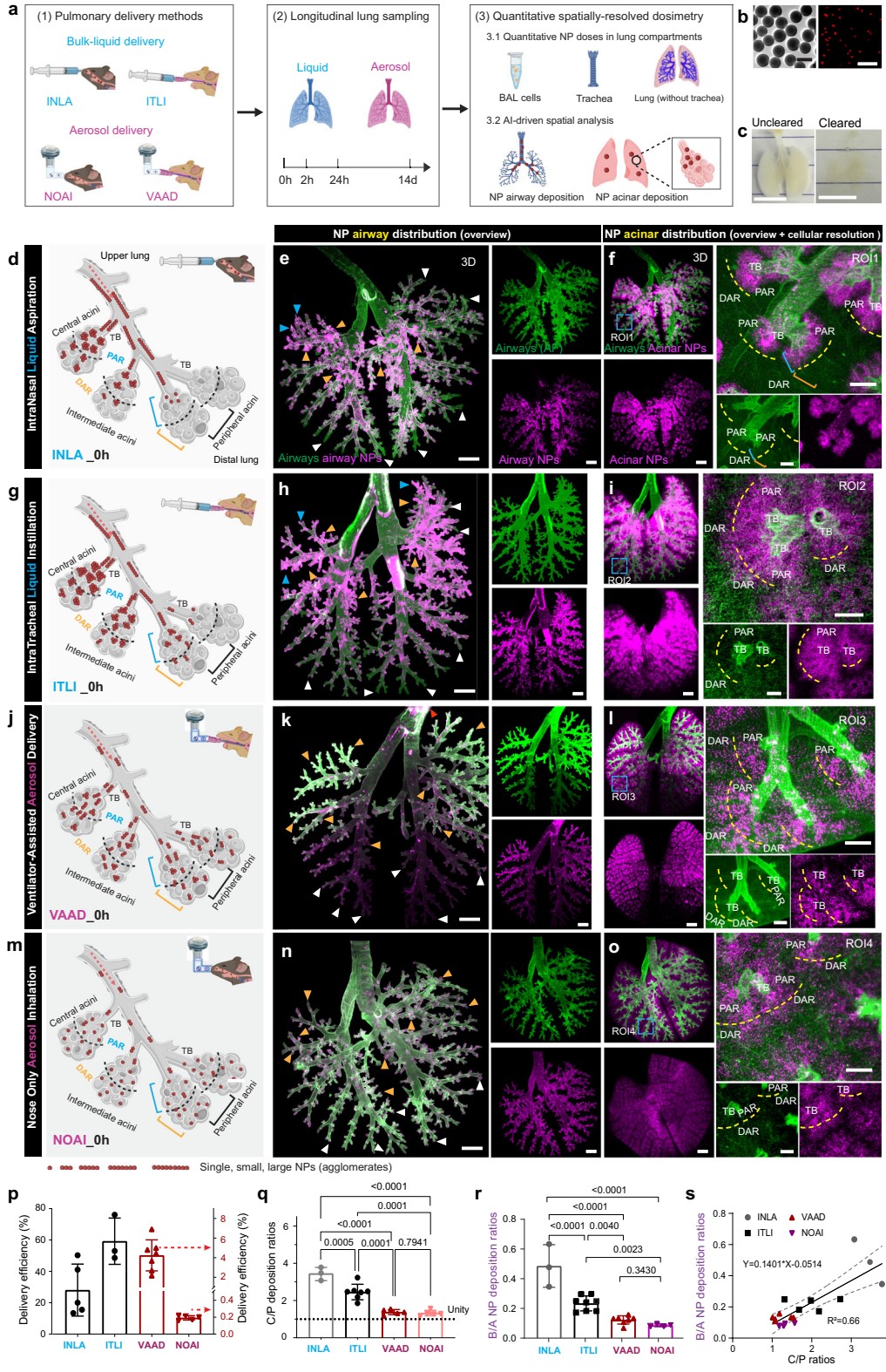

Single, small, large NPs (agglomerates)

NP spatial profiling (Fig. 2r, Supplementary Fig. 6, and Supplementary Data 1). For all four types of application, more than two thirds of the delivered NPs (68–92%) were deposited in the alveolar region (Supplementary Fig. 6h) but the corresponding bronchial delivery fractions for bulk liquid application (INLA = 0.32 ± 0.06, ITLI = 0.19 ± 0.03) were substantially higher than those for aerosol inhalation (VAAD = 0.11 ± 0.02, NOAI = 0.08 ± 0.01) (Supplementary Fig. 6i). Consequently, bronchial-to-acinar (B/A) dose ratios of bulk-liquid applications

resulted in 2- to 4-fold higher bronchial deposition (B/A: INLA = 0.49 ± 0.14, ITLI = 0.23 ± 0.05) as compared to aerosol-based delivery (B/A: VAAD = 0.12 ± 0.02, NOAI = 0.08 ± 0.01) (Fig. 2r). Linear regression analysis revealed a significant correlation between the C/P ratio and B/A ratio, with a coefficient of determination ($R^2$) of 0.66 (Fig. 2s), indicating that the former could be a good predictor of the latter not only for mice, but for humans. This underscores the usefulness of the clinically most widely used C/P ratio as a coarse but predictive

**Fig. 2 | Qualitative and quantitative profiling of initial NP distribution in lung airways and acini enabled by LungVis 1.0. a** Schematic illustration delineates pulmonary NP delivery methodologies, longitudinal sampling, and quantitative NP dose in tissue homogenates and spatial NP dosimetry (created with Biorender.com). **b** Representative TEM and epi-fluorescence images of MF NPs. $n = 3$, independent replicates with similar results. Scale bars: 500 nm (left) and 15 μm (right). **c** Optical transparency of a mouse lung prior to and after tissue clearing. Scale bars: 1 cm. **d–o** AI-powered holistic mapping and cellular resolution views of NP lung distribution for four delivery routes stratified for the airway (**e, h, k, n**) and acinar region (**f, i, l, o**). **d, g, j, m** Schematic depiction of NP airway and acinar deposition (created with Biorender.com). Orange and white arrowheads indicate central/upper and lower/peripheral bronchioles, respectively. Blue arrowheads show NP deposition in upper peripheral regions (**e, h**). Red arrowheads show tracheal-deposited NPs (**k**). An acinus is delineated into the proximal and distal acinar region (PAR and DAR) artificially separated by black (**d, g, j, m**) or yellow (**f, i, l, o**) dashed lines. TB: terminal bronchioles. Scale bars: 1000 μm (overview), 100 μm (cellular resolution). **p** Quantitative analysis of NP-lung delivery efficiency. $n = 5$ INLA, $n = 3$ ITLI, $n = 7$ VAAD, $n = 5$ NOAI, biological replicates. **q** Central/peripheral NP deposition ratio. $n = 3$ INLA, $n = 7$ ITLI, $n = 5$ VAAD, $n = 5$ NOAI, biological replicates. **r** Bronchial/acinar NP deposition ratio. Representative data (**d–o**) from $n = 3$ INLA, $n = 8$ ITLI, $n = 7$ VAAD, $n = 4$ NOAI, biological replicates. **s** Linear correlation between C/P ratio and B/A deposition ratio with mean and 95% confidence intervals indicated by solid and dashed lines, respectively. $n = 3$ INLA, $n = 5$ ITLI, $n = 6$ VAAD, $n = 4$ NOAI, biological replicates. Data are presented as mean ± SD, calculated using one-way ANOVA with Holm-Šídák's multiple comparison test (**q, r**). Source data are provided as a Source Data file.

parameter for the spatial uniformity of particle deposition within the lung[22,26,46,47].

NPs often form large agglomerates when delivered as bulk liquid or aerosol droplets, impacting their cellular fate via e.g., size-dependent endocytic uptake pathways and hence toxicological/therapeutic responses[48,49]. To quantify NP agglomeration states in bronchial and acinar regions the apparent regional NP-positive volume was normalized to the respective NP mass dose. A higher NP-volume/dose value (μm³ μg⁻¹) indicates wider spreading of NPs in the lung. As expected, aerosol delivery via NOAI displayed significantly higher apparent NP-volume/dose values (>10-fold higher) in both bronchial and acinar regions compared to bulk liquid-based deliveries (Supplementary Fig. 6j, k). This is qualitatively confirmed by the presence of larger NP clusters following bulk-liquid application as compared to the wider spreading of NPs (small-dot-like pattern) after aerosol delivery (Fig. 2e, h, k, n). Interestingly, VAAD aerosol delivery does not show a different agglomeration state than bulk liquid application possibly due to the more than 4-fold higher NP dose delivered by VAAD as compared to NOAI. Moreover, NOAI was expected to yield the lowest agglomeration state in the acinar region, since dried 0.7 μm aerosol experiences at least partially polydirectional diffusive deposition, while 3 μm liquid aerosol (VAAD) experiences unidirectional settling, leading to more spatially focused VAAD NP deposition. On the other hand, the lack of improved spreading for VAAD as compared to bulk liquid applications could at least partially be an artifact of the exaggerated NP volume (as compared to the geometric volume) due to light scattering effects, which currently obfuscates not only smaller differences in agglomeration state, it also renders LungView1.0 incapable of obtaining correct values for the material density of NPs in the lung. The almost constant B/A ratio of apparent NP-volume/dose (0.26–0.38) indicates 3- to 4-fold higher NP packing density in the NP covered regions of the airways than in that of the acini, independent of delivery route (Supplementary Fig. 6l). This is consistent with a spatially focused, inertial impaction-governed deposition profile as expected for the bronchial region. Collectively, LungVis 1.0 provides (semi-)quantitative insights into pulmonary deposition profiles, including bronchial and acinar NP doses and dose fractions, and NP agglomeration states in airway and acinar regions.

## LungVis 1.0 elucidated intra-acinar NP relocation independent of delivery route

The in-depth longitudinal analysis of NP spatial and temporal profiling was performed for ITLI and VAAD delivery as representatives of the bulk-liquid and aerosol-based applications, respectively, since (1) their B/A ratio is relatively similar (23% *versus* 12%), and (2) the general distribution profiles were similar for the two bulk liquid and two aerosol application routes (Fig. 2d–o, q, r). For ease of visualization, the results of the longitudinal study are presented for the left half (left lobe) of the lung, which reliably represents both lung morphology and NP dose profile of the entire lung[25,26]. It is important to note, that for each time

point the animal has to be sacrificed, i.e., mouse-to-mouse variability in initial NP profile will contribute to longitudinal variability. Analogous to the whole lung (Fig. 2), airway visualization enabled by LungVis 1.0 clearly indicates that for ITLI NPs were preferentially delivered to the central/upper airways particularly to the central terminal bronchioles (Fig. 3a). NPs were also seen on the surface of the airway at later time points, with their burden progressively reducing over time up to 14 d (Fig. 3e, i, m). Analysis of the inter-acinar NP distribution post-ITLI reveals a significant accumulation of NPs within the central acini, far surpassing that in the peripheral acini, as depicted in Fig. 3b, c and Supplementary Movie 6. Observations at later stages (2 h, 24 h, 14 days) indicate that NPs continue to concentrate in the central acini without apparent spreading throughout the lung (Fig. 3e–p, Supplementary Fig. 7a–d and Supplementary Movies 7–9), suggesting the absence of transport from central/upper to peripheral/lower acini. The substantial NP signal difference in adjacent acini at both 24 h and 14 days post-ITLI delivery (ca. 40-50-fold dose ratio, Supplementary Fig. 8a–b) further suggests an absence of inter-acinar NP exchange, as such exchange would likely alleviate these large NP dose differences between closely situated acini. Upon analyzing the intra-acinar NP distribution following ITLI, a significant clustering of NPs at the entrances of acini (PAR) is observed with both macroscopic and microscopic resolution, marked by blue arrowheads and white dotted circles in Fig. 3c, d. The initially high proximal-to-distal dose gradient within the acini gradually diminished over time. By day 14, there is no dose gradient anymore and the NPs are appearing as uniformly spaced, scattered single dots (small NP agglomerates) all the way out into the very distal acinar region, with no large NP clusters in the PAR, as depicted in Fig. 3h, l, p. As few NPs were initially delivered to the very distal part of acini (Fig. 3c, d) this pattern change indicates intra-acinar NP relocation. Longitudinal 2D ITLI images (Supplementary Fig. 9a–d) also revealed a transformation in NP distribution from dense clustering to a more dispersed, dot-like arrangement over time. These inter-acinar and intra-acinar NP biokinetic profiles post-ITLI were schematically illustrated in Fig. 3q.

At 0 h the VAAD lung exhibited a relatively uniform NP deposition profile across the central and peripheral bronchial tree with the highest dose in the smallest and terminal bronchioles (Fig. 4a). Similar to the NP airway burden in ITLI lungs, the NP signal in VAAD lungs exhibited a sustained presence on the airway surface with a gradual reduction in NP signal (dose) over time (Fig. 4e, i, m, Supplementary Fig. 7e–h). Surprisingly, we observed lower NP deposition in the airway and acini located in the upper part of the lung in some cases (purple arrowheads, Fig. 4e–j). This could be a result of non-homogeneous ventilation of the upper part of the lung when animals are lying on their back. In agreement with ITLI inter-acinar observations, those acini with no/low initial NP deposition are not gaining more NPs over time in VAAD lungs (Fig. 4a–p and Supplementary Movies 10–13). Also, histogram profiling of pixel intensity shows a substantial NP signal difference between high and no/low NP acinar regions, with the latter

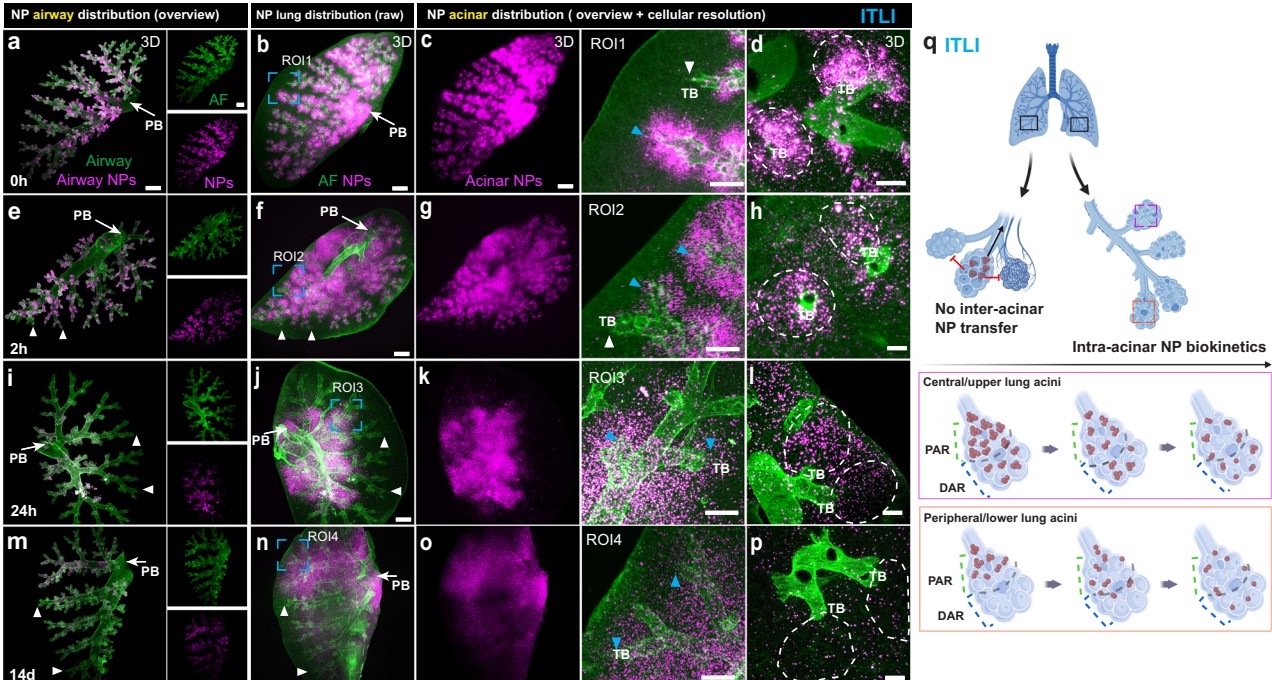

**Fig. 3 | Time-resolved 3D qualitative and quantitative profiling of NP bioki-netics in intact ITLI lungs enabled by LungVis 1.0. a–p** Longitudinal 3D view of NP distribution pattern in the left lung after intratracheal liquid instillation (ITLI) at time points of 0 h, 2 h, 24 h, and 14 d. **a, e, i, m** AI-powered visualization of NP airway deposition pattern. **b, f, j, n** Visualization of NP lung distribution (raw LSFM ima-ges). AI-powered global (**c, g, k, o**) and cellular resolution views (**d, h, l, p**, within white dashed lines) of NP acinar deposition pattern. White arrowheads indicate no/ low NP deposition in ITLI lungs and blue arrowheads indicate different NP (re-) distribution features in the proximal acinar region (PAR). PB: Primary bronchus, TB: Terminal bronchioles. Scale bars: 1000 µm (**a–c, e–g, i–k, m–o**), 400 µm (ROIs), 200 µm (**d, h, l, p**). Representative data from $n = 8$ ITLI_0 h, $n = 3$ ITLI_2 h, $n = 8$ ITLI_24 h, $n = 5$ ITLI_14 days, independent biological replicates. NP intensity (brightness) optimized for spatial visualization, was scaled to match longitudinal lung homogenate dose trends but may not reflect the actual dose. **q** Schematic depiction of NP inter-acinar and intra-acinar transfer in the ITLI lungs (created partially with Biorender.com).

receiving only about 4–5% of the particles at both the 24 h and 14 days time points (Supplementary Fig. 8c, d). This indicates no active NP exchange among acini (inter-acinarly). Considering intra-acinar NP relocation, it is evident that the initially higher and more localized dose in the PAR, which is due to gravimetric settling of NP-laden micron-sized liquid aerosol along the region of higher air flow - center of the alveolar duct - leads to NP agglomeration (white dotted circles, Fig. 4c, d, Supplementary Fig. 7e), turned gradually into a less loca-lized, spatially uniform distribution of smaller NP agglomerates (single dot-like fluorescence pattern) within 14 days after NP delivery (Fig. 4g, h, k, l, o, p, Supplementary Fig. 7f–h, Fig. 4q). This reduction in NP agglomerate size and localization state is also observed in the corresponding 2D images (Supplementary Fig. 9e–h). This is corro-borated by quantitative LungVis 1.0 analysis, revealing - for ITLI - a significant decrease in residual bronchial NP dose fraction from 0.19 at 0 day to 0.07 at 14 days after having remained relatively constant from 2 h to 24 d at 0.10-0.11, whereas VAAD remained constant at 0.10-0.11 until 24 h before dropping to 0.06 at 14 d (Fig. 4r). This indicates that there is a considerably larger fast clearance fraction of NPs (within 2 h) for bulk liquid application (ITLI), than for aerosol application (VAAD). Conversely, the acinar dose fraction for ITLI increased from $0.81 \pm 0.03$ at 0 h to $0.92 \pm 0.01$ by 14 d, with VAAD showing a notable rise from $0.89 \pm 0.02$ to $0.93 \pm 0.01$ between 24 h and 14 days post-administration (Fig. 4s). Overall, LungVis 1.0 has allowed us to reveal the previously underappreciated longitudinal and spatially resolved regional NP distribution profiles, elucidating both lack of inter- and presence of intra-acinar NP dynamics. NP relocation occurs indepen-dently of the initial uniformity of NP deposition.

Quantitative analysis of NP retention in tissue homogenates or bronchoalveolar lavage (BAL) fluid reveals different retention trends for NPs administered via VAAD and ITLI. ITLI-treated lungs showed a constant retention fraction within the first 24 h (92% at 2 h and 97% at 24 h relative to applied dose at 0 h), which then declined to 27% by 14 days (Fig. 4t). In comparison, for VAAD-treated lungs, NP retention levels at 2 h, 24 h, and 14 days were found to be approximately 66%, 80%, and 23% of the initial dose, respectively (Fig. 4u). From LSFM images of lavaged lungs we showed MF NPs were not fully washed out by BAL fluid collection, neither for ITLI nor VAAD administration (Supplementary Fig. 10a–h). From a global perspective, initial NP dis-tribution patterns in ITLI and VAAD lavaged lungs appeared similar to non-lavaged counterparts. On a more detailed level, NP clusters in ITLI lungs have been pushed through alveolar epithelium into the inter-stitial tissue as evidenced by numerous NP clusters near the blood vessels (white arrowheads, Supplementary Fig. 10a, b). VAAD at 0 h demonstrates less affected by BAL collection, showing NP clusters slightly compacted in alveolar epithelium (yellow arrowheads, Sup-plementary Fig. 10e, f). In both ITLI_24h and VAAD_24h lavaged lungs, no apparent BAL-induced features were observed, leaving even-sized NP clusters within acinar regions (Supplementary Fig. 10c, d, g, h). These qualitative observations align with the quantification of lavageable NP fractions in BAL fluid (relative to retained dose), which showed a decrease from 0.51 to 0.13–0.17 after VAAD application and from 0.64 to 0.30–0.27 at 0 h and 14 days post ITLI exposure, respectively (Supplementary Fig. 10i, j). The comparable lavageable fractions at 24 h for both applications (0.16–0.30) are consistent with literature values on NP retention of approximately 0.2 for various sub-100 nm NPs, such as 20 nm gold, titanium oxide, and iridium particles[21,29,33,34,50]. The higher initial lavageable NP fraction for ITLI compared to VAAD is consistent with more central/upper airway and acinar NP deposition in ITLI lungs (Supplementary Fig. 10k, l), while

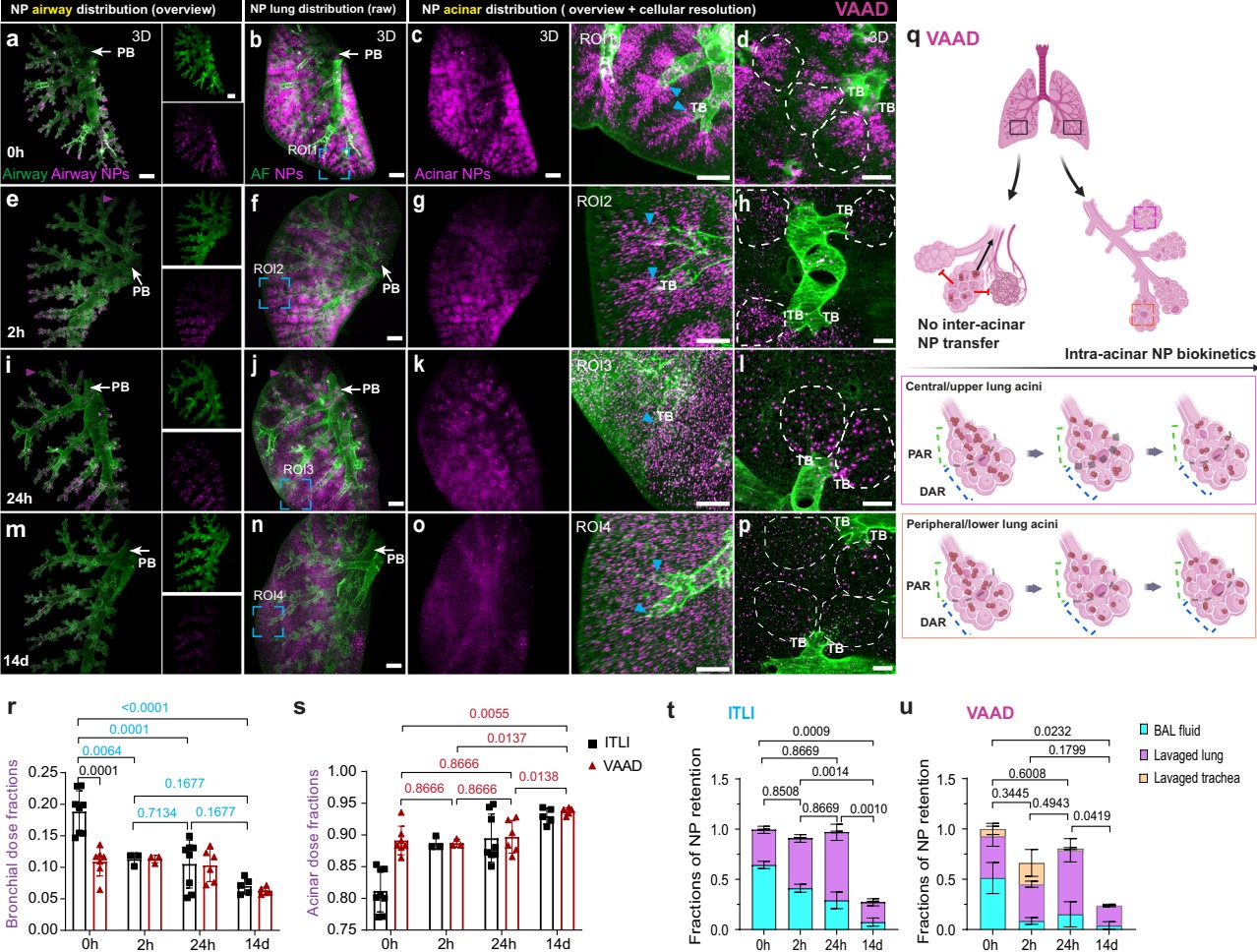

**Fig. 4 | Time-resolved 3D qualitative and quantitative profiling of NP bioki-netics in intact VAAD lungs enabled by LungVis 1.0. a–p** Longitudinal 3D view of NP distribution pattern in the left lung after ventilator-assisted aerosol delivery (VAAD) at time points of 0 h, 2 h, 24 h, and 14 d. **a, e, i, m** AI-powered visualization of NP airway deposition pattern. **b, f, j, n** Visualization of NP lung distribution (raw LSFM images). AI-powered global (**c, g, k, o**) and cellular resolution views (**d, h, l, p**, within white dashed lines) of NP acinar deposition pattern. Purple arrowheads refer to low NP deposition in parts of the upper lung after VAAD. Blue arrowheads indicate different NP (re-)distribution features in PAR. PB: Primary bronchus, TB: Terminal bronchioles. Scale bars: 1000 μm (**a–c, e–g, i–k, m–o**), 400 μm (ROIs), 200 μm (**d, h, l, p**). Representative data from n = 7 VAAD_0 h, n = 3 VAAD_2 h, n = 6 VAAD_24 h, n = 5 VAAD_14 days, independent biological replicates. NP intensity (brightness) optimized for spatial visualization, was scaled to match

temporal lung homogenate dose trends but may not reflect the actual dose. **q** Schematic depiction of NP inter-acinar and intra-acinar transfer in the VAAD lungs (created partially with Biorender.com). **r, s** Longitudinal bronchial and acinar NP deposition fractions in the lung. n = 8 ITLI_0 h, n = 3 ITLI_2 h, n = 8 ITLI_24 h, n = 5 ITLI_14 days, n = 7 VAAD_0 h, n = 3 VAAD_2 h, n = 6 VAAD_24 h, n = 5 VAAD_14 days, independent biological replicates. **t, u** Fraction of NP retention in different compartments of the lung. The retention fractions are normalized to the mean dose at 0 h. n = 3, independent biological replicates. Data are presented as mean ± SD, calculated using one-way ANOVA with Holm-Šídák's multiple comparison test (**r–u**). p values are indicated separately for ITLI (**r**) and VAAD (**s**) groups. Additional p values for bronchial dose fractions between ITLI_0 h and VAAD_0 h determined using unpaired two-tailed t-test (**r**). Source data are provided as a Source Data file.

substantial initial tracheal deposition of VAAD lungs (40% of total delivered dose[20]) likely explains the high 2 h NP trachea dose (30%, Fig. 4u, Supplementary Fig. 10l).

Longitudinal imaging showed that the initial hotspot NP deposition in the trachea disappeared within 2 h for ITLI and 24 h for VAAD (Supplementary Fig. 11). However, fine NP streaks persisted up to 24 h and individual NPs persisted up to 14 days in the trachea for both ITLI and VAAD indicating a sustained, albeit diminishing, mucociliary clearance of NPs from the lung for at least 14 days. This is consistent with the well-known biphasic lung clearance rate attributed to the initial fast clearance of NPs from the mucus-covered tracheal and bronchial region (within a few hours) and the later slower long-term clearance pathway for particles from the acinar region of the lung (between a few days to years)[29–31,35]. Of note, the large amount of NPs deposited in the distal part of the trachea at 0 h (here 2 h in

Supplementary Fig. 11) after VAAD application, is due to direct impaction of the aerosol jet exiting the relative narrow intubation cannula, which is placed in the trachea for VAAD application as reported in our previous studies[20,26]. The observation that in contrast to VAAD application very few particles were detected in the trachea at 14 days after ITLI (signal below detection limit) is consistent with preferential acinar deposition (prolonged clearance) for VAAD. Yet we cannot rule out that this may be at least partially due to residual NPs from the initial strong VAAD hotspot deposition in the trachea.

**Airway and acinar-deposited NPs were primarily and efficiently phagocytosed by TRMs**

To explore immune responses linked to NP relocation, we conducted dedicated experiments with a suite of complementary analytical techniques for NP localization through tissue sectioning and flow

cytometry, dynamic tracking of NPs via intravital and living tissue microscopy, and for discrimination of active and passive NP transport employing an ex vivo NP-lung interaction model (Fig. 5a). It is worth noting that the applied dose of NPs did not induce any recruitment of inflammatory cells to the lung (see below). 3D localization of NPs in immunostained precision cut lung slices (PCLS) indicated that at 2 h after ITLI, NPs were located freely on the alveolar septum (yellow arrowheads) or within F4/80+ TRMs (white arrowheads) around the PAR. In contrast, 2 h post-VAAD revealed single NPs and small clusters within distal alveolar sacs and septum (Fig. 5b, c, Supplementary Fig. 12). By 24 h and 14 days, NPs from both ITLI and VAAD had primarily accumulated in F4/80+ macrophages across various PCLS locations (white arrowheads, Fig. 5b, c, Supplementary Fig. 12), indicating phagocytosis and TRM migration as key mechanism for intra-acinar NP relocation. Detailed analysis of NP-laden TRMs at 2 h revealed very different localization of macrophages within single imaging frames (Supplementary Fig. 12a, d). Quantitative fluorescence analysis revealed that a fraction of 0.22–0.25 of the total retained NP doses in PCLS was phagocytosed by F4/80+ TRMs at 2 h post-ITLI and VAAD and the remaining NPs were found on the epithelium. The fraction of TRM-contained NPs increased within 24 h significantly to 0.83–0.93 and remained at this level at 14 d in both ITLI and VAAD lungs (Fig. 5d). At 24 h, the ratio of NP-laden to total F4/80+ TRMs was approximately 0.71–0.73 (Fig. 5e), suggesting that TRMs were not yet saturated with NPs for either route of NP application. In Fig. 5f, we identified four typical alveolar locations of TRMs: epithelium-surface macrophages (I), epithelium-attached macrophages (II), tisse-crossing macrophages (III), and inner-tissue macrophages (IV). At 24 h post-ITLI and VAAD, NP-laden tissue-crossing and inner-tissue macrophages constituted just under half of all NP-laden macrophages, with proportions of 0.45 ± 0.08 and 0.45 ± 0.03, respectively (Fig. 5f). Those TRMs could be likely considered as the traditionally recognized interstitial macrophages (IMs)[36,51,52], indicating their potentially active role in phagocytosis post NP lung exposures.

Indeed, 3D reconstruction of F4/80+ TRMs within lung tissue architecture allowed for precise visualization of tissue-crossing and inner-tissue macrophage locations, as shown for 2 h post-ITLI and VAAD lungs (Fig. 5g, h, Supplementary Movies 14, 15). Specifically, some NP-laden tissue-crossing macrophages were located in the interalveolar pores, known as the pores of Kohn[53–55], suggesting their ability to migrate across alveoli within an acinus (not inter-acinar) for alveolar clearance of NPs or cell debris, which is crucial for maintaining tissue homeostasis.

Flow cytometry of whole lungs was used to explore the role of AMs (NP+SiglecF+) and IMs (NP+CD11b+) in NP relocation in 24 h ITLI mouse lungs[52,56], comparing lavaged and non-lavaged conditions. The gating strategy for NP-laden macrophages is depicted in Fig. 5i. We observed a significant reduction in the percentage of NP-laden cells, from 3.0% in non-lavaged lungs to 1.8% in lavaged lungs (Fig. 5j). Although the proportion of SiglecF+ AMs among NP-laden cells decreased post-lavage, not all SiglecF+ AMs were removed by BAL fluid collection (Fig. 5k). Consistently, the fraction of NPs within CD11b+ IMs rose to 28% of the total NP dose in lavaged lungs (Fig. 5l). Consistent with PCLS finding, flow cytometry demonstrated that about 86–89% of NP+ cells are either AMs or IMs, ingesting about 92-96% NPs in the normal and lavaged lungs (Fig. 5k, l).

BAL cytology revealed no signs of pulmonary inflammation as evidenced by the absence of polymorphonuclear neutrophils (PMNs)[57] in the airspace after ITLI and VAAD exposures (Supplementary Fig. 13a, b). BAL cytology was performed with two quantitative methods: bright field microscopy of May-Grünwald-Giemsa stained cells (Supplementary Fig. 13a–c) and fluorescence-activated cell sorting (FACS) analysis (Supplementary Fig. 13d). A high linear correlation ($R^2 = 0.81$) was observed between the two quantitative methods (Supplementary Fig. 13e). However, microscopic examination identified 30–40% more

NP+ macrophages compared to FACS, indicating that microscopy has higher detection sensitivity. Both methods indicate no difference in NP+ macrophage fraction for VAAD and ITLI application (except for an elevated NP+ macrophage fraction for 24 h VAAD from microscopy) albeit the former had received a *ca.* 3-fold lower acinar dose. Both methods showed a similar increase of the NP+ macrophage fraction between 2 h and 24 h and between 24 h and 14 days, with VAAD reaching its maximum value of ca. 0.7 already at 24 (for microscopy method only) (Supplementary Fig. 13f). Overall, quantitative analysis and visualization of NPs in BAL cells and lung tissues using PCLS and flow cytometry demonstrates that NPs were efficiently taken up not only by AMs, but also by IMs.

## Phagocytosis and dynamic patrolling of TRMs caused intra-acinar NP relocation

Real-time imaging of phagocytosis and patrolling of TRMs can further elucidate the static (longitudinal) information on macrophage uptake, migration, and associated NP relocation presented above. Lung IVM was recently used to unveil cellular circuits during e.g., viral infection, bacterial clearance, disease progression, tissue inflammation, and immune responses in the lungs of living mice[38,58,59]. This study adapted lung IVM and ex vivo living tissue microscopy[60] to observe in situ cellular dynamics of TRMs during NP ingestion after both forms of administration (Fig. 6 and Supplementary Fig. 14a). Fluorescence labeling (PKH26)[38] of macrophages in living mice allowed in vivo monitoring of the migration of TRMs over 2 h (Fig. 6a). A small fraction of stationary PHK-TRMs (16–28%) sprawled or pirouetted around their initial position without significant net displacement (<0.2 μm per minute). The larger fraction of more motile PHK-TRMs migrated multi-dimensionally with a speed of 0.2–1.0 μm per minute either along the epithelial surface or - much less frequently - across the epithelium through pores of Kohn connecting two adjacent alveoli. This was observed for both 0 h VAAD and 24 h ITLI lungs (Fig. 6a, b, Supplementary Movie 16). Migrating NP-laden PHK-TRMs are likely to contribute to the LSFM-observed pulmonary redistribution including interalveolar NP transport through pores of Kohn (Fig. 5g, h). No clear patrolling directions during 2 h IVM were observed, implying TRMs patrolling of the alveolar walls followed random walk patterns (Fig. 6c). Flow cytometric analysis revealed that the percentage of PKH+ cells in the 24 h ITLI lung was reduced from 1.8–0.8% due to lavaging (Supplementary Fig. 14b, c). Among them, 32–34% cells were NP+ cells, independent of BAL fluid collection (Fig. 6d), referring to a large ratio of PKH-labeled cells that were NP-free. However, approximately 83–87% of all PKH+ cells and PKH+NP+ cells were SiglecF+ cells (Supplementary Figs. 14d, 6e), confirming the previous finding that PKH dye primarily labels to AMs[38]. A small fraction of PKH+ cells and PKH+NP+ cells (17%) showed CD11b+ signal (Supplementary Fig. 14d, 6e), which could be attributed to the simultaneous expression of CD11b and SiglecF in a few AMs and/or PKH labeled IMs. Notably, about 75% NP+ cells were PKH+ cells (Supplementary Fig. 14e), indicating NP+ cells contained at least 25% of other cells rather than AMs.

Ex vivo living lung sections from MacGreen ITLI transgenic mice (CSF1R-EGFP) were further used to track migratory routes of GFP+ macrophages for 18 h using LSFM (Zeiss) and laser scanning confocal microscopy (LSM). Notably, we revealed a dynamic location exchange within the macrophage population: a NP+GFP+ macrophage (white arrowhead) was seen slowly navigating towards another NP-GFP+ macrophage (yellow arrowhead). Upon the approach of the NP+GFP+ macrophage, the NP-GFP+ macrophage initiated movement to a different location, a process that unfolded gradually over an 18 h period (Fig. 6f and Supplementary Movie 17). Additionally, two other NP+GFP+ macrophages were observed engaging in active movement, each crawling in opposite directions (Supplementary Fig. 14a). GFP+ cells constitute about 20% of all lung single cells and only 3.8% and 3.0% of GFP+ cells are NP+ cells in normal and lavaged lungs, respectively

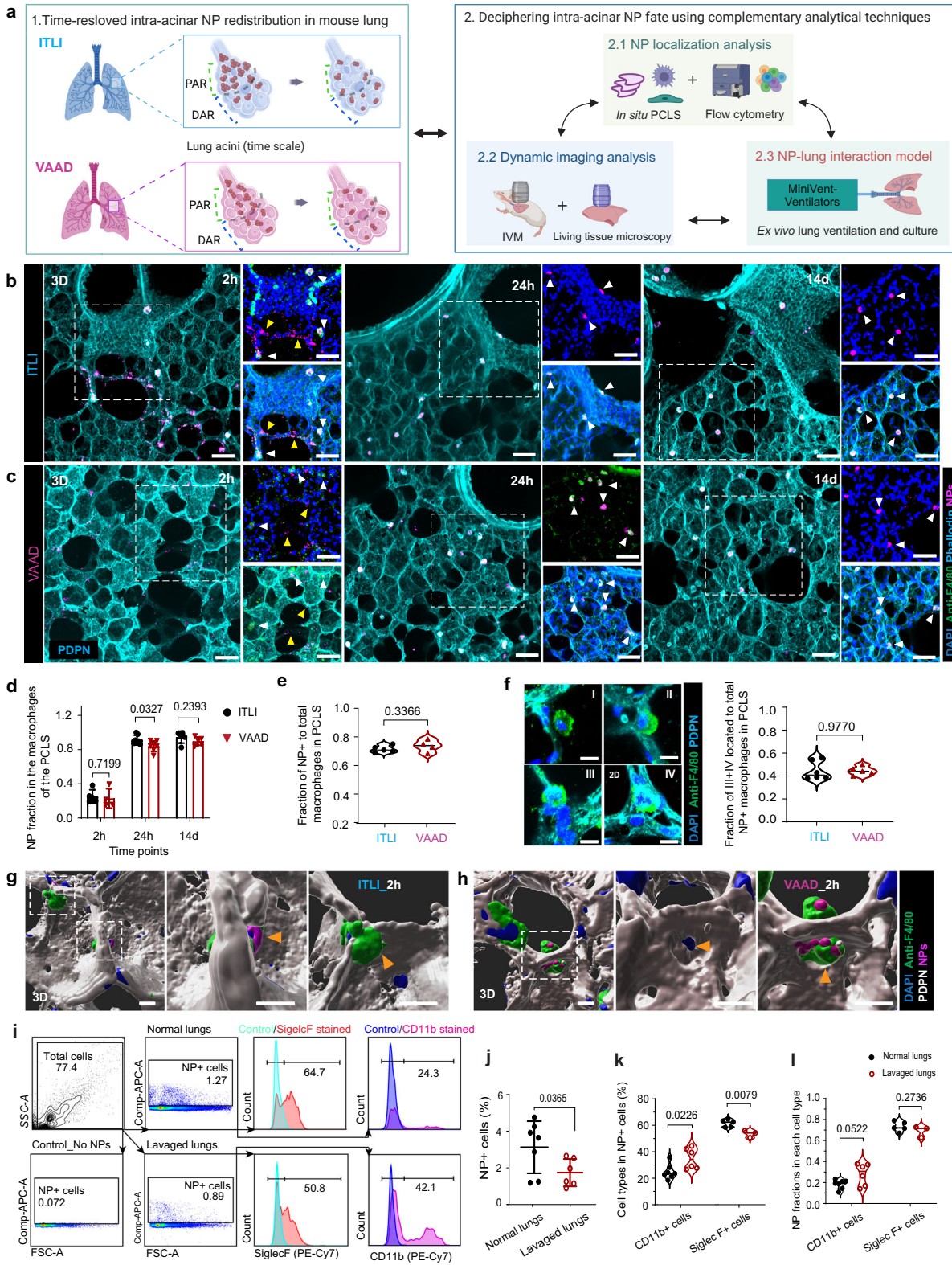

(Supplementary Figs. 14f, and Fig. 6g, h). However, total GFP⁺ cells had very high percentages of CD11b⁺ cells (82–90%) with a small ratio of SiglecF⁺ cells (17–11%) in both types of lungs (Supplementary Fig. 14g). Also, the percentage of CD11b⁺ cells in total GFP⁺NP⁺ cells increased 45% in lavaged lungs, while the percentage of SiglecF⁺ cells decreased accordingly (Fig. 6i). Whole-mount imaging of an entire lung lobe indicated that numerous GFP⁺NP⁺ phagocytes were observed in a 4 h

ITLI lung (Fig. 6j and Supplementary Movie 18), indicating quick and efficient uptake of NPs by TRMs. Analysis of BAL cells in 24 h ITLI lungs harvested from PKH labeled MacGreen mice showed that GFP⁺ cells (blue arrows) might refer to other macrophage subtypes besides AMs, since they were not labeled by PKH (Fig. 6k). Some GFP⁺PKH⁺NP⁺ cells had weak PHK fluorescence intensity but considering their strong NP intensity, which was found to leak to the PKH channel, suggesting

**Fig. 5 | Lung airway and acinar deposited particles are effectively phagocytosed by tissue-resident macrophages. a** Complementary analytical technique for deciphering intra-acinar NP fate (created with Biorender.com). **b, c** Representative cellular-resolution 3D views of NP distribution in PCLS at different time points. White and yellow arrowheads indicate NP aggregates engulfed by F4/80⁺ tissue-resident macrophages (TRMs, anti-F4/80) and free NPs in alveolar epithelium, respectively. Podoplanin (PDPN): alveolar epithelial cell Type 1, Phalloidin: cell actin filaments, DAPI: cell nuclei, Scale bars: 50 μm. **d** Fraction of pulmonary retained NP dose in lung PCLS F4/80⁺ macrophages. **b–d** n = 5 all groups, except n = 7 ITLI_24 h and VAAD_24 h, n = 4 VAAD_2 h, independent biological replicates. **e** The fraction of NP⁺F4/80⁺ to total F4/80⁺ macrophages in 24 h ITLI (n = 5) and VAAD (n = 4) lungs. **f** Four typical localizations of F4/80⁺ macrophages (inset: type I, II, III and IV) and the fraction of tissue-crossing (III) and inner-tissue macrophages (IV) to total NP⁺ macrophages in both 24 h ITLI (n = 6) and VAAD lungs (n = 5). Scale bars: 10 μm. **g, h** Typical localization of tissue-crossing or inner-tissue NP⁺F4/80⁺ macrophages in 3D-reconstructed PCLS from 2 h ITLI (n = 5) and VAAD (n = 4) lungs. Scale bars: 10 μm. **i** Exemplary cell population analysis of NP⁺ macrophages stained with SiglecF or CD11b and quantifications on SiglecF⁺ (AM) and CD11b⁺ (IM) cells to all NP⁺ cell (**j, k**), and the NP fraction attributed to each type (**l**) in 24 h ITLI normal and lavaged lungs. **j** n = 7 normal, n = 6 lavaged lungs. **k, l** n = 7 CD11b⁺_normal, n = 5 SiglecF⁺_normal, n = 6 CD11b⁺_lavaged, n = 3 SiglecF⁺_lavaged, independent biological replicates. Data are presented as mean ± SD, calculated using two-tailed (**e, f**) or one-tailed (**j**) unpaired t test and multiple two-tailed unpaired t test with Holm-Šídák correction (**d, k–l**). Source data are provided as a Source Data file.

those cells were likely the IMs (white arrows) instead of AMs (white arrowheads). Taken together, these findings suggested that MF NPs were primarily phagocytosed by AMs, but also partially by IMs, and that BAL collection could not completely remove all AMs from the lung, supporting the notion that both AMs and IMs contribute to the relocation of NPs.

## Cellular activity rather than passive transport governed NP relocation

The inspiration-expiration cycle of the lung leads to cyclic changes in pressure and volume across the bronchioles and acini. These mechanical motions of the lung particularly, the acini as functional breathing units, can have a direct impact on initial aerosol delivery and subsequent redistribution. Whether the breathing-induced mechanical strain of the lung affects the long-term biokinetics and cellular fate of NPs is uncertain. To discriminate between active cellular transport (e.g., via macrophages) and passive breathing-induced motion as the cause of NP relocation over time, an ex vivo lung model was established and modified from the previously developed organ-restricted vasculature delivery (ORVD) lung model for the transportation of therapeutic silica NPs to lung tumors[61]. Employing this tailor-made NP-lung interaction bioreactor (Fig. 7a), we replicated in vivo breathing movements in ex vivo murine lungs by applying physiological mechanical ventilation after the treatment of 4% paraformaldehyde (PFA) or PBS. Indeed, utilizing PFA to inhibit cellular activity, NPs in fixed ITLI lungs after 24 h of ventilation showed a patchy distribution of NP clusters in central airways and acini (Fig. 7b and Supplementary Fig. 15a), mirroring the pattern of 0 h ITLI lungs (Fig. 3a). In contrast, ex vivo ventilated "living" (PBS) lungs displayed a less patchy distribution profile with more clearly defined individual NP dots, closely resembling but not quite reaching the in vivo 24 h ITLI lung distribution (Fig. 7c, d). This underscores the pivotal influence of cellular activities on NP redistribution over passive breathing-induced mechanical effects. Additionally, the dissimilar NP pattern in an ex vivo "living" lung without ventilation compared to in vivo suggests that breathing also impacts macrophage-mediated NP relocation (Supplementary Fig. 15b, c). Using a comparable ex vivo model, NP delivery through vascular perfusion in a mouse zombie incubated with and without PFA revealed that endothelial transcytosis plays a key role in cancer nanomedicine delivery, surpassing the contributions of enhanced permeability and retention (EPR) effects (a form of passive diffusion)[62]. The present study further revealed that cellular activity, particularly NP uptake and TRM mobility, rather than passive respiration-induced NP motion, is governing NP redistribution within the lung.

## Multispectral 3D imaging revealed holistic views of NP distribution and lung networks

Tissue-cleared LSFM of immunostained whole organs and organisms provides qualitative or quantitative holistic information regarding multiple tissue/organ/body networks such as a transgenic fluorescent cell type[63], nervous system[64], brain vasculature system[65], and whole-body cancer metastasis[66]. This study adopted the recent immunolabeling protocols[67] to gain a holistic view of NPs and phagocytes as well as the entire vasculature system including large blood vessels, capillaries and lymphatics (Fig. 7e–g, and Supplementary Fig. 16). 3D and 2D visualization of the large blood vessels (LYVE1⁺) and lymphatics (PDPN⁺, LYVE1⁺) in a 24 h ITLI lung lobe (Fig. 7e, f, and Supplementary Movie 19) revealed that NPs were primarily located in the alveolar septum/epithelium, less in the interstitium, and occasionally in proximity to blood vessels. Albeit NPs might be touching the surface of the endothelium, NPs were not found in the endothelial layer (Fig. 7f). Larger NP clusters were observed in a 14 days ITLI lung as compared to those of a 4 h ITLI lung, implying that initially deposited single NPs were efficiently removed from the epithelium via uptake by and gradual accumulation in TRMs (Supplementary Fig. 16b). Simultaneous labeling of lectin and α-smooth muscle actin (A-SMA) illuminated the entire network of blood vessels, revealing the widespread distribution of NPs in the lung but not inside the endothelial vessels in a 24 h ITLI lung (Supplementary Fig. 16c). As expected, NP accumulation was observed in the tracheobronchial (TB) lymph nodes isolated from 24 h and 14 days ITLI mice (Fig. 7g and Supplementary Fig. 16d). This could be partially attributed to phagocytosis and translocation of TRMs to the TB lymph nodes[30,40], as evidenced with NP⁺CD45⁺, NP⁺CD68⁺, NP⁺CD11b⁺, and NP⁺CD11c⁺ macrophages in the TB lymph nodes of 24 h and 14 days VAAD mice (Fig. 7h). Altogether, multispectral LSFM of intact lungs disclosed the NP distribution with respect to multiple lung networks particularly the vasculature and lymphatic vessels suggested that time-dependent NP redistribution was governed by TRMs activity.

## Discussion

Despite regionally targeted pulmonary drug delivery being crucial for treating lung infections and diseases such as SARS-CoV-2 and lung cancer[3,4,17,18], techniques to map the bronchial and acinar dose or even more spatially resolved distributions of delivered substances (here NPs) in the entire lung are still lacking for both humans and animal models of disease. Spatially resolved deposition of NPs in the lung is one of the determinants of immune response and biological effect in vivo. Our previous research has integrated in vivo X-ray phase contrast imaging with ex vivo tissue-cleared LSFM to enable real-time monitoring of NP delivery and 3D cellular-resolution visualization of NP distribution in the entire murine lung[20,26]. The present study further developed an AI-driven 3D imaging ecosystem LungVis 1.0, enabling automatic and precise segmentation of the whole bronchial tree and qualitative and quantitative determination of NP bronchial and acinar deposition profiles in murine lungs. Manually annotating a single 3D LSFM lung image stack, with 200-500 2D slices, demands over 100 h. Moreover, imaging artifacts like inconsistent illumination, variable signal-to-noise ratios, out-of-focus, and artificial structures can be observed in lung LSFM images, significantly hindering the manual annotation process. Consequently, manual segmentation of 78 LSFM lung image stacks in this study would be unfeasible. To address these

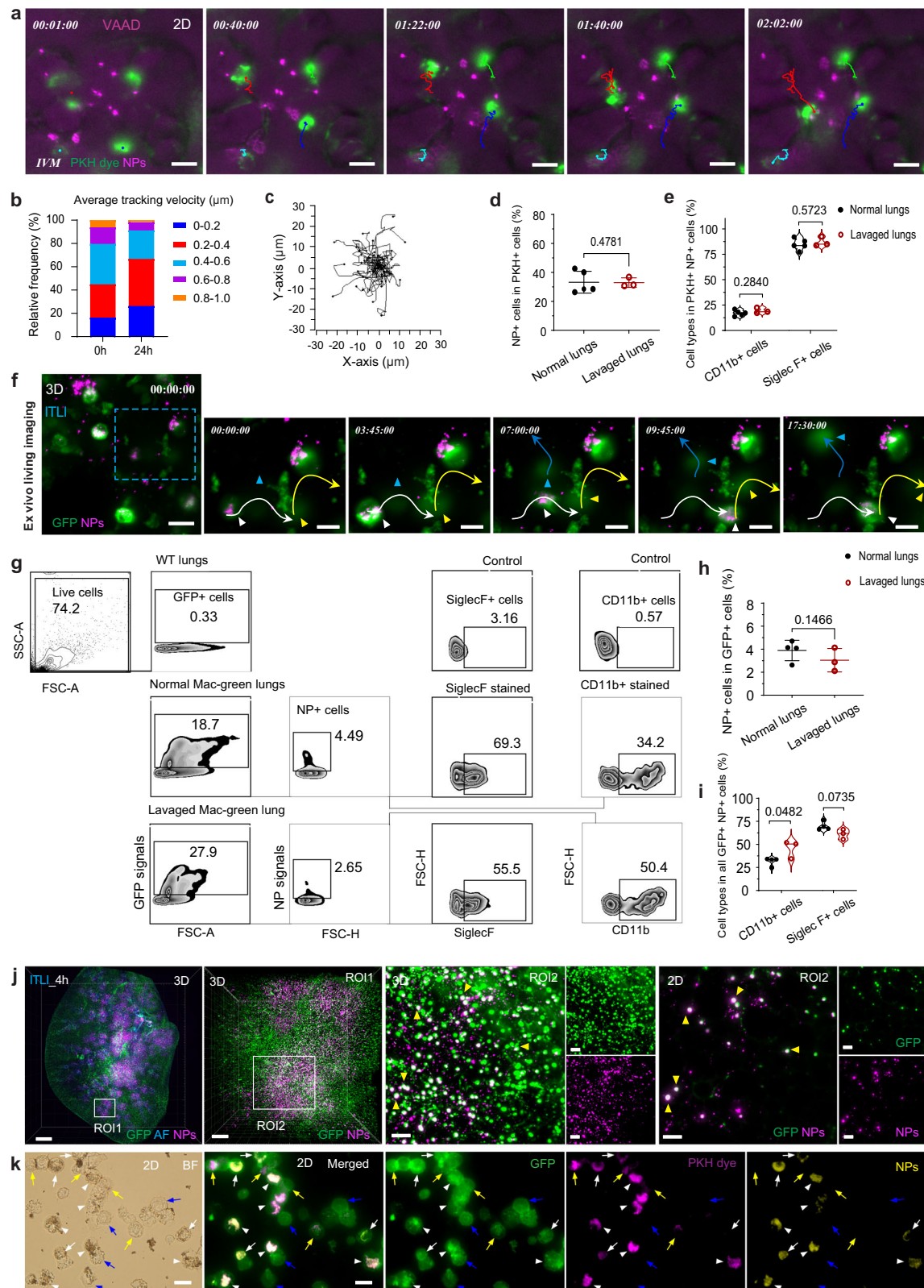

challenges, LungVis 1.0 has been developed, which combines initial expert ground truth annotations with iterative manual corrections of model predictions and utilizes dataset-specific data augmentation techniques to create reliable segmentations. LungVis 1.0, powered by a modified nnU-Net[42], demonstrates remarkable proficiency in segmenting airways in staining-free LSFM images across 78 full lungs. It achieves near-complete accuracy in bronchial tree segmentation compared to ground truth annotations, even in the most challenging scenarios.

LungVis 1.0 delivers highly detailed segmentation of airway trees with precise morphometric accuracy and completeness. Comparative morphometric analysis with traditional lung casts revealed consistent results in length, diameter and branching angle for up to the 20 generations reported for lung casts (C57BL/6 mice) and LungVis 1.0

**Fig. 6 | Redistribution of acinar deposited NPs can be attributed to phagocytosis and migration of TRMs. a** Lung intravital microscopy (IVM) revealed the dynamic movement of PKH labeled macrophages toward epithelial-deposited NPs by patrolling the alveolar epithelial surface. $n = 3$, VAAD_0 h lungs, independent biological replicates. Scale bar: 20 μm. **b** Relative frequencies of average tracking velocities of PKH-labeled macrophages in vivo imaged by IVM. $n = 3$ VAAD_0 h, $n = 4$ ITLI_24 h, independent biological replicates. **c** Trajectory plot outlines relative patrolling behaviors of individual PKH-labeled macrophages from 3 VAAD_0 h lung imaged by IVM (Start points of the migration tracks set to 0,0). **d, e** Fractions of NP$^+$ cells in PKH$^+$ cells and fractions of CD11b$^+$ or SiglecF$^+$ in NP$^+$PKH$^+$ cells in normal and lavaged 24 h ITLI lungs. $n = 3$ (lavaged) or 5 (normal), independent biological replicates. **f** Ex vivo lung living microscopy showed the migration of GFP$^+$NP$^+$ macrophages in the PCLS ($n = 4$). Scale bar: 20 μm and 10 μm (ROI). **g** Exemplary cell

population analysis of GFP$^+$NP$^+$ macrophages and (**h, i**) Fractions of NP$^+$ cells in GFP$^+$ cells and fractions of CD11b$^+$ or SiglecF$^+$ in NP$^+$GFP$^+$ cells in normal and lavaged lungs. $n = 3$ (lavaged) or 4 (normal), independent biological replicates. Data are presented as mean ± SD, calculated using one-tailed unpaired $t$ test (**d, h**) and multiple two-tailed unpaired $t$ test with Holm-Šídák correction (**e, i**). **j** Macroscale and microscale views of GFP$^+$ macrophages/monocytes and NP deposition in an 4 h ITLI lung lobe ($n = 3$). White arrowheads indicate merged signals of GFP and NPs. Scale bars: 1000 μm (overview), 200 μm (ROI1), and 100 μm (ROI2). **k** Co-expression/staining of GFP, PKH dye, and NP$^+$ BAL cells obtained from 24 h ITLI mac-green mice ($n = 3$). White arrowheads and arrows, yellow, and blue arrows indicate triple positive cells, double positive cells (PKH$^+$ and GFP$^+$), and GFP$^+$ cells, respectively. Scale bar: 20 μm. Source data are provided as a Source Data file.

revealed 5 more previously unnoticed generations with diameters down to 90 μm[25]. Thus, lung casts are often not able to resolve the most intricate distal part of the airways. The completeness of the 3D LungVis 1.0 airway trees was confirmed by visual inspection of the point of transition from the bronchial into the alveolar region marking the terminal bronchioles. At the last generations of the LungVis 1.0 airway trees there is an abrupt loss of tissue autofluorescence accompanied by the expected sudden appearance of a line-like hot-spot NP deposition pattern in VAAD lungs (0 h post-application) originating from effective sedimentation of micron-sized drops along the centerline of the air-conducting proximal acinar region. Both features indicate the transition from the airway (bronchial) to the acinar region, thus confirming the completeness of LungVis 1.0 airway tree.

LungVis 1.0 also reveals underappreciated NP delivery and deposition features in airways and acini for four common preclinical routes of respiratory delivery, two liquid-based (ITLA and ITLI) and two aerosol-based (VAAD and NOAI) approaches. Qualitatively, both liquid-based deliveries resulted in larger central/upper airway and acinar deposition of NPs in the lung, while aerosol-based routes provided a much more uniform deposition between central/upper and peripheral/lower airways and acini (inter-acinar). High PAR deposition of NPs was observed in both liquid-based deliveries, particularly for INLA. Even for aerosolized delivery, NP deposition does not reach the most distal parts of the acini mostly likely due to lack of convective air transport into this region as the high diffusivity of oxygen molecules can bridge the remaining distance to the blood vessels. This study thus indicates that 3 μm VAAD aerosol deposits very efficiently in the most distal bronchioles and the PAR, which was speculated to be most relevant region for fast SARS-CoV-2 infections in the deep lung with substantial degradation of the integrity of the pulmonary-vascular barrier[68,69]. Leveraging the power of LungVis 1.0, we revealed that all application routes delivered most of the NPs in the acinar regions (68–93%), yet liquid-based deliveries induced much higher bronchial to acinar (B/A) ratios (>2.3) than aerosol-based administrations (1.1 to 1.4). Traditionally, the B/A ratio of patients was determined by measuring the total lung dose at 0 h and 6 h (or 24 h), where the "missing" pulmonary dose fraction was attributed to fast mucociliary dose clearance from the airways. LungVis 1.0 now enables direct measurement of bronchial and acinar deposition fraction and efficiency. In clinics, 2D whole lung imaging is widely used to determine C/P ratios as surrogate for the relevant B/A ratios[47]. Since the 2D-projected C and P regions contain different 3D fractions of the B and A regions, a perfect linear correlation between C/P and B/A cannot be expected. However, the reasonably linear correlation between these two parameters supports the hypothesis that the 2D C/P ratio is a valid indicator of 3D B/A ratios[26,46]. While this result lends credibility to the common practice of accepting C/P ratios in human lungs as surrogate for 3D B/A ratios, this finding may be less robust in dichotomous (human) than in monopodial lungs (mouse).

There is currently no clear understanding to what extent the NP agglomeration state affects therapeutic or toxicological responses. In

part this has been due to the lack of a readily available measure of the agglomeration state in the lung. LungVis 1.0 introduces a semi-quantitative characteristic parameter for NP agglomeration state by measuring the apparent NP-volume per deposited NP-dose in the bronchial and acinar regions. In spite of the clearly different regional deposition profiles for bulk liquid and aerosol delivery on the coarse level (C/P or B/A), the apparent agglomeration state is consistent for all delivery routes except for NOAI delivering less agglomerated NPs to the lung. This may be due to the fact that NOAI was conducted with dried aerosol droplets (ca. 0.7 μm in diameter) as compared to all other routes of application using aerosol or spray droplets with diameters of 3 μm or more. On the other hand, the apparent agglomeration state depends on the applied dose and is biased for larger doses by overestimating the geometric volume covered by NPs due to optical artifacts. The difference in agglomeration state for VAAD and NOAI delivery could be at least partially due to the more than 5-fold higher VAAD dose. Thus, LungVis 1.0 can only serve as a semi-quantitative measure of NP agglomeration. The constant B/A ratio of apparent NP-volume per NP-dose below unity suggests that NP agglomeration is larger in the bronchial region, but this effect is independent of the route of application. This might be due to the fact that impaction focuses bronchial deposition to a narrow region near airway bifurcations, while gravitational setting and diffusion spread the NPs over a wider area.

Interestingly, while VAAD and NOAI have identical C/P and B/A ratios (NOAI is considered the gold standard for uniform aerosol deposition), VAAD provides a 20-fold higher delivery efficiency with respect to invested dose than NOAI. Moreover, VAAD completely avoids the typically rather large nasal drug deposition in rodents (nose-to-lung dose ratio >10 in mice)[70], which is crucial for mimicking clinical inhalation therapy with no nasal aerosol deposition. As the VAAD methodology described here combines a rapid-delivery process (40–50 s versus 10–20 mins for NOAI) with uniform-distribution into the deep lung, high-dose efficiency (2.4% versus 0.2%; important for high-cost drugs) and dose-controlled delivery similar to those in clinical settings, it may replace the currently widely used bulk liquid applications and pave the way for introduction of "inhalation therapy" into preclinical drug testing with mouse models. This VAAD inhalation protocol has been implemented to a commercial ventilator (flexiVent FX system, Scireq Inc., Canada), which should facilitate its widespread use in relevant pulmonary delivery fields. All of these insights discovered in this study can be leveraged to select and optimize drug delivery strategies and are expected to enhance the predictive power of preclinical studies for clinical outcome.

LungVis 1.0 highlights a notable time-dependent rise in acinar dose fractions for ITLI and VAAD, alongside a decrease in bronchial NP dose. This is associated with a consistent intra-acinar NP kinetics pattern, regardless of the delivery method: Pulmonary NP distribution evolved from an initial patchy clustering to a more uniform and diffuse dot-like pattern in the lungs. To probe the causes of this NP redistribution, we found that 83–93% of the lung-retained NPs were

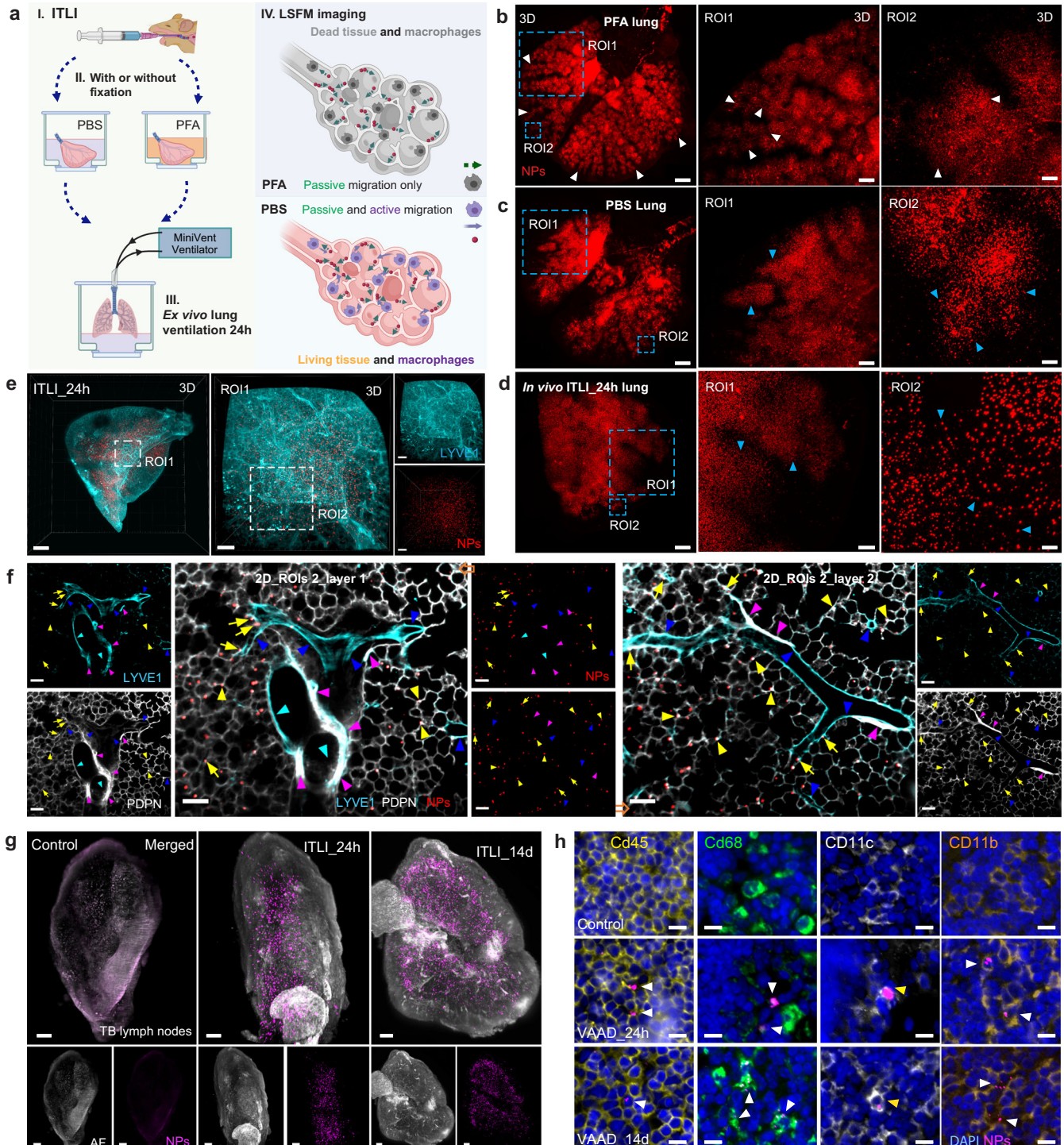

**Fig. 7 | Cellular activity governs phagocytosis-mediated relocation and wide dispersion of NPs throughout different lung networks. a** Schematic illustration of an ex vivo whole-lung ventilation model to investigate the role of cellular activity in NP-lung relocation. Image illustrations created partially with Biorender.com. **b**, **c** Holistic NP distribution in whole lung following 4% PFA fixation or PBS treatment, respectively, succeeded by 24 h of ex vivo ventilation. $n = 4$, independent biological replicates. **d** NP (re-)distribution of an in vivo 24 h ITLI lung is characterized by more uniform distribution (less patchy than at 0 h) and its dot-like pattern. White arrowheads: NP-cluster distribution, and blue arrowheads: single NP-dot distribution. $n = 8$, independent biological replicates. Scale bars (**b**–**d**): 1000 μm (overview), 400 μm (ROI1), and 100 μm (ROI2). **e**, **f** microscopical 3D and cellular-resolution 2D views of NP distribution in the lung and its association with the endothelial systems including the lymphatics and vasculature networks (LYVE1:

lymphatics and blood vessels, PDPN: strong lymphatics and weak alveolar epithelium cell Type I staining). This reveals NPs or NP-laden macrophages in close proximity to the blood vessels (yellow arrows) throughout the entire lung lobe scanned by LSFM. The blood vessels, lymphatics, and airways indicated with arrowheads in blue, purple, and cyan, and the NPs in alveolar epithelium or septum marked with yellow arrowheads. Scale bars: 1000 μm (overview) and 100 μm (ROIs). Representative data from $n = 3$, independent biological replicates. **g** LSFM imaging of tracheobronchial (TB) lymph nodes at 24 h and 14 days after ITLI exposure and untreated control. Scale bars: 100 μm. **h** 2D immunostaining of TB lymph nodes with several TRM markers at 24 h and 14 days post-VAAD exposure, as well as in untreated controls. Arrowheads indicate the NP-laden macrophages in TB lymph nodes. Scale bars: 10 μm. Representative data from $n = 3$, independent biological replicates (**g**, **h**).

ingested by F4/80[+] TRMs in PCLS at 24 h and 14 days after both types of delivery. Those NP-laden TRMs were distributed in various acinar locations, with about 40% in tissue-crossing and inner tissue areas. Notably, the tissue-crossing TRMs were located in the pores of Kohn, which are small alveolar wall openings facilitating inter-alveolar air flow, gas exchange, and immune cell migration[38,53]. Flow cytometry confirms that NPs are predominantly engulfed by AMs (73%), with a smaller portion also by IMs (19%). Lung IVM and ex vivo living microscopy have demonstrated the dynamic behavior of PKH-AMs and CSF1R-EGFP TRMs, actively patrolling the epithelial surface or pores of Kohn to clear NPs from the alveolar tissue. These observations highlight the TRM mobility as a key driver of NP redistribution within the acinar region. Additionally, an ex vivo NP-lung interaction model proves that NP redistribution is driven by cellular activity rather than passive respiration-induced motion, although breathing also stimulates macrophage-mediated NP redistribution. Multispectral LSFM of intact lungs further discloses widely distributed NPs with respect to multiple other lung networks particularly the endothelial systems such as vasculature and lymphatics. All data gathered here suggests that both AMs and -to a lesser extent- IMs contribute to this relocation of NPs. This research challenges the conventional perception of lung TRMs as static entities[37], showing instead their active involvement in intra-acinar NP transport through surveillance, phagocytosis, and migration[38,39]. This activity underscores the significance of active rather than passive processes in the deposition and retention of NPs within the lung interstitium. These insights resonate with recent findings from the cancer field that emphasize active NP transport occurs through transcytosis in endothelial cells and drainage of lymphatic vessels moving beyond the traditional concept of enhanced EPR effects for tumor targeting[62,71]. Our findings underscore the essential role of active mechanisms in NP delivery and call for the development of nanocarrier-based therapeutics that harness the dynamic roles of lung TRMs for targeted drug delivery and release strategies.

LSFM imaging of lavaged lungs shows that a large fraction of NPs remained in the lung post-BAL fluid collection for both ITLI and VAAD administrations. Notably, after collection of BAL from 0 h ITLI lungs, some NP clusters had breached the alveolar epithelium and gathered in the interstitial tissue, whereas 0 h VAAD showed NPs less concentrated in the vicinity of blood vessels. By 24 h post-administration, the NP distribution patterns within and between acini were mostly unaffected by BAL collection. Quantitative analysis in lung homogenates reveals a decrease in the lavageable NP fraction over time following both VAAD and ITLI administrations, with lavaged NP fractions of 0.16–0.30 after 24 h for both delivery, aligning with prior findings for NPs under 100 nm[21,33,34,50]. Prior research indicates differing lavage efficiencies at 24 h post administration for particles smaller than 100 nm and larger than 500 nm, with efficiencies around 0.2 and 0.8, respectively, with no information on the size range between 100 and 500 nm. Interestingly, our 480 nm melamine NPs showed a similar lavaging efficiency of 0.2 as sub-100 nm NPs, suggesting that the transition point from 0.2 to 0.8 lavaging efficiency is very close to 500 nm. However, this cannot be solely attributed to size-dependent uptake efficiency of NPs[32,33,55], as more than 85-96% of our MF NPs were taken up by TRMs. We found that within 14 days, 73–77% of the administered NPs were cleared out of the lung in both VAAD and ITLI administrations. The initial clearance of NPs within 24 h was primarily due to direct mucociliary clearance, while at 14 days, long-term clearance was attributed to macrophage uptake and patrolling and subsequent mucociliary clearance[21,29,31,72]. However, passive diffusion of NPs from the epithelium to the interstitium was found to be an insignificant factor on the relocation of MF NPs. Instead, this study suggested that quick uptake of MF NPs by TRMs within 24 h after application and likely the slow recycling of NP-laden macrophages back to the ciliated airway epithelium, followed by elimination through mucus, dominated long-term MF NP clearance in the lung.

The LungVis 1.0 ecosystem has its limitations, including the semi-quantitative nature of quantifying NP spatial profiles due to variable attenuation of fluorescence intensity in LSFM images, and no direct observation of NP deposition within lung blood vessels and lymphatics. LungVis 1.0, however, integrates AI and deep learning pipelines with 3D lung imaging for gaining systematic insights into the bronchial and acinar NP dose, whole lung NP profiling, and associated immune responses in murine lung. Synergistic insight can be obtained from real-time tracking of individual macrophages in living animals (IVM), a ventilated ex vivo NP-lung interaction model, ex vivo PCLS, and whole-mount staining and multispectral imaging. The repository of raw LSFM lung images, reference annotations, AI segmentations of lung airways, and AI training models developed with LungVis 1.0 are available for analyzing spatial dose and distribution of various fluorescence-labeled substances, including drugs, vaccines, and antibodies in mouse lungs. Leveraging the stain-free nature of LSFM imaging, we are expanding our focus to include additional lung network systems like the vasculature, aiming to establish a comprehensive LungVis ecosystem. This ecosystem will facilitate unbiased, 3D evaluation of physiopathological changes and therapeutic effects across multiple lung networks in murine models, greatly advancing the development of precision inhalation therapy.

## Methods

### Ethics statement

Animal studies were conducted in accordance with the relevant guidelines and protocols approved by the ethical review board of the District Government of Upper Bavaria (Regierung von Oberbayern, AZ55.2-1-54-2532-108.13 and AZ55.2-1-54-2532-67-2015), following the regulations of the European directive 2010/63/EU for animal research and institutional guidelines of Helmholtz Munich.

### Animal handling and pulmonary delivery methods

Wildtype (WT) C57BL/6 female and male mice were purchased from Charles River, Germany and were bred in house. WT mice with age 9–18 weeks and 19–25 g of body weight (BW) were housed in individually ventilated cages (IVC-Racks; Bio-Zone, Margate, UK) supplied with filtered air in a 12 h light / 12 h dark cycle. Mice were provided with food (standard chow) and water ad libitum. A macrophage colony-stimulating factor receptor enhanced green fluorescent protein transgene (CSF1R-EGFP) is predominantly expressed throughout the mononuclear phagocyte system of the mouse (strain # 018549, purchased from The Jackson Laboratory)[73]. The MacGreen colony (age 10–12 weeks) was maintained as homozygote and all offspring were positive for an enhanced green fluorescent protein that was subsequently used for ex vivo living tissue imaging and tissue-cleared LSFM.

Intratracheal liquid instillation (ITLI) was applied to deliver a certain amount of MF particles (50 µL of 1:50 dilutions of MF stock suspension in 0.9% saline water) to the mouse lung as previously described[26]. Mice were anesthetized by the intraperitoneal injection of a MMF mixture (medetomidine, midazolam, and fentanyl: 0.5, 5, and 0.05 mg kg[−1] BW) and a 20 G cannula were inserted into mouse trachea, allowing for direct administration of MF to the mouse lung. Mice were divided into 4 groups randomly and sacrificed at consecutive time points of 0 h, 2 h, 24 h, and 14 d post-administration. It took about 3 min from lung instillation to organ withdrawal for 0 h ITLI lungs, since the lungs were immediately isolated after ITLI application. The ITLI delivery process lasted about 5 s and the lungs were perfused and isolated immediately for 0 h lungs.

Ventilator-assisted aerosol delivery (VAAD) was used to transport liquid aerosols to the mouse lung as previously described[26]. Briefly, the mice were anesthetized as above and connected to a 20 G cannula with tubing extended to a mechanical ventilator (flexiVent FX system, Scireq Inc., Canada), enabling precise control of its respiration during aerosol inhalation. Meanwhile, the ventilator was equipped with a

nebulizer (Aeroneb Lab Small, Aerogen Inc., Galway, Ireland) generating liquid aerosols with sizes 2.5–4 μm to be transported to mouse lung (applied dose: 20 μL 1:3 dilutions of stock suspension). The VAAD delivery method follows the custom-designed HMGU protocol developed in Helmholtz Munich, Germany, maximizing the lung deposition dose and deep-lung delivery. The mice were mechanically ventilated with HMGU protocol at 120 breaths/min, 0.4 mL tidal volume, and an inspiration-expiration time ratio of 2:1 with nebulizer activation of 40 ms/breath at the onset of inspiration. Mice were divided into 4 groups randomly and sacrificed at consecutive time points of 0 h, 2 h, 24 h, and 14 days post-administration. The VAAD delivery process lasted about 40–50 s and the lungs were perfused and isolated immediately for 0 h lungs.

Intranasal liquid aspiration (INLA) was performed after the mice were anesthetized by 5% isoflurane in a sealed exposure chamber as described by Wu and co-authors[74]. The 50 μL 1:50 diluted MF solution was slowly dispensed into one nostril using a pipette allowing for spontaneous inhalation to the mouse lung via the nose. The mice were applied via INLA and lung samples were collected at 0 h after application. The INLA delivery process lasted about 15–30 s and the lungs were perfused and isolated immediately for 0 h lungs.

Nose-only aerosol inhalation (NOAI) enables a physiologically realistic means of drug delivery on non-sedated mice leading to a spatially uniform drug distribution throughout the lung. Using a Pari Sprint LC nebulizer (volume median droplet diameter ca.2 μm) filled with 2 mL 1:10 MF NP suspension (i.e., each 2 μm drop of suspension will contain only one (or none) NPs as the probability for a 2 μm drop to contain a NP is only 18%) 1.14 mL of this suspension was aerosolized and delivered to mouse nostrils via an in-house newly developed nose-only inhalation system allowing for simultaneously delivery to 6 mouse ports in parallel[75]. NOAI mice were sacrificed at 0 h post administration. The NOAI delivery process lasted about 10–15 mins and the lungs were perfused and isolated immediately for 0 h lungs.

## Nanoparticles and Characterizations

Melamine resin fluorescence particles (MF, ex/em = 636 nm/686 nm; volume median diameter (VMD): 474 nm, stock suspension: 25 mg mL$^{-1}$, microParticles GmbH, Berlin, Germany) were employed as the imaging tracer for multiple delivery routes and longitudinal observations under LSFM as described in our previous studies[20,26]. Native preparations of MF NPs were examined by transmission electron microscopy (TEM, Zeiss Libra 120 Plus, Carl Zeiss NTS GmbH, Oberkochen, Germany). Pictures were acquired at 4000-12500x magnification, using a Slow Scan CCD-camera and iTEM software (Olympus Soft Imaging Solutions, Münster, Germany).

## LungVis 1.0 data acquisition: Tissue clearing and light sheet fluorescence microscopy (LSFM)

Followed by mice sacrifice at 4 time points (0 h, 2 h, 24 h, and 14 days) after ITLI and VAAD exposures, lung perfusion via right ventricle with 10 mL 1 × PBS, and organ harvest, LungVis 1.0 non-dissected lung specimens were undergoing an optical tissue clearing process as prior described[20,26]. Briefly, tetrahydrofuran (THF, Sigma 186562- 1 L), dichloromethane (DCM, Sigma 270997-1 L) and dibenzyl ether (DBE, Sigma 108014-1KG) were used for dehydration, lipid removal, and refractive index matching, respectively. Optically transparent lung samples were then scanned with LSFM (Ultramicroscope II, LaVision Biotec) equipped with a sCMOS camera (Andor Neo) and a 2× objective lens (Olympus MVPLAPO 2×/0.5 NA) equipped with an Olympus MVX-10 zoom body. The whole-lung scans were generated with 0.63× zoom magnification with specific excitation/emission bandpass filters for MF particles and lung tissue autofluorescence at 640/690 nm and 520/585 nm or 740/790 nm, respectively with a step size of 10–15 μm. High magnification scans like 2× or 6.3× were further performed with smaller step sizes of 3–7 μm to obtain clearer views of cellular

deposition patterns of delivered particles. To achieve the best imaging quality for 78 murine lungs presented here, LSFM instrumental settings were adjusted for each delivery method and longitudinal measurement to account for the up to 15-fold variation in deposited NP dose. It is also important to note that optimal and individualized LSFM imaging is essential for accurate spatial profiling of NPs in the lung, such as the C/P and B/A dose ratios.

## LungVis 1.0 AI pipelines: Active learning AI-driven airway segmentation

The automatic segmentation of airways has been accomplished using nnU-Net[42], which is a self-adapting framework that automatically adapts a U-Net-based segmentation pipeline[41] to the dataset it is applied to. This is made possible by creating a dataset fingerprint which is then used to find a suitable hyperparameter configuration. The nnU-Net fingerprinting uses rules and heuristics based on the characteristics of the dataset such as the modality, intensity distribution, the median shape as well as the distribution of spacings to derive well working hyperparameters. For airway segmentation, the first training using the initial three labeled lungs was done out-of-the-box without additional adaptations (Fig. 1c). This means that the hyperparameters described in the following were completely determined by the nnU-Net fingerprint. Briefly, the nnU-Net uses a U-Net-like encoder-decoder architecture with skip-connections. Images are normalized using a z-score transformation with per image mean and standard deviation. The model is trained for 1000 epochs where each epoch consists of 250 batches (each consisting of 2 patches). Due to the large image size, training is performed in a patch-based fashion, where smaller crops of size 48 × 224 × 224 voxels are sampled from the training images. One batch consists of two patches, where one patch is taken from a random location and the other is guaranteed to contain annotated airway pixels. The loss function is the sum of the Dice and cross-entropy loss. During training the model is optimized using the SGD optimizer with an initial learning rate of 0.01, which is decayed using a poly learning rate schedule. Data augmentation (or parameter adaptation) contains rotations, Gaussian noise and blur, brightness and contrast transformations, gamma corrections, simulation of low resolution and mirroring. For inference a sliding-window approach is utilized with half-patch size overlap as well as a Gaussian patch center weighting. Moreover, mirroring along all axes is applied as a test time augmentation.

Throughout the data-centric model development further dataset-specific adjustments were employed. The target spacing, which nnU-Net automatically sets to the median spacing of each axis, was doubled to decrease the image size by a factor of 2, allowing a more efficient training. Moreover, the data augmentation pipeline for training has been adapted to the relevant imaging artifacts present in the dataset. A custom slice illumination transform was used for simulating dark lung centers caused by illumination issues. Also blurring, contrast, brightness and sharpness transforms were applied locally in images to imitate blurred areas, halos, and different signal-to-noise ratios. Lastly, blank rectangle transforms were used to mimic missing information due to artifacts in the images. Using the additional augmentations, we were able to train a model that is highly robust.

Once the model was trained, unlabeled data was predicted and resulting airway structures were assessed. Afterwards, selected images were corrected and used for another iteration of model training. Manual ground truth annotations (3 full lungs) and interactive correction of selected images were accomplished with a segmentation workflow using ImageJ (https://imagej.nih.gov/ij/) with the processes of e.g., setting an intensity threshold to obtain binary 2D images, applying a cut function to clean false airway structures, and using the paintbrush tool to draw missing airways and to close holes. The labeling of the initial images was performed by four lung specialists. Labeling a whole lung consisting of 200–500 images with pixels of e.g.,

2048 × 2048 for each image takes a few weeks even for lung anatomists. Consequently, only correcting predictions of our AI model instead of labeling a whole lung highly increased time efficiency. The selection of cases for a manual correction was based on a visual inspection of all validation images. Selected examples needed to have severe segmentation errors. However, diversity within the types of errors as well as dataset properties was considered to cover a maximum range of possible errors in each iteration, leading to improved models after every training iteration. Quality control in the form of cross-examination and corrections was implemented, resulting in a highly precise 3D segmentation of complete airway trees.

### LungVis 1.0 visualization and quantification: Spatial profiling of delivered NPs

Visualization of NP delivery features: Spatial profiling of NPs in the lung is primarily based on 2D single slice/cross-sectional LSFM images which were processed into 3D maximum intensity projection (MIP) images using Fiji/ImageJ v1.53. 3D volume reconstruction (rendering) images and movies with manipulation were generated using Bitplane Imaris (versions 9.5.1 and 9.9.1). With the complete 3D lung airway structure generated from LungVis 1.0 AI pipelines, the NP dose in the airway (bronchial) and non-airway (acinar/alveolar) regions can be readily visualized and quantified. Briefly, by overlaying the AI-driven airway masks (binary images: 1 or 0 indicate pixels inside or outside the airway tree) with raw tissue autofluorescence or NP fluorescence images slice-by-slice over the entire lung LSFM stack, one can visualize both airways (green; tissue autofluorescence) and NPs in airways as 3D MIP images (see images of "Airways (AF)" and "Airway NPs" in Fig. 2, Fig. 3, Fig. 4 and Supplementary Fig. 7). Similarly, the acinar deposited NPs can also be visualized outside of the bronchial tree as indicated in the 3D MIP images of "Acinar NPs" of Fig. 2, Fig. 3, Fig. 4, and Supplementary Fig. 7. By merging the images of "Airway NPs" or "Acinar NPs" with corresponding regions from the AF channel ("Airways AF"), NP deposition in airway or acinar regions is displayed as shown in Fig. 2, Fig. 3, Fig. 4, and Supplementary Fig. 7.

As mentioned above, optimal LSFM imaging with appropriate instrument settings is essential for each lung to profile the spatial patterns of NP deposition qualitatively and quantitatively (e.g., C/P and B/A dose ratios) after four delivery routes and at various time points. It is important to note that NP total intensity in LSFM images (Fig. 2, Fig.3, Fig. 4, Supplementary Fig. 5a, b, Supplementary Fig. 7) does not necessarily reflect the deposited NP dose, which is accurately determined from spectrofluorimetric analysis of lung homogenates (described below). This information is shown in Fig. 2a. To prevent misinterpretation of NP dose in LSFM images, the NP intensity was therefore scaled to relatively match the lung homogenate-based dosage ranking in the whole lung or left lung images shown in Figs. 2d–g, 3a–o, 4a–o, Supplementary Fig. 5a, b, Supplementary Fig. 7.

Quantification of NP delivery features: To quantify NP dose fractions in the airway and acinar regions of the lung, we measured the total NP fluorescent intensity using 3D MIP images of these regions (Supplementary Fig. 6a–g). The NP signals were quantified in 3D using ImageJ and Bitplane Imaris, following the methodology outlined in our previous studies[26,76]. Briefly, an intensity threshold was set to eliminate tissue AF from the NP-related intensity, and the bronchial and acinar NP signals were calculated in separated regions of "Airway NPs" and "Acinar NPs" using ImageJ/Bitplane Imaris (Supplementary Fig. 6d–f). Typically, these threshold values are varying somewhat depending on the lung and/or operational parameters during image acquisition, but they are at least two-fold lower than the mean fluorescence intensity of all positive NP pixel intensities and 5 to 50 times lower than the highest NP pixel intensity. 3D volume reconstructions of AI-driven segmented airways, "Airway NPs", and "Acinar NPs" were further built in Imaris to determine the total airway volume ($\mu m^3$), bronchial NP signal, and

acinar NP signal, respectively. The bronchial and acinar NP dose fractions (Eq. 1) and their B/A dose ratio can then be computed from both software tools with consistent results (Fig. 2r and Supplementary Fig. 6h, i).

$$\begin{aligned} & Bronchial\ or\ acinar\ NP\ dose\ fraction \\ & = \frac{Bronchial\ or\ acinar\ NP\ fluorescence\ signal}{(Bronchial + Acinar\ NP\ fluorescence\ signals)} \end{aligned} \quad (1)$$

Apparent NP-positive volumes in both regions ("Airway NPs" and "Acinar NPs") were computed directly either in 3D using Imaris or in 2D slice-by-slice using Fiji/ImageJ or Anaconda (Python), with units in cubic micrometers ($\mu m^3$). Given the known total lung deposited dose ($\mu g$) from spectrofluorometric analysis in lung homogenates and the NP dose fractions in bronchial and acinar regions obtained from the LSFM images, the exact doses for bronchial and acinar regions are provided in Supplementary Data 1. From this, the apparent NP-volume/dose can then be determined in each region, which is a relative measure of the NP agglomeration state representing the inverse packing density after delivery (Eq. 2), since this agglomeration parameter is largest for complete spreading (i.e., single NP spheres) and smallest, if all NPs are closely packed into one large agglomerate. The apparent NP-volume normalized to bronchial or acini NP dose, as well as the corresponding B/A ratio is provided in Supplementary Fig. 6j–l for each of the four application routes.

$$\begin{aligned} & NP\ agglomeration\ state \\ & = \frac{(B\ or\ A)\ Apparent\ NP\text{-}positive\ volume\,(\mu m^3)}{(B\ or\ A)\ NP\ dose\,(\mu g)} \end{aligned} \quad (2)$$

### Traditional NP spatial deposition analysis

Quantitative analysis of C/P deposition ratio using ImageJ (Fig. 2q): the lung regional deposition pattern was determined with a conventional approach, the C/P ratio. Briefly, central and peripheral lung regions are defined as 50% of inner and outer area in the 3D MIP images, respectively. C/P ratio then determines the ratio of aerosol dose deposited to the central lung region to that of the peripheral lung area. C/P ratio provides important insights into the distal *versus* proximal lung aerosol distribution but has no discrimination of aerosol accumulation between airways and acini since both central and peripheral regions consist of bronchial and alveolar regions.

### Fluorescence-based particle dosimetry in the lung homogenates

The BAL fluid, trachea, lung lobe specimens were separately collected to determine respective dose in each compartment at all time points (0 h, 2 h, 24 h, and 14 d) as previously described[26]. The tissue samples were mechanically homogenized at a 1:10 (m/v) ratio of tissue to 1× PBS using a disperser (T10 basic ULTRA-TURRAX®) until no visible tissue pieces were observed. The particle tissue burden was then calculated using the standard curve of several known doses of MF particles added into blank tissue samples to the fluorescent intensities measured with a standard multi-well plate reader (Tecan Safire 2) at the optical wavelength of 635/685 nm with a 10 nm bandwidth of optical filters. The 1 mL BAL fluid from each mouse lung experienced several rounds of rapid freezing and thawing (to release the particles from the cell pellet by breaking the cell membranes), and vortexing. Subsequently, fluorescence analysis was performed by setting up the standard curve, which was prepared from the BAL fluid lavaged from blank lungs (no particle exposure) according to the same protocol used above for tissue samples[77]. The delivery efficiency for each application was then determined by the total pulmonary-deposited dose (including all doses from BAL fluid, lavaged trachea and lavaged lungs) normalized to the initially invested dose, where the latter refers to the amount of liquid filled into the application device which is subsequently emptied

as much as possible. Three values for INLA[74] and four values for VAAD[26] were additionally included to analyze the pulmonary delivery efficiency.

## Lung inflammation analysis

To evaluate the particle-induced lung inflammation, the influx of polymorphonuclear neutrophils (PMNs) as the most often used parameter in determining particle toxicity into the lung alveolar compartment was assessed after various exposure routes[57]. Bronchoalveolar lavage (BAL) fluid were then collected after VAAD, ITLI, and saline exposures at 4 time points of 0, 2, 24 h and 14 d to determine the recruitment of multiple inflammatory cells (PMN, monocytes/macrophages, and lymphocytes) by counting the respective cell counts and frequencies[78,79]. Lungs were lavaged by infusing with $8 \times 1$ mL PBS (Gibco, Life Technologies) and 7 mL BAL fluid was used to collect the cells by centrifugation (400 g, 20 mins, 4 °C) and the residual (1 mL) was employed to determine the particle dose in BAL fluid (4 extra ITLI lungs were solely used to determine the PMN influx). Pellet cells were resuspended in RPMI-1640 medium (Gibco, Life Technologies) supplemented with 10% fetal calf serum (Seromed, Berlin, Germany), and living cells were counted by the trypan blue exclusion method using a haemocytometer. Two glass slides were prepared for cytospin of each mouse (approximately 30,000 cells into each cytospin device) and the residual pellet cells were used to quantify particle fraction in macrophages by flow cytometry (mentioned below). BAL cell differentials were evaluated based on morphological criteria on cytospin preparations (May-Grünwald-Giemsa staining; $2 \times 200$ cells counted). Quantification of NP[+] macrophages in all collected BAL cells (cytospin slices, Supplementary Fig. 13a, b, e) after May-Grünwald-Giemsa staining was performed independently by several lung specialists. NP[+] macrophages to all collected macrophages in 2 slices of each mouse was determined.

## Generation of 3D precision cut lung slices (PCLS) for immunofluorescence (IF)

To generate the PCLS specimens after VAAD and ITLI deliveries at 2 h, 24 h and 14 d, the mice were anesthetized by the intraperitoneal injection of a ketamine and xylazine mixture and then intubated with a 20 G cannula inserted into the trachea, as previously described[26,80]. Mice were transcardially perfused with 10 mL of 1× PBS and their lungs were filled with about 1 mL warm, low-melting agarose (2 wt%, kept at 40 °C; Sigma) in sterile cultivation medium [DMEM-F-12 (GIBCO) supplemented with penicillin-streptomycin and amphotericin B (both Sigma)] via the cannula-intubated trachea. The trachea was tightly closed with a thread until the agarose cooled down to stiffen the lung tissue and maintain the lung inflation state. The lobes were separated and cut with a vibratome (Hyrax V55, Zeiss, Germany) to a thickness of 300 µm. The generated PCLS sections were washed twice in 1× PBS and fixed in 4% paraformaldehyde (PFA) in PBS for 30 min, and permeabilized in 4% PFA containing 0.3% Triton X-100 in PBS for 5 min. Primary antibodies were diluted in 1% bovine serum albumin (BSA; Sigma) in PBS, incubated for 16 h at 4 °C, and subsequently washed three times with PBS for 5 min each. Secondary antibodies were diluted in 1% BSA (Sigma) in PBS, incubated for 4 h at room temperature, and subsequently washed three times with PBS for 5 min each. During multi-tracked imaging using a laser scanning microscopy (LSM 710, Zeiss), the stained, wet PCLS were placed in the sealed glass bottom dish 35 mm (ibidi, Germany) and scanned with 10X, 20X and 40X objectives with z-stack sizes ranging from 50–200 µm. The following primary antibodies (1) and secondary antibodies (2) were used: (1) anti-F4/80 (ab90247, 1:100, rat; Abcam), LYVE-1 (ab14917, 1:100, rat; Abcam), Podoplanin (AF3244, 1:200, goat; R&D systems), a-SMA (A5228, 1:5,000, mouse; Sigma), and (2) goat anti-mouse IgG Alexa Fluor-488 (1:200; Invitrogen), goat anti-rat IgG Alexa Fluor-488 (1:200; Invitrogen), donkey anti-mouse IgG Alexa Fluor-488 (1:200; Invitrogen),

donkey anti-rat IgG Alexa Fluor-488 (1:200; Invitrogen), and donkey anti-goat IgG Alexa Fluor-567 (1:200; Invitrogen). Cell nuclei were stained with DAPI (40,6-diamidino-2-phenylindole, D9564-10 mg, 1:2,000; Sigma) and actin stress fibers with Alexa Fluor™ 594 Phalloidin (A12381, 1:300; Invitrogen).

Particle quantification in PCLS was performed independently by lung specialists (Fig. 5d–f). NP fraction in NP[+] macrophages was determined by the fluorescent intensity of NP[+] within macrophages normalized to the total NP fluorescent intensity in three regions of interest (ROIs) per lung. According to their localization in the immunostained PCLS section, macrophages were categorized into 4 groups: epithelium-surface macrophages (I), epithelium-attached macrophages (II), tissue-crossing macrophages (III), and inner-tissue macrophages (IV). The fraction of NP[+] to total macrophages and ratio of each NP[+] macrophage group (based on their locations) in three ROIs per lung were counted.

## Immunofluorescence staining and microscopy of tracheobronchial lymph nodes

Formalin-fixed paraffin-embedded sections of TB lymph nodes of VAAD mice were prepared for IF staining[76]. Briefly, the sections were dried at 60 °C for 15 mins, deparaffinized with xylene (2×, 3–5 min each), and rehydrated through ethanol series (2 × 100%, 90%, 80%, 70%, 2–5 min each) before washing in 1X PBS. Antigen retrieval was performed using 10 mM citrate buffer (pH 6.0) in a pressure cooker (30 sec at 125 °C, 10 s at 90 °C). Sections were blocked with 10% normal donkey serum and incubated overnight at 4 °C with primary antibodies (CD11b 1:2000, ab133357, Abcam; CD11c 1:200, #97585, Cell signaling; CD68 1:100, ab125212, Abcam; CD45 1:250, ab10558, Abcam). The sections were further incubated with a secondary antibody (donkey anti-rabbit Alexa Fluor 488, 1:200, A21206, Invitrogen) and DAPI for 2 h at room temperature, followed by imaging using the Axioscan 7 slide scanner (Zeiss).

## Flow cytometry

BAL cell preparation: The residual pelleted cells (about 0.2–0.45 million cells) collected from 7 mL BAL fluid of each lung (1 mL BAL fluid and the lavaged lungs were used for particle dosimetry described above) at four time points were prepared for flow cytometry. The BAL pellets were mixed with 1 mL MACS buffer (0.5% fetal calf serum (FCS) and 2 mM EDTA in 1× PBS) and centrifuged at 300 g for 5 min. The supernatant was discarded and the cells were stained with 100 µL CD11c-FITC antibody or isotype control (1: 50, 130-110-837, Miltenyi Biotec) for 15 min at 4 °C in the dark. The cells were fixed by adding 100 µL 8% PFA into the staining tube for 20 min at RT. The cells were washed in 1x MACS 300 g for 5 mins at 4 °C and 200 µL MACS were added to the samples.

Lung tissue single cell preparation: lung single cell suspensions were prepared as previously described[81,82]. MacGreen and WT mice were used to analyze the particle ingestion by tissue resident macrophages (CD11b[+] and Siglec F[+]). Briefly, oropharyngeal aspiration of PKH dye (0.5 µM) into the WT mouse lungs were applied at least 5 days prior to the ILTI application and lung samples were collected at 24 h after ITLI application with 50 µL 1:25 MF particles. Lavaged lungs (3 MacGreen and 3 WT mice), non-lavaged lungs (4 MacGreen and 5 WT mice) and single staining controls were transferred for enzymatic digestion for 30 min at 37 °C in a mixture of dispase (50 caseinolytic U mL⁻¹), collagenase (2 mg mL⁻¹), elastase (1 mg mL⁻¹), and DNase (30 µg mL⁻¹). Single cell suspensions were harvested by softly smashing tissue through 100 µm mesh using the plunger of a 3 mL syringe and then followed with filtering through a 40-µm mesh. After centrifugation at 300 g for 5 min, red blood cells were removed with 1 mL lysis buffer for 30–40 sec. Single lung cells were then generated with addition of 5–6 mL RPMI + 10%FCS to the sample tube, washed in 1 mL 1× fluorescence-activated cell sorting (FACS) buffer, and centrifuged for

5 mins at 300 g at 4 °C, and counted for total cell numbers and overall cell viability as described above. The single cell numbers were counted at most 1 million per 100 μL FACS buffer/tube and incubated with 50 μL Fc block (CD16/CD32, 1:100, 14-0161-82, ThermoFisher) in FACS buffer for 20 mins at 4 °C. After 30 min incubation with antibody suspension for 30 mins at 4 °C, the single cell suspensions in 200 μL of MACS were then analyzed using BD FACSFortessa running FACSDiva software v8.0.1. Cell types differentiation and particle quantifications were performed using FlowJo v10.8.1. FACS antibodies used for lung tissue single cell suspension are as follows: CD11b-PE (1:50, 130-113-806, Miltenyi Biotec), SigelcF-PE-Vio770 (1: 50, 130-112-334, Miltenyi Biotec), and CD11b-PE-Vio770 (1:50, 130-113-808, Miltenyi Biotec).

## Ex vivo 4D living tissue microscopy
For ex vivo living tissue imaging, MacGreen mice were sacrificed immediately after application of 50 μL 1 mg mL⁻¹ MF suspension via ITLI and the cardiac perfusion through the right ventricle and pulmonary artery was performed as mentioned above. The lungs were then extracted from the mouse body and kept in sterile cultivation medium (DMEM/F12, Gibco, Germany, supplemented with penicillin/streptomycin and amphotericin B; both Sigma) until sample preparation for live imaging[39]. The left lung lobe was separated and punched with a 5 mm biopsy puncher, producing a lung core biopsy covering the entire thickness of the lung lobe with a core-diameter of approximately 5 mm. Next, the lung core biopsy was mounted into a "size 4" glass capillary (Zeiss) containing 1% low-melting agarose and then put into the sample holder of the light-sheet fluorescence microscope (Lightsheet Z.1 equipped with an incubation system, Zeiss). The lung core biopsy was submerged into the sample chamber containing phenol-red-free DMEM/F12 (Gibco) and 1% FBS. The sample chamber was heated to 37 °C and 5% CO₂ was applied. After sample positioning, Z-stacks were acquired for every 20 min using a water-dipping 20× Plan-Apochromat objective (Zeiss). For subsequent analysis, all z-stacks were converted into a MIP image by using ZEN2009 software (Zeiss). Confocal time-lapse microscopy was implemented on an LSM710 system (Zeiss) containing an inverted AxioObserver.Z1 stand equipped with phase-contrast and epi-illumination optics and operated by ZEN2009 software (Zeiss). Here, lung tissue was used as precision-cut lung slices (PCLS), which were submerged in phenol-red-free DMEM/F12 (Gibco) and 1% FBS in an imaging cell-culture dish with a glass-bottom. This imaging dish was placed into a PM S1 incubator chamber (PeCon/Zeiss) and kept at 37 °C and 5% CO₂. Z-stacks were acquired at a resolution of 512 × 512 pixels for every 20 min using an EC Plan-Neofluar DICI 10×/0.3 NA (Zeiss). Z-stacks were taken according to the thickness of the PCLS and were ranging between 150 and 300 μm. For subsequent analysis, all z-stacks were converted into a MIP image by using ZEN2009 software (Zeiss).

## Lung intravital microscopy (IVM)
Surgical preparation of lung IVM was performed as described by Headley et al[83]. with minor modifications. Briefly, mice were anesthetized by intraperitoneal injection of MMF with body temperature maintained at 37 °C using a heating pad. After loss-of-reflexes, Bucaine (50 μg per site) was administered before surgical incisions via subcutaneous injection. Tracheostomy was performed to insert a tracheal cannula that connected to a mechanical ventilator (MiniVent, Harvard Apparatus) equipped with a nebulizer used for MF application AerogenPro nebulizer (Aerogen Small VMD, Kent Scientific Corporation, USA). Mice were ventilated with a stroke volume of 10 μL g⁻¹ BW, a respiratory rate of 130–140 breaths per minute, and a positive-end expiratory pressure of 0.1 cm H₂O per gram BW. The mice were then placed in right lateral decubitus position and a custom made flanged thoracic suction window was inserted into a 5 mm intercostal incision through the parietal pleura between ribs 3 and 4 of the left chest. 20–25 mm Hg of suction was used to immobilize the lung by a custom-

made system consisting of a differential pressure gauge (Magnehelic, Dwyer Instruments, nc, USA) and a negative pressure pump (Nupro, St Willoughby, USA).

Lung IVM was performed with a VisiScope.A1 imaging system (Visitron Systems GmbH, Puchheim, Germany), equipped with an LED light source for fluorescence epi-illumination (pe-4000, CoolLed, Andover, UK). For optical excitation LED modules 488 nm, 550 nm and 655 nm, were applied with 50% output power and 50 ms exposure time. Light was projected onto the sample via a quadband dichroic filter set (DAPI/FITC/Cy3/Cy5 Quad LED ET Set; AHF Analysentechnik AG, Tuebingen, Germany). Microscopic images were obtained with a water dipping objective (20x, NA 1.0, Zeiss, Oberkochen, Germany). Light from the specimen was separated with a beam splitter (T 580 lpxxr Chroma Technology Corp., Bellows Falls, VT, USA) and acquired with two Rolera EM2 cameras and VisiView Imaging software (Visitron)[84]. For visualization of TRMs, the PKH26 dye (PKH26 Red Fluorescent Cell Linker Kit for Phagocytic Cell Labeling, Sigma-Aldrich, Germany) was applied at least 5 days before imaging via oropharyngeal application[38]. For particle inhalation, 40 μL 1:50 diluted MF particles were added to the nebulizer. Anaesthetized animals were euthanized by cervical dislocation at the end of the experiments.

## Ex vivo lung model for NP-redistribution analysis
Ex vivo lung ventilation model is a custom-designed, simplified system according to the prior ORVD delivery model[61]. Intact lungs were immediately extracted and submerged in Falcon tubes for either 3–4 h cross-linking fixation in 4% paraformaldehyde (PFA) or in 1× PBS as a positive control at room temperature after instillation with 50 μL 1:50 diluted MF particles (Fig. 7a). This step is functionally and manually similar to the ex vivo zombie fixation and nanoparticle perfusion system[62], aiming to block the active particle transportation by cellular activity (this study: phagocytosis and migration of TRMs, prior study: endothelial cell mediated particle transportation). After fixation or PBS incubation, the intact lungs were rinsed in PBS and placed into a custom-designed bioreactor with temperature control (37 °C or room temperature) and mechanically ventilated (MiniVent, Harvard Apparatus) with a physiologically frequency of 120 strokes per minute and 150 μL stroke volume over the 24 h exposure. This was done to mimic the spontaneous breathing of mice. The lung blocks were placed in horizontal or vertical positions, with only the tissue tips submerged into the 1× PBS to prevent tissue drying during ventilation. Following this procedure, the lungs were fixed overnight, processed with tissue clearing agents, and then imaged using LSFM.

## Whole-mount immunostaining and tissue-cleared LSFM imaging
The whole-mount IF staining protocol for complete lung lobes was adapted from the previous protocols[67]. Briefly, the lung lobes were incubated in PBSGT (PBS with 0.2% gelatin, 0.5% Triton X-100, 0.01% thimerosal) for 1-3 days blocking and permeabilization with rotation at RT. The lung lobes were then incubated with primary antibodies in PBSGT + 0.1% saponin (10 μg mL⁻¹) for 3–7 days at RT with rotation (70 rpm) and rinsed in PBS 1X + 0.5% Triton (PBST), 6 times during 1 day at RT with rotation. Secondary antibodies in PBSGT + 0.1% saponin (10 μg mL⁻¹) were incubated for 1–3 days at RT with rotation (70 rpm) and followed with 6 times washes in PBST during 1 day at RT with rotation. Samples could be conserved at 4 °C until optical clearing. Due to the quenching of fluorophores (i.e., DAPI) by optical tissue clearing agents, additional antibodies, dyes, or fluorophores were applied here as following: anti-GFP (ab13970, 1:1000, Chicken; Abcam), goat anti-chicken IgG Alexa Fluor-488 (1:1000; Invitrogen), propidium iodide (PI, P3566, 1:1500; Invitrogen), and VivoTag-S 750 fluorochrome (NEV10124, PerkinElmer) labelled lectin (L2380, Sigma) for vasculature staining. Tissue clearing was then applied to the lung specimens with the following procedures: 50% THF overnight, 50%, 80%, and 100%

THF each for 1 h, 100% THF overnight, 100% THF 1 h, DCM for 20-40 min, DBE for at least 4–6 h until LSFM imaging (Ultramicroscope II, LaVision Biotec).

## Statistical analysis

All data are presented as mean ± standard deviation (SD) and plotted using GraphPad Prism v9 (GraphPad Software, San Diego, CA). Comparison of results between two groups for normal distributed data was carried out using a two-sided or one-sided student t-test. Comparison of results between two groups with multiple subgroup comparisons were calculated using multiple unpaired *t*-tests with Holm-Šídák correction for multiple comparisons (alpha = 0.05). Comparisons of more than two groups were performed using a one-way analysis of variance (ANOVA) followed by pairwise multiple comparison procedures (Holm-Šídák method). Significances are defined as exact values in the graphs.

## Reporting summary

Further information on research design is available in the Nature Portfolio Reporting Summary linked to this article.

## Data availability

The data supporting the finding of this study are available within the article and its Supplementary information. Any additional information required to reanalyze the data is available from the corresponding author upon request. The raw and AI-enabled LSFM imaging datasets, reference lung annotations, and AI-generated airway models generated from LungVis 1.0 are publicly available via Zenodo (https://doi.org/10.5281/zenodo.7413818)[43]. Source data are provided with this paper.

## Code availability

All computing codes for the data-centric active learning AI training model generated in this study are provided via GitHub (https://github.com/MIC-DKFZ/MurineAirwaySegmentation)[44].

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

## Acknowledgements

This research was partially funded through the EU Horizon 2020 research and innovation program under grant agreement no. 686098 (project SmartNanoTox, O.S., L.Y.) and grant agreement No. 953183 (HARMLESS, O.S., L.Y.). This study was also partially funded by Helmholtz AI through a HGF research grant (ALEGRA, grant no. ZT-I PF-5-121, O.S., L.Y.). Part of this work was funded by Helmholtz Imaging (F.I.), a platform of the Helmholtz Incubator on Information and Data Science. The authors are grateful to David Kutschke, Andreas Schroeppel, Anna Fuchs, Dr. Carola Voss, and Dr. Annette Feuchtinger for their technical support.

## Author contributions

L.Y. and O.S. conceived the project and designed the study. L.Y., Q.L., P.K., A.S., A.D., performed in vivo and ex vivo experiments. L.Y., Q.L., P.K., and A.F. contributed to annotation of the reference lungs and methodology. R.S., D.T., S.Z., and N.D. developed and optimized the AI training models and methodology. L.Y., O.S., Q.L., F.I., R.S., and S.Z. performed data analysis and figure preparation. O.S., T.S., M.R., G.B., F.I., M.P., D.R., H.B.S., and A.O.Y provided funding support and/or the resources. L.Y. and O.S. wrote the original draft and final manuscript, W.G.K., F.I., T.S., M.R., G.B., and M.E.D. contributed to the manuscript editing/writing, and all authors reviewed, discussed, and commented on the manuscript.

## Funding

## Competing interests

The authors declare no competing interests.

## Additional information

**Supplementary information** The online version contains Supplementary Material available at https://doi.org/10.1038/s41467-024-54267-1.

¹Institute of Lung Health and Immunity (LHI), Helmholtz Munich, Comprehensive Pneumology Center (CPC-M), Member of the German Center for Lung Research (DZL), Munich, Germany. ²Helmholtz AI, Helmholtz Munich, Munich, Germany. ³Helmholtz Imaging, German Cancer Research Center (DKFZ), Heidelberg, Germany. ⁴Division of Medical Image Computing, German Cancer Research Center (DKFZ), Heidelberg, Germany. ⁵Institute of Pharmacology and Toxicology and Institute for Biomedical Engineering, Faculty of Medicine, University of Zurich, Zurich, Switzerland. ⁶Institute for Biomedical Engineering, Department of Information Technology and Electrical Engineering, ETH Zurich, Zurich, Switzerland. ⁷Ludwig Maximilian University Munich, Munich, Germany. ⁸DKTK Munich, Munich, Germany. ⁹Research Unit for Precision Regenerative Medicine (PRM), Helmholtz Munich, Munich, Germany. ¹⁰Institute of Epidemiology (EPI), Helmholtz Munich, Munich, Germany. ¹¹Present address: Department of Thoracic Surgery, Shanghai General Hospital, Shanghai Jiao Tong University School of Medicine, Shanghai, China. ✉e-mail: lin.yang@helmholtz-munich.de; otmar.schmid@helmholtz-munich.de

