## [Transparent Peer Review file · Nature Communications]

LungVis 1.0: an automatic AI-powered 3D imaging ecosystem unveils spatial profiling of nanoparticle delivery and acinar migration of lung macrophages

Corresponding Author: Dr Otmar Schmid

Version 0:

Reviewer comments:

Reviewer #1

(Remarks to the Author)

Yang et al. present an automatic AI-powered analysis approach to understand nanoparticle distribution and macrophage migration in mouse lungs. The study is important as various drugs/vaccines are delivered via inhalation. Thus, understanding their deposition, tissue interaction, and impact on macrophages can impact decisions for accurate dosing. This is especially interesting, as macrophages are starting to be acknowledged as active participants in tissue development, changes, and healing.

The authors show beautiful imaging data, and I particularly enjoyed seeing an insight into the dataset challenges, which were very well explained. Additionally, the authors delivered a great showcase for how AI iteration numbers can impact the segmentation outcomes.

Code and training data available – and this is clearly stated at the end of the introduction

Another important point was that the authors took the time and thoroughness to compare PFA, PBS, and in vivo patterns. Oftentimes, fixation and ex vivo artefacts can be unknowingly introduced.

Major corrections

1. LungView is a great name, but unfortunately, it already exists as a lung cancer program management (<https://lungview.com/>). To avoid confusion, I would suggest changing the name.
2. The authors frame this study as “AI-driven 3D imaging technology LungView1.0”. Throughout the text, the focus pivots between LungView being an imaging technology (title) or image analysis (abstract) approach. Even after reading the introduction, I was not 100%; e.g. line 150 “AI-driven 3D imaging technology” and then line 151 “AI and deep learning pipelines, specifically convolutional neural networks (CNNs), with tissue-cleared light sheet fluorescence microscopy (LSFM) facilitates the automated, precise, and rapid segmentation of the entire lung bronchial tree”. After reading the manuscript it seems that “LungView1.0” is an adaption of nnU-Net with transfer learning and parameter adaption.
3. It is unclear which parameters were extracted from the data and how these were assessed. There seems to be a lot of data and information, but it is unclear how the information was derived, validated, and categorized as important.
4. The experimental “Material and Methods” section is very detailed, but very superficial description of code and parameters.
5. The use of red-green is not colourblind-friendly, and I would suggest changing the colours to magenta-green.

Minor corrections

- Line 219 “An unified terminology” to “A unified terminology”
- Figure 2a(3) typo “resloved” to “resolved”; Fig 2d illustrations are very helpful
- In the results section, result 1, the first third is very repetitive to the introduction.
- In Figure 2, the authors compare the airway and acinar signal. To achieve this, the authors coloured them both red to make them comparable, but I would make this greyscale to avoid confusion with the NP signal.
- Especially the introduction is written in a very long-winded way with very long sentences; often with up to three “and”.

Simplification would help the readers to follow the train of thought.

Question out of curiosity: The authors show the left lung lobe for data standardization. Do they observe the differences when

comparing left-right lobes? E.g. Fig S8d

Reviewer #2

(Remarks to the Author)

This manuscript establishes and tests a new AI powered 3-D technology (LungView) developed to investigate the distribution profiles of nanoparticles (NP) administered into the lungs. The manuscript describes how LungView was set up and optimised using training images. The programme builds on the nnU-Net AI approach (Isensee, F. et al. 2021) that enables biomedical image segmentation.

The manuscript shows the pattern of nanoparticle distribution in the lungs following 4 different administration methods: intranasal, intratracheal, ventilator-assisted aerosol and nose only aerosol. The patterns of NP distribution are carefully mapped and differences in distribution due to the different dosing methods are easily discerned in images and by quantification e.g. initially, intranasal dosing results in more nanoparticles in the bronchial region of lungs whereas fewer NPs were deposited in the bronchial regions following aerosol delivery. Interestingly the analysis shows that whatever the initial administration method, over time NP become primarily distributed throughout the acini and are much reduced in the airways.

The second half of the paper carefully analyzes and illustrates that the re-distribution of NPs in the lungs overtime occurs via macrophage mediated phagocytosis and active cellular re-distribution. The authors show tissue resident macrophages can take up NPs and move between different acini to redistribute them.

LungView is a step forward that identifies important information on the distribution of NPs over time, comparing different administration methods. The programme will be extremely useful as we move towards examining and optimising the delivery of medicines to treat lung diseases. This will be equally important across many fields including genomic and regenerative medicine approaches and toxicology. The authors have clearly described each part of their technology and the results that it produces, using high quality images, movies and associated schematic diagrams to facilitate the readers' understanding.

Minor comments:

- Line 130- typo -whole-lung change to whole-lung
- Discussion line 690- should say SARS-CoV-2

Reviewer #3

(Remarks to the Author)

Comment to authors

Manuscript Title:

LungView1.0: an automatic AI-powered 3D imaging technology unveils spatial profiling of nanoparticle delivery and acinar migration of lung macrophages

Summary:

This manuscript introduced LungView1.0, an AI-powered 3D imaging technology that integrates advanced AI algorithms and deep learning pipelines with tissue-cleared light sheet fluorescence microscopy (LSFM). LungView1.0 system automates and refines the analysis of NP deposition in whole-lung airways and alveoli of mice, offering quantitative and qualitative insights into multiple routes of pulmonary delivery. The authors conducted extensive work to analyze and verify the results, making the data reliable and the manuscript interesting to read. But there are some concerns regarding the paper which are mentioned below:

Major concerns:

1. In the introduction, the novelty or relevance of this work compared to multitudes of others that have been published are not clear. Please cite literature that has been published already in the field. It would be helpful to know where this work stands in comparison to published works.
2. In the manuscript, all quantifications are based on fluorescent dyes. However, it is crucial to ascertain the stability of these dyes and ensure that they have not detached from the nanoparticles.
3. (Fig2-h) From the quantitative perspective, the delivery efficiency of NOAI is the lowest. Why does j III have a higher NP fluorescence intensity compared to f III?
4. (Fig3-b,c) Why does the 24-hour NP fluorescence signal exhibit a greater intensity compared to that of the 2-hour signal? The authors should explain.

5. In the manuscript, the authors argue that “there is no inter-alveolar exchange of nanoparticles”. The rationale is based on “those acini with no/low initial NP deposition are not gaining more NPs over time in VAAD lungs”. The phenomenon is intriguing, but the signal intensity comparison should be quantified rather than visual observation.

6. Line 649-652 “NP accumulation was observed in the tracheobronchial (TB) lymph nodes isolated from blank and NP-exposed lungs at 24h and 14d post ITLI administrations (Figure 6g and Figure S14d)”. The authors mentioned, “this could at least partially be due to phagocytosis and translocation of TRMs to the TB lymph nodes as suggested by previous studies^{31,40}”.

Please provide relevant experimental verification.

Minor concerns:

1. Line 351-352: The author argues those acini with no/low initial NP deposition are not gaining more NPs over time in VAAD lungs (Figure 2f-i and Video S7_1-4). However, it seems the graph has been mislabeled. The error persists in line 355-356. The authors should check and correct it.

2. In Fig6-g, LSMF imaging of tracheobronchial (TB) lymph nodes at 24h and 14d after ITLI exposure. Scale bars: 1000 μm . In the manuscript, the scale of the lymph node images is equivalent to the panoramic view of the lungs. The accuracy of the scale bar is questionable.

3. The entire manuscript should also be re-examined by authors to check for grammatical errors

4. The manuscript contains 23 references to research endeavors conducted before 2014.

Suggest more recent references should be included in the manuscript. It is important to ensure that the information provided is accurate.

5. In the reference section, the references are not presented in Nature Communication style. Moreover, there are no page numbers in many references, including references 9,18,70.

The superscript position of the reference citation format is inconsistent.

Version 1:

Reviewer comments:

Reviewer #1

(Remarks to the Author)

The authors performed considerable work to improve the manuscript during revision.

The name has been changed from LungView to Lung Vis, and the manuscript has been improved to make it significantly clearer that it is an imaging ecosystem.

Sections have been added throughout the manuscript to clarify which parameters were extracted from the data and how these were assessed. In addition, Supplementary Fig. 6 has been added to clarify data. Particularly the “Materials and Method” section is significantly more clear.

Information has been added to the “Materials and Method” section to provide more detail on code and parameters.

The authors have done an excellent job to address all scientific questions, and included a large amount of information to provide depth across the biology and data science side of the manuscript. Additionally, major improvements have been made to readability (e.g. flow and shorter sentences) and accessibility (e.g. magenta-green rather than red-green images). In particular the improved “Introduction” helps to set the stage and the “Materials and Method” aids to supply depth and information.

Reviewer #3

(Remarks to the Author)

The author has addressed the issues I raised and I recommend acceptance.

Brief summary of the main improvements

We appreciate the reviewer's constructive evaluations and have highlighted the main improvements based on their comments. All aspects of this response letter, now integrated into the revised manuscript, have made it technically more robust and enhanced its potential impact.

1. We renamed LungView to LungVis and refined its description to emphasize its capabilities. The title was modified to "imaging ecosystem" to reflect LungVis's multidimensional nature.
2. We expanded on demonstrating the uniqueness and validation of LungVis 1.0-derived parameters, including AI-derived airway segmentation and regionally resolved nanoparticle/drug dose accuracy.
 - Comparative morphometric analysis confirmed LungVis's superior resolution over traditional lung casts, revealing additional generations in the airway tree.
 - We enhanced the explanation of AI pipelines and parameters, refining the AI code description and updating our GitHub repository with more details.
 - We provided detailed protocols for determining LungVis 1.0 derived parameters including bronchial and acinar NP doses, dose fractions, and NP agglomeration states, highlighting their enhanced descriptive power and clinical relevance.
 - We clarified the importance of LungVis-derived parameters by emphasizing their clinical relevance for region-specific lung disease treatment via precision inhalation therapy.
 - We verified the integrity of our melamine resin fluorescent nanoparticles in harsh chemical clearing conditions and up to 14 days in vivo through additional experiments and our previous studies.
3. We demonstrated tissue-resident macrophage migration to tracheobronchial lymph nodes through additional immunostaining experiments.
4. We clarified that optimal LSFM imaging is crucial for accurate NP spatial profiling in the lung (e.g., B/A and C/P dose ratios). LSFM images show optimal visualization of spatial NP distribution, but actual doses were measured via spectrofluorimetric analysis of lung homogenates.
5. We rewrote the "Introduction" to clarify the work's novelty and revised all figures and movies to avoid red-green color schemes.

P.S. All cited references here are listed in the order of their appearance in the main manuscript.

Point-by-point response to the reviewers' comments

Reviewer #1 (Remarks to the Author):

Yang et al. present an automatic AI-powered analysis approach to understand nanoparticle distribution and macrophage migration in mouse lungs. The study is important as various drugs/vaccines are delivered via inhalation. Thus, understanding their deposition, tissue interaction, and impact on macrophages can impact decisions for accurate dosing. This is especially interesting, as macrophages are starting to be acknowledged as active participants in

tissue development, changes, and healing.

The authors show beautiful imaging data, and I particularly enjoyed seeing an insight into the dataset challenges, which were very well explained. Additionally, the authors delivered a great showcase for how AI iteration numbers can impact the segmentation outcomes.

Code and training data available – and this is clearly stated at the end of the introduction

Another important point was that the authors took the time and thoroughness to compare PFA, PBS, and in vivo patterns. Oftentimes, fixation and ex vivo artefacts can be unknowingly introduced.

We appreciate the reviewer's positive and considerate assessment and recognition of the quality and impact of our work. All concerns raised have been addressed in a point-by-point manner as detailed below.

Major corrections

1. LungView is a great name, but unfortunately, it already exists as a lung cancer program management (<https://lungview.com/>). To avoid confusion, I would suggest changing the name. We thank the reviewer for highlighting this issue. After considering numerous alternative names, we have decided to use the term “LungVis” - for which no “competing” use was found on the internet.

2. The authors frame this study as “AI-driven 3D imaging technology LungView1.0”. Throughout the text, the focus pivots between LungView being an imaging technology (title) or image analysis (abstract) approach. Even after reading the introduction, I was not 100%; e.g. line 150 “AI-driven 3D imaging technology” and then line 151 “AI and deep learning pipelines, specifically convolutional neural networks (CNNs), with tissue-cleared light sheet fluorescence microscopy (LSFM) facilitates the automated, precise, and rapid segmentation of the entire lung bronchial tree”. After reading the manuscript it seems that “LungView1.0” is an adaption of nnU-Net with transfer learning and parameter adaptation.

We appreciate the reviewer's feedback and we have now clarified the term "LungVis" in the manuscript. In brief, LungVis is an innovative imaging ecosystem that integrates lung-optimized imaging technologies and AI-powered data processing tools for enhanced visualization and quantification capabilities. In particular, optimized tissue-clearing light sheet fluorescence microscopy (LSFM) is coupled with an active-learning, AI-driven image analysis tool, providing i) holistic 3D segmentations of various networks of the lung architecture (e.g., airway, vascular, and lymphatic system) from the raw, staining-free LSFM images, ii) an active learning AI training pipeline designed specifically for lung architecture, and iii) qualitative and (semi-)quantitative spatial profiling of lung-delivered substances within these 3D physiologic networks in murine lungs (from cellular to whole-lung scale). LungVis 1.0 thus includes raw and AI-enabled LSFM datasets of ca. 80 murine lungs, reference annotations, AI airway segmentations, and the AI model (modified nnU-Net source code), which is publicly available for further investigations performed by the relevant readers.

The AI part of LungVis 1.0 is specifically designed for the holistic labeling of airways from staining-free LSFM autofluorescence images (LungVis2.0 and Lung Vis3.0 will also include the vascular and lymphatic system, respectively).

In response to the last sentence of the reviewer's comment, we would like to clarify that the AI-part of LungVis 1.0 does not entail transfer learning. One of the key advances in

our AI-driven airway segmentation is an adaptation of nnU-Net framework with data-centric interactive annotations and parameter optimization, which achieves an efficient and precise labeling of entire 3D bronchial trees.

We revised the manuscript accordingly:

Abstract, lines 7-10:

Revised: Our study introduces LungVis 1.0, an artificial intelligence (AI)-powered 3D imaging ecosystem that integrates tissue-clearing light sheet fluorescence microscopy (LSFM) with deep learning-driven airway segmentation for holistic mapping and quantitative profiling of NP deposition in the bronchial tree and acini of murine lungs.

Results, lines 105-112:

Revised: To faithfully recapitulate NP distribution and transport in vivo, we developed LungVis 1.0, an AI-driven 3D imaging ecosystem for murine lungs. This integrated methodology includes several key components: optimized tissue-cleared light sheet fluorescence microscopy (LSFM) for data acquisition, AI and deep learning pipelines for whole-airway segmentation, as well as visualization and quantification of spatially resolved NP deposition (Fig. 1a). Using convolutional neural networks (CNNs), LungVis 1.0 enables automated, precise, and rapid segmentation of the entire bronchial tree from LSFM images, facilitating comprehensive visualization and quantification of NP deposition in airways and acini (Fig. 1a).

Method and Materials. Lines 878, 896, 942

Revised/Added three subtitles as the following:

- LungVis 1.0 data acquisition: Tissue clearing and light sheet fluorescence microscopy (LSFM)
- LungVis 1.0 AI pipelines: Active learning AI-driven airway segmentation
- LungVis 1.0 visualization and quantification: Spatial profiling of delivered NPs

3. It is unclear which parameters were extracted from the data and how these were assessed. There seems to be a lot of data and information, but it is unclear how the information was derived, validated, and categorized as important.

We thank the reviewer for giving us the opportunity for further clarification of the manuscript with respect to derivation, validation and significance of the information provided here.

Great care was taken to not only assure high data quality and adequate consideration of potential data bias, but also validate the obtained results. We have now further clarified these aspects and expanded the *Materials and Methods*, *Results*, and *Discussion* sections (for details, see below).

1. Derivation and validation of LungVis 1.0-powered segmentation of the entire 3D airway tree

From the LSFM images, accurate **segmentation of the entire 3D airway tree** of each lung (bronchial part of lung) is the prerequisite for obtaining the **bronchial and alveolar NP dose** in these lungs. The basis for these parameters is foremost high image quality, which mostly depends on the high quality of optical clearing of the lung

and secondly our newly developed AI-powered image processing tool for accurate 3D lung segmentation even in the presence of image artifacts (blurring, out-of-focus images, etc.) (see Fig. 1).

The accuracy of the AI-derived bronchial tree from LSFM images was confirmed by quantitative comparison of the obtained lung morphology with the ones obtained using alternative measurement methods (e.g., using lung casts) and by visual inspection of the terminal bronchioles - the most distal part of the airways (terminal bronchioles), which is most difficult to resolve due to low tissue autofluorescence and its small dimensions ($< 100 \mu\text{m}$). As described in our recently published paper (Estaji et al., JAS, 2024)²⁵, quantitative validation was accomplished by the excellent agreement of the quantitative lung morphology parameters of our LSFM lungs from C57BL/6 mice with those reported in the literature (lengths, diameters and branching angles of each of the up to ca. 25 lung airway generations (See the Below Fig. 7 from original publication). It is noteworthy that with LungVis 1.0 up to five more generations in the most distal part of the lung can be identified indicating the superior resolution of LSFM images as compared to previously used morphometric lung methods.

Fig. 7. The morphometric data of the left lung airways reported here are identical with those five C57BL/6 lung casts from Oldham et al. (Oldham et al., 2021, pp. 2050–2067) except for the airway length of the 2nd generation. A) Airway diameter, B) airway length, and C) branch angle. The error bars (only shown if larger than the size of the symbol) are representing the standard deviation about the mean value of our (left) lung. * $p < 0.05$ (t-test). This figure is retrieved from Estaji et al., JAS, 2024²⁵. (doi:10.1016/j.jaerosci.2024.106425)

By definition each terminal bronchiole leads directly into the alveolar duct, which can be visually validated from the (green) tissue autofluorescence and the (magenta) NP channel of the LSFM images. The completeness of the segmented airway tree was validated indirectly from the abrupt loss of tissue autofluorescence directly after the terminal bronchioles and directly by merging the airway mask obtained with LungVis 1.0 from green tissue autofluorescence images with the magenta NP channel slice-by-

slice (Supplementary Movie 1, Movie 5, Movies 6_1-6_4, Movies 7_1-7_4). Fig. 2d-2g_IV, Fig. 3a-3d_IV, and Fig. 3f-3i_IV show that the end of the terminal bronchioles are directly “connected” to a line-like hotspot NP deposition pattern, which is expected to appear there due to large sedimentation of the micron-sized droplets of NP suspension along the conducting alveolar region characterized by large aerosol residence time and small alveolar dimension. From the above, we conclude that our LSFM-derived airway trees are not only accurate with respect to their morphometric structure but also with respect to completeness of the airway tree.²⁵

2. Derivation and validation of quantitative regional NP dose profile

Once the 3D bronchial tree is segmented, the bronchial NP dose was obtained from the cumulative NP fluorescence intensity of the LSFM images in the bronchial region where the segmented bronchial tree served as mask (1: inside bronchial tree; 0: outside = alveolar) to select all NP intensity pixels in the bronchial region. The remaining NP signal was categorized as alveolar dose. The determination of bronchial and alveolar dose fraction was elaborately described in added Supplementary Fig 6a-6g.

The accuracy of the regionally resolved NP dose requires constant quantum yield from NPs. In a previous study we have shown that the total NP LSFM intensity data can be used to calculate the correct total NP lung dose independent of the route of NP application using fluorescence in lung homogenates as reference (see Fig. 5a and 5b in Yang ACS Nano,2019b)²⁶. Considering the substantial difference of the spatial NP distribution after intratracheal instillation (ITLI; patchy central delivery to upper part of lung) and aerosol VAAD delivery (uniformly distributed, reaching most distal parts of lung) as (seen in Fig. 2e and 2f of the present manuscript, this suggests that the **quantum yield from the NPs is relatively independent of their location in the lung**. It has been shown in our previous study that incomplete tissue clearing lowers the quantum yield from NPs located in this region (see Fig. 5a in Yang ACS Nano,2019b)²⁶. However, the regions of incomplete tissue clearing are typically relatively not very localized so that light attenuation affects both the airway tree and the acinar region in a magnitude. Consequently, incomplete tissue clearing is not expected to have a substantial impact on regional NP profiling with respect to bronchial to acinar and central to peripheral dose ratio (no outliers detected in Fig2j and 2k).

For biokinetics assessment one also needs to assure that the fluorescence signal of MF NPs is not degraded in lung tissue. In response to comment #2 of reviewer 3, we have conducted *in vitro* experiments with pulmonary macrophages indicating that there is neither any loss of NP signal nor leaching of dye from the NPs (see below).

Combined, this provides validation of the primary results of our LungVis 1.0 ecosystem, namely 3D structure of the entire bronchial airway tree and spatially resolved NP dosimetry derived from LSFM images. This enables us to visualize and quantify NP deposition and dose patterns in airways and acini (alveolar regions) at different time points (biokinetics) across multiple pulmonary delivery routes, as shown in Fig. 2j-2k and 3k-3l, Supplementary Fig. 6, as well as in Supplementary Movie 1, Movie 5, Movies 6_1-6_4 and Movies 7_1-S7_4.

3. Innovative parameters derived from LungVis 1.0 and their importance/clinical relevance

Building on the validated LSFM data we used simple arithmetics to derive several characteristic parameters of spatially resolved dosimetry. The rationale for considering them as important mainly relies on pathophysiologic and clinical relevance in the context of inhalation therapy for lung diseases.

We discriminate between two types of parameters, namely **conventional and innovative parameters** in the context of NP lung deposition/dosimetry and spatial NP lung profiling. The innovative parameters are designed to overcome some limitations of the conventional parameters leading to enhanced relevance.

3.1 NP lung deposition/dosimetry

3.1.1 Conventional: Total NP delivery efficiency (Fig. 2h, Fig. 3m-3n)

The NP delivery efficiency is calculated from the ratio of the total NP dose deposited in the lung (as described above) normalized to the invested NP dose. In principle, LungView 1.0 can be used for total NP dosimetry, if all LSFM images are obtained with the same operational settings (e.g., laser intensity, photon recording period). Due to the relatively large difference in deposited doses obtained with the different routes of NP application (here: 0.91 to 14.1 μg , see Supplementary Data 1), this was not possible in the present study. Also, to avoid inter-sample variation of NP signal, we therefore performed total dose measurements for each experimental condition with fluorescence analysis of lung homogenates as pointed out more clearly in revised Fig. 2a_ (3). In case of lavaged lungs, the total deposited lung dose can be derived from the total dose in the (lavaged) lung and trachea, as well as the NP dose in the bronchoalveolar lavage fluid (BAL) using standard curves of blank BAL (Yang et al. 2019a²⁰, 2019b²⁶). The delivery efficiency was determined for each route of pulmonary application. (Fig. 2h, Fig. 3m-3n)

Rationale/Importance: With the delivery efficiency one can calculate the total deposited drug/NP dose for each mouse from the (known) invested dose, which is essential for dose-response studies and dose-matched studies under varying drug and NP delivery conditions in the context of e.g., pharmacokinetic and pharmacodynamic studies in lung research.

3.1.2 Innovative: Bronchial (B) and acinar (A) NP deposition fraction (Supplementary Fig. 6h-6i)

The bronchial NP deposition fraction is calculated by dividing the bronchial NP intensity by the total NP intensity in both the bronchial and acinar regions (i.e. $B/(A+B) = 1/(1+A/B)$). Analogous, the acinar deposition fraction can be obtained from $1/(1+B/A)$. With the known bronchial or acinar NP/drug fractions, the accurate bronchial or acinar doses can be determined by multiplying these fractions by the initial delivery efficiency (Supplementary Data 1).

Rationale/Importance: Due to the preferential manifestation of many lung diseases in either the central (bronchial) or peripheral (distal, acinar) region of the lung, accurate knowledge of the bronchial and acinar drug/NP dose is important for targeted and efficient treatment of lung diseases. Having the capacity to measure bronchial and acinar deposition efficiency is the prerequisite for optimized aerosol delivery to the bronchial or alveolar lung

region.

3.2. Regionally resolved NP deposition (or retained dose) ratio

3.2.1 Conventional: Central-to-peripheral (C/P) deposition ratio (Fig. 2i)

Due to technical limitations, in clinical settings the information on the regional distribution of the inhaled drug/NP dose in the lung can only be obtained from the the *socalled* C/P dose ratio, i.e. the ratio of the dose deposited in the central and peripheral lung region defined as the inner 50% and outer 50% areas of 2D projection images of the 3D lung, respectively ^{22,26,44,45}

Rationale/Importance: The C/P deposition ratio is widely used in clinical and preclinical radiological studies,^{22,44,45} with C/P ratios close to unity indicating good peripheral deposition of aerosol in human or animal lungs. This is confirmed by this study, since only aerosol delivery provides C/P ratios close to unity, all other methods have C/P ratios > 2 (Fig. 2i).

3.2.2 Innovative: Bronchial-to-alveolar (B/A) deposition ratio (Fig. 2j, k)

Our concomitant analysis of C/P and B/A deposition ratios for 4 different pulmonary application routes with vastly different NP deposition profile revealed for the first time that the C/P ratio is reasonably well correlated with the B/A ratio ($R^2 = 0.66$) (Fig. 2k). As both the C and P region contain different fractions of B and A regions, perfect linear correlation between C/P and B/A cannot be expected.

Rationale/Importance: Hence, the clinically available C/P ratio is reasonably predictive of the more relevant B/A ratio. Albeit this analysis was done here for mice, the structural similarities of murine and human lungs suggest that this finding can be translated to clinical settings. Thus, C/P ratios can be roughly used to assess bronchial or alveolar targeting of pulmonary delivered drugs in clinical settings.

3.2.3 Innovative: NP agglomeration states in bronchial and acinar regions (Supplementary Fig. 6j-6l)

Rationale/Importance:

The size of single particles or particle agglomerates affects their cellular fate and response, since endocytic cellular uptake is size-dependent and thus size can affect toxicological/therapeutic responses.^{46,47} NPs tend to form large agglomerates when delivered as bulk liquid suspension (patchy deposition in the central lung region), while aerosolized NP delivery appears more uniformly dispersed (Fig. 2d-2g). However, the agglomeration state can also increase after NP deposition due to cellular uptake and subsequent sequestering in lysosomal compartments (no lysosomal degradation for biopersistent NPs as used here - see our response to comment #2 of reviewer 3 for more details). Thus, quantitative monitoring of the degree of NP agglomeration can provide new insights into both uniformity of NP deposition and NP uptake by cells.

Results: To determine the apparent NP agglomeration states upon bulk liquid or aerosol droplet delivery, we measured the apparent NP-positive volume

(from rendering of NP fluorescence in LSFM images) normalized to the respective mass dose in bronchial or acinar regions (from lung homogenate measurements). This former aspect is now more clearly indicated by inserting Supplementary Figure 6e und 6g - see below). Per definition, a higher NP-volume/dose value ($\mu\text{m}^3/\mu\text{g}$) indicates wider spreading of NPs in the lung. As expected, aerosol delivery via nose-only inhalation (NOAI) displayed significantly higher apparent NP-volume/dose values in both bronchial and acinar regions compared to bulk liquid-based deliveries (Supplementary Fig. 6j and 6k). This is qualitatively confirmed by the presence of larger NP clusters following bulk-liquid application as compared to the wider spreading of NPs (small-dot-like pattern) after aerosol delivery (Fig. 2d_II-2g_II). Interestingly, VAAD aerosol delivery does not show a different apparent agglomeration state than bulk liquid application. NOAI was expected to yield the lowest agglomeration state, since dried $0.7 \mu\text{m}$ aerosol experiences diffusive deposition, while $3 \mu\text{m}$ liquid aerosol (VAAD) leading to more focused NP deposition. On the other hand, the lack of reduced apparent agglomeration for VAAD as compared to bulk liquid application (INLA, ITLI) is at least partially due to an imaging artifact, since light scattering exaggerates the measured NP volume as compared to the geometric volume and thus obfuscates smaller differences in agglomeration state. Moreover, even the most wide-spread NP deposition profile will converge to a high NP-positive volume to NP dose ratio for high enough doses, since eventually the entire lung epithelium will be covered with NPs. Thus, especially the significant difference in VAAD and NOAI could be at least partially due to the more than 5-fold higher VAAD dose.

Measurement of agglomeration state: Using the LungVis 1.0 AI pipeline, not only the bronchial NP fluorescence intensity (NP dose in μg) but also the apparent volume of airway-deposited NPs can be obtained from the 3D reconstructed NP volume (μm^3) using quantitative tools in Imaris or ImageJ (see Supplementary Fig. 6e and 6g). Thus, the agglomeration parameter can be expressed in units of $\mu\text{m}^3/\mu\text{g}$ for both the bronchial and acinar regions (Supplementary Fig. 6f and 6g). It is important to note that due to light scattering effects the apparent NP volume is larger than the actual geometric volume, i.e. this method cannot be used to obtain the material density of the NPs.

4. Concluding final remarks:

LungVis 1.0 provides validated data on regionally resolved NP (or drug) dosimetry in the lung, it allows for validated visualization of NP deposition patterns in the bronchial and acinar regions and it also provides unprecedented quantitative information about regional NP dosimetry profiles, which was exemplified here for four types of pulmonary drug/NP delivery and the biokinetics of the NPs in the lung. Much of the importance of the innovative parameters originates from their enhanced descriptive power as compared to that of conventional parameters and their higher clinical relevance. Especially the unique determination of the bronchial and acinar drug/NP dose is crucial because lung diseases like asthma and squamous cell carcinomas primarily originate in the bronchi, while emphysema and adenocarcinomas occur in the acinar region.^{29,30} Harnessing these novel tools for customizing and optimizing

pulmonary delivery techniques will enable precise, disease-specific treatments in inhalation therapy.

It is important to note that the numerical values of all those characteristic parameters are provided in Supplementary Data 1.

We have made the following revisions to improve the clarity throughout the manuscript.

Added Supplementary Fig. 6a-6g

Supplementary Fig. 6: Holistic assessment of spatial NP deposition profiles is achieved through airway segmentation and NP intensity-based quantitative analysis. a-g: The methodology illustration for determining bronchial and acinar NP deposition fractions and NP agglomeration state (after INLA NP application). h-l: The corresponding

quantitative results. The raw LSFM autofluorescence image (a) was used to extract airway masks using LungVis 1.0 AI pipelines, enabling the reconstruction of the 3D airway structure(b). By overlaying the AI-driven airway masks with raw NP signals slice-by-slice over the entire lung LSFM stack (c), the delivered NPs in airways can be selectively visualized in 3D maximum intensity projection (MIP) and reconstructed images (d). Subsequently, the airway (bronchial) NP fluorescence intensity and apparent NP-positive volume can be determined in the 3D reconstructed NP-positive airway areas (e, in yellow) using Imaris or ImageJ. Similarly, NPs in acini (excluding airways) can be selectively visualized in 3D MIP and reconstructed 3D images (f), allowing for the determination of acinar NP fluorescence intensity and apparent NP-positive volume in 3D reconstructed NP-positive acinar regions (g, in magenta: NPs in acini). h and i: The relative bronchial or acinar NP dose fractions were determined by normalizing the bronchial or acinar NP intensity to the total NP intensity, respectively. The bronchial or acinar NP dose can then be determined by multiplication of the bronchial or alveolar dose fraction with the known deposited total dose from lung homogenization data. j and k: The bronchial or acinar NP agglomeration state was then calculated from the NP-positive volume in 3D reconstructed images normalized to the NP dose in the bronchial or acinar region. This parameter is proportional to NP spreading in the lung. k: The B/A ratio of NP agglomeration state. n=3-8 per route of application. Comparisons were performed with one-way ANOVA followed by Holm-Sidak multiple comparisons tests. Scale bars: 1000 μm .

Results lines 255-276

Revised: NPs often form large agglomerates when delivered as bulk liquid suspension or aerosol droplets, impacting their cellular fate via *e.g.*, size-dependent endocytic uptake pathways and hence toxicological/therapeutic responses.^{46,47} To quantify NP agglomeration states in bronchial and acinar regions the apparent NP-positive volume was normalized to the respective NP mass dose. A higher NP-volume/dose value ($\mu\text{m}^3\mu\text{g}^{-1}$) indicates wider spreading of NPs in the lung. As expected aerosol delivery via NOAI displayed significantly higher apparent NP-volume/dose values in both bronchial and acinar regions compared to bulk liquid-based deliveries (Supplementary Fig. 6j and 6k). This is qualitatively confirmed by the presence of larger NP clusters following bulk-liquid application as compared to the wider spreading of NPs (small-dot-like pattern) after aerosol delivery (Fig. 2d_II-2g_II). Interestingly, VAAD aerosol delivery does not show a different agglomeration state than bulk liquid application. NOAI was expected to yield the lowest agglomeration state, since dried 0.7 μm aerosol experiences at least partially polydirectional diffusive deposition, while 3 μm liquid aerosol (VAAD) does not, leading to more spatially focused NP deposition. On the other hand, the lack of improved spreading for VAAD as compared to bulk liquid applications could at least partially be an artifact of the exaggerated NP volume (as compared to the geometric volume) due to light scattering effects, which obfuscates smaller differences in agglomeration state. Thus, LungView1.0 can currently not be used to obtain correct values for the material density of NPs in the lung. The almost constant B/A ratio of apparent NP-volume/dose (0.26 - 0.38) indicates 3- to 4-fold higher NP packing density in the NP covered regions of the airways than in the acini independent of delivery route (Supplementary Fig. 6l). Collectively, LungVis 1.0 provides innovative (semi-)quantitative insights into pulmonary deposition profiles,

including bronchial and acinar NP doses and dose fractions, NP agglomeration states in airway and acinar regions.

Discussion, lines 688-700

Revised: LungVis 1.0 provides morphometrically accurate AI-driven segmented airway trees with unprecedented resolution and completeness. Comparative morphometric analysis with traditional lung casts revealed excellent agreement in length, diameter and branching angle for up to the 20 generations reported for lung casts (C57BL/6 mice) and LungVis 1.0 revealed 5 more previously unnoticed generations with diameters down to 90 μm .²⁵ Thus, lung casts are often not able to resolve the most intricate distal part of the airways. The completeness of the 3D LungVis 1.0 airway trees was confirmed by visual inspection of the point of transition from the bronchial into the alveolar region marking the terminal bronchioles. At the last generations of the LungVis 1.0 airway trees there is an abrupt loss of tissue autofluorescence accompanied by the expected sudden appearance of a line-like hotspot NP deposition pattern in VAAD lungs (oh post-application) originating from effective sedimentation of micron-sided drops along the centerline of the air-conducting proximal acinar region. Both features indicate the transition from the airway (bronchial) to the acinar region, thus confirming the completeness of LungVis 1.0 airway tree.

Discussion, lines 725-741

Revised: There is currently no clear understanding to what extent the NP agglomeration state affects therapeutic or toxicological responses. In part this has been due to the lack of a readily available measure of the agglomeration state in the lung. LungVis 1.0 introduces a semi-quantitative characteristic parameter for NP agglomeration state by measuring the apparent NP-volume/dose in the bronchial and acinar regions. In spite of the clearly different regional deposition profiles for bulk liquid and aerosol delivery on the coarse level (C/P or B/A), the apparent agglomeration state is consistent for all delivery routes except for NOAI delivering less agglomerated NPs to the lung. This may be due to the fact that NOAI was conducted with dried aerosol droplets (ca. 0.7 μm in diameter) as compared to all other routes of application using aerosol or spray droplets with 3 μm or more. On the other hand, the apparent agglomeration state depends on the applied dose and is biased by overestimating the geometric volume covered by NPs due to optical artifacts. The difference in agglomeration state for VAAD and NOAI delivery could be at least partially due to the more than 5-fold higher VAAD dose. Thus, LungVis 1.0 can only serve as a semi-quantitative measure of NP agglomeration. The constant B/A ratio of apparent NP-volume/dose below unity suggests that NP agglomeration is larger in the bronchial region, but this effect is independent of the route of application. This might be due to the fact that impaction focuses bronchial deposition to a narrow region near airway bifurcations, while gravitational setting and diffusion spreads the NPs over a wider area.

Materials and Methods. Lines 943-957 and 967-993

Revised:

LungVis 1.0 visualization and quantification: Spatial profiling of delivered NPs

Visualization of NP delivery features: Spatial profiling of NPs in the lung is

primarily based on 2D single slice/cross-sectional LSFM images which were processed into 3D maximum intensity projection (MIP) images using Fiji/ImageJ v1.53. 3D volume reconstruction (rendering) images and movies with manipulation were generated using Bitplane Imaris (versions 9.5.1 and 9.9.1). With the complete 3D lung airway structure generated from LungVis 1.0 AI pipelines, the NP dose in the airway (bronchial) and non-airway (acinar/alveolar) regions can be readily visualized and quantified. Briefly, by overlaying the AI-driven airway masks (binary images: 1 or 0 indicate pixels inside or outside the airway tree) with raw tissue autofluorescence or NP fluorescence images slice-by-slice over the entire lung LSFM stack, one can visualize both airways (green; tissue autofluorescence) and NPs in airways as 3D MIP images (see images of “Airways (AF)” and “Airway NPs” in Fig. 2, Fig. 3, and Supplementary Fig. 7). Similarly, the acinar deposited NPs can also be visualized outside of the bronchial tree as indicated in the 3D MIP images of “Acinar NPs” of Fig. 2, Fig. 3, and Supplementary Fig. 7. By merging the images of “Airway NPs” or “Acinar NPs” with corresponding regions from the AF channel (“Airways AF”), NP deposition in airway or acinar regions is displayed as shown in Fig. 2, Fig. 3, and Supplementary Fig. 7.

Quantification of NP delivery features: To quantify NP dose fractions in the airway and acinar regions of the lung, we measured the total NP fluorescent intensity using 3D MIP images of these regions (Supplementary Fig. 6a-6g). The NP signals were quantified in 3D using ImageJ and Bitplane Imaris, following the methodology outlined in our previous studies.^{26,74} Briefly, an intensity threshold was set to eliminate tissue AF from the NP-related intensity, and the bronchial and acinar NP signals were calculated in separated regions of “Airway NPs” and “Acinar NPs” using ImageJ/Bitplane Imaris (Supplementary Fig. 6d-6f). Typically, these threshold values are varying somewhat depending on the lung and/or operational parameters during image acquisition, but they are at least two-fold lower than the mean fluorescence intensity of all positive NP pixel intensities and 5 to 50 times lower than the highest NP pixel intensity. 3D volume reconstructions of AI-driven segmented airways, “Airway NPs”, and “Acinar NPs” were further built in Imaris to determine the total airway volume (μm^3), bronchial NP signal, and acinar NP signal, respectively. The bronchial and acinar NP dose fractions (equation 1) and their B/A dose ratio can then be computed from both software tools with consistent results (Fig. 2j and Supplementary Fig. 6h and 6i).

$$\text{Bronchial or acinar NP dose fraction} = \frac{\text{Bronchial or acinar NP fluorescence signal}}{(\text{Bronchial} + \text{Acinar NP fluorescence signals})} \quad (1)$$

Apparent NP-positive volumes in both regions (“Airway NPs” and “Acinar NPs”) were computed directly either in 3D using Imaris or in 2D slice-by-slice using Fiji/ImageJ or Anaconda (Python), with units in cubic micrometers (μm^3). Given the known total lung deposited dose (μg) from spectrofluorometric analysis in lung homogenates and the NP dose fractions in bronchial and acinar regions obtained from the LSFM images, the exact doses for bronchial and acinar regions are provided in Supplementary Data 1. From this, the apparent NP-volume/dose can then be determined in each region, which is a relative measure of the NP agglomeration state representing the inverse packing density after delivery (equation 2), since this agglomeration parameter is largest for complete spreading (*i.e.*, single NP spheres) and smallest, if all NPs are closely packed into one large agglomerate. The apparent NP-volume normalized to

bronchial or acini NP dose, as well as the corresponding B/A ratio is provided in Supplementary Fig. 6j-6l for each of the four application routes.

$$NP \text{ agglomeration state} = \frac{(B \text{ or } A) \text{ Apparent NP-positive volume } (\mu\text{m}^3)}{(B \text{ or } A) \text{ NP dose } (\mu\text{g})} \quad (2)$$

4. The experimental “Materials and Methods” section is very detailed, but very superficial description of code and parameters.

We thank the reviewer for pointing out the missing clarity in describing the used parameters.

Regarding the AI segmentation pipeline, we revised the explanation of how specific hyperparameters for nnU-Net training were chosen. In short, hyperparameters were not adapted in the initial training and were solely extracted by the nnU-Net fingerprinting. This process uses heuristics and rules that extract certain dataset characteristics such as e.g. the median shape of the images and based on that adapts parameters such as patch size. Therefore, the initial training with all the hyperparameters was completely out-of-the-box nnU-Net without modifications.

Only after the initial training some modifications were employed which were targeting specific problems. To ensure a more efficient training the image size was decreased by a factor of 2. Additionally, augmentations (e.g., rotations, Gaussian noise and blur, brightness and contrast transformations, gamma corrections, simulation of low resolution, and mirroring) were added that mimic the relevant imaging artifacts such as illumination issues, halos, blurring etc. These modifications were incorporated in nnU-Net and we highlighted the respective parts and how to use them in the code repository (<https://github.com/MIC-DKFZ/MurineAirwaySegmentation>).

Regarding the definition and relevance of the characteristic parameters calculated from our images, we have already updated the *Materials and Methods*, *Results*, and *Discussion* sections extensively in response to this reviewer’s comment 3 (see above for details).

Materials & Methods. Lines 899-917

Revised:

This is made possible by creating a dataset fingerprint which is then used to find a suitable hyperparameter configuration. The nnU-Net fingerprinting uses rules and heuristics based on the characteristics of the dataset such as the modality, intensity distribution, the median shape as well as the distribution of spacings to derive well working hyperparameters. For airway segmentation, the first training using the initial three labeled lungs was done out-of-the-box without additional adaptations (Fig. 1c). This means that the hyperparameters described in the following were completely determined by the nnU-Net fingerprint. Briefly, the nnU-Net uses a U-Net-like encoder-decoder architecture with skip-connections. Images are normalized using a z-score transformation with per image mean and standard deviation. The model is trained for 1,000 epochs where each epoch consists of 250 batches (each consisting of 2 patches). Due to the large image size, training is performed in a patch-based fashion, where smaller crops of size 48x224x224 voxels are sampled from the training images. One batch consists of two patches, where one patch is taken from a random location and the other is guaranteed to contain annotated airway pixels. The loss function is the

sum of the Dice and cross-entropy loss. During training the model is optimized using the SGD optimizer with an initial learning rate of 0.01, which is decayed using a poly learning rate schedule. Data augmentation (or parameter adaptation) contains rotations, Gaussian noise and blur, brightness and contrast transformations, gamma corrections, simulation of low resolution and mirroring. For inference a sliding-window approach is utilized with half-patch size overlap as well as a Gaussian patch center weighting. Moreover, mirroring along all axes is applied as a test time augmentation.

5. The use of red-green is not colour blind-friendly, and I would suggest changing the colours to magenta-green.

We acknowledge the importance of avoiding confusion for individuals with red-green color blindness. Therefore, we have switched all of our images and movies from red-green to magenta-green, red-cyan, and other colour schemes throughout the manuscript. These colour adjustments have been implemented in Fig. 2-6, Supplementary Figs. 5-14, Supplementary Fig. 16, and Supplementary Movie 1 and Movies 5-13.

Minor corrections

- Line 219 “An unified terminology” to “A unified terminology”

Done - see line 176 in the revised manuscript.

- Figure 2a(3) typo “resloved” to “resolved”; Fig 2d illustrations are very helpful

Done.

- In the results section, result 1, the first third is very repetitive to the introduction.

It has been revised as follows.

Lines 102-105

Revised: Despite the significant interest in *pulmonary drug delivery*, as evidenced by the ca. 1,500 annual publications (since 2019) on this issue in Web of Science Core Collection (Supplementary Fig. 1a), accurate spatial profiling of pulmonary delivered substances (here NPs) throughout the entire lung continues to be a major challenge.

- In Figure 2, the authors compare the airway and acinar signal. To achieve this, the authors coloured them both red to make them comparable, but I would make this greyscale to avoid confusion with the NP signal.

We are not sure what the reviewer is trying to indicate to us here. The images in Fig 2d - 2g depict the NP fluorescence signal of the LSFM images originating from either the airway (II) or the acinar region only (III) and are therefore colored red (now magenta), since they do show the “Airway NPs” and “Acinar NPs” signals (ex/em=640/690 nm). For example, the image showing Airway NPs was obtained by multiplication of our AI-enabled airway masks (binary image with airway 1 and other regions 0) with the magenta NP fluorescence channel.

We have tried to clarify this issue by modifying the caption of Fig. 2d - 2g as follows:

Line 289

d-g) Holistic mapping of NP distribution in the lung stratified for the airway (II) and

acinar region (III).

- Especially the introduction is written in a very long-winded way with very long sentences; often with up to three “and”. Simplification would help the readers to follow the train of thought.

We have largely rewritten the “Introduction” section to better focus on the current state of pulmonary drug delivery, nanoparticle inhalation, and lung macrophage immunity (including cellular mobility and macrophage function) based on previous studies. Additionally, we have shortened long sentences and improved conciseness and logical flow.

Lines 24-99

Details in the “Introduction” section, which has been largely rewritten.

Question out of curiosity: The authors show the left lung lobe for data standardization. Do they observe the differences when comparing left-right lobes? E.g. Fig S8d

The murine lung consists of five lobes in total: four lobes in the right side of the lung and one large lobe in the left side. According to literature and our own studies^{25,26}, the left lung accounts for approximately 35-40% of the total lung volume.

There is evidence from the literature that the morphological and biological structure of the (murine and human) lung is not fundamentally different on the right and left side of the lung. However, for pulmonary drug delivery, there are a few caveats to be considered. 1) Literature data indicates that spontaneous (nose-only) inhalation of aerosols (here: NOAI) delivers aerosol uniformly into each lobe with the fractional dose scaling according to the fractional volume of each lobe. This is to be expected, since the fractional volume determines the lobe-specific inhaled air volume and thus the lobe-specific delivered aerosol dose. 2) For intranasal bulk liquid aspiration (INLA) it is evident from our data that the upper part of the lung receives more dose than the lower parts of the lung - both on the left and the right side (see Fig. 2d). However, our data indicate that the left-right dose ratio roughly scales with the volume ratio of the left and right lung even for INLA. This is probably due to the fact that the mouse actively aspirates a spray (with large droplet diameter of ca. 40 μm or more - due to the low liquid dispersion energy obtained with natural breathing) and the amount of spray-laden droplets into the right and left half of the lung scales with the volume of the left and right half of the lung. 3) The situation is slightly more complicated for ITLI and VAAD, since here we apply the drug via an intubation cannula. The intubation depth of an intratracheal cannula is essential, since pushing the cannula beyond the end of the trachea will place it either in the left or right primary bronchus. Under those conditions, both ITLI and VAAD will deliver the NPs almost exclusively into only one side of the lung. We avoid this complication, by inserting the cannula to a consistent depth (2 cm from the upper teeth). This value was determined by visually inspecting the location of the tip of the intubation cannula for ca. 20 W57BL/6 mice of different sex and age by surgically opening of the fur along the trachea down to the 1st bifurcation (from the trachea into the left and right primary bronchus). Over the past >10 years of ITLI and VAAD NP/drug delivery, we have not seen any case with only one-sided substance delivery.

Having said this, we acknowledge that deviations from the volume-predicted, lobe-specific NP dose can sometimes occur. We have observed lobe-specific deviations of up to 40% (mean bias only 20%) for VAAD (up to ca. 10% for ITLI), but only in the right lung (Yang et al., 2019b²⁶ - see fig. 6c). In the left lung, the NP dose scaled very reproducibly with its volume (SD: <5% and <15% for VAAD and ITLA, respectively). This was one of the main reasons why we selected the left lung for data standardization.

We revised the following sentence:

Lines 315-317

For ease of visualization, the results of the longitudinal study are presented for the left half (left lobe) of the lung, which reliably represents both lung morphology and the NP dose profile of the entire lung.^{25,26}

Reviewer #1 (Remarks on code availability):

The code is available on GitHub with relevant explanation and readme files. Additionally, the authors shared datasets via zenodo. However, it would be useful to get a better understanding on the parameters and adjustments chosen by the authors.

We have updated our GitHub repository to include a more detailed explanation of the parameters employed in our training scripts. These enhancements aim to provide an understanding of the configurations and modifications applied during the model training process. For further technical specifics and implementation details, we invite the reviewers and readers to refer directly to the code files available in the repository.

Reviewer #2 (Remarks to the Author):

This manuscript establishes and tests a new AI powered 3-D technology (LungView) developed to investigate the distribution profiles of nanoparticles (NP) administered into the lungs. The manuscript describes how LungView was set up and optimised using training images. The programme builds on the nnU-Net AI approach (Isensee, F. et al. 2021) that enables biomedical image segmentation.

The manuscript shows the pattern of nanoparticle distribution in the lungs following 4 different administration methods: intranasal, intratracheal, ventilator-assisted aerosol and nose only aerosol. The patterns of NP distribution are carefully mapped and differences in distribution due to the different dosing methods are easily discerned in images and by quantification e.g. initially, intranasal dosing results in more nanoparticles in the bronchial region of lungs whereas fewer NPs were deposited in the bronchial regions following aerosol delivery. Interestingly the analysis shows that whatever the initial administration method, over time NP become primarily distributed throughout the acini and are much reduced in the airways.

The second half of the paper carefully analyzes and illustrates that the re-distribution of NPs in the lungs overtime occurs via macrophage mediated phagocytosis and active cellular re-distribution. The authors show tissue resident macrophages can take up NPs and move between different acini to redistribute them.

LungView is a step forward that identifies important information on the distribution of NPs overtime, comparing different administration methods. The programme will be extremely useful as we move towards examining and optimising the delivery of medicines to treat lung diseases. This will be equally important across many fields including genomic and regenerative medicine approaches and toxicology. The authors have clearly described each part of their technology and the results that it produces, using high quality images, movies and associated schematic diagrams to facilitate the readers' understanding.

We appreciate the reviewer's positive and considerate assessment and recognition of the quality and impact of our work. All concerns raised have been addressed in a point-by-point manner as detailed below.

We would like to make the reviewer aware that LungView1.0 has been renamed to LungVis 1.0 (to avoid conflicts with an already existing LungView technology).

Minor comments:

-Line 130- typo -whole-lung change to whole-lung

We thank the reviewer for pointing out this typo. It has been changed (see line 90 in the revised manuscript).

-Discussion line 690- should say SARS-CoV-2

Done (line 668).

Reviewer #2 (Remarks on code availability):

As above, I am not qualified to assess the code.

Reviewer #3 (Remarks to the Author):

Comment to authors

Manuscript Title:

LungView1.0: an automatic AI-powered 3D imaging technology unveils spatial profiling of nanoparticle delivery and acinar migration of lung macrophages

Summary:

This manuscript introduced LungView1.0, an AI-powered 3D imaging technology that integrates advanced AI algorithms and deep learning pipelines with tissue-cleared light sheet fluorescence microscopy (LSFM). LungView1.0 system automates and refines the analysis of NP deposition in whole-lung airways and alveoli of mice, offering quantitative and qualitative insights into multiple routes of pulmonary delivery. The authors conducted extensive work to analyze and verify the results, making the data reliable and the manuscript interesting to read. But there are some concerns regarding the paper which are mentioned below:

We appreciate the reviewer's positive and considerate assessment and recognition of the quality and impact of our work. All concerns raised have been addressed in a point-by-point manner as detailed below.

We would like to make the reviewer aware that LungView1.0 has been renamed to LungVis 1.0 (to avoid conflicts with an already existing LungView technology).

Major concerns:

1. In the introduction, the novelty or relevance of this work compared to multitudes of others that have been published are not clear. Please cite literature that has been published already in the field. It would be helpful to know where this work stands in comparison to published works. We thank the reviewer for pointing out this important issue. We have revised the

“Introduction” section to better focus on the current state of pulmonary drug delivery, nanoparticle inhalation, and lung macrophage immunity (including cellular mobility and macrophage function) based on previous studies. We have emphasized the current limitations in optical imaging of whole organs, such as the lung, and highlighted the novelty of this study in terms of imaging methods and computational tools. Additionally, we have shortened long sentences and improved conciseness and logical flow (as requested by one of the other reviewers). As a result of these requests, we have rewritten most of the introduction.

The key information addressing the novelty of the present study compared to previous studies is now found in the manuscript in Lines 44-61:

Experimental validation of their local dosage and spatial distribution in murine lungs often relies on either non-optical or optical imaging techniques. Non-optical imaging techniques including X-ray imaging, planar γ -scintigraphy, positron emission tomography (PET) or single photon emission computed tomography (SPECT), generally provide quantitative deposition data but do not offer microscopic visualization of delivered substances.¹⁹⁻²² Optical measurements are predominantly conducted using low-resolution whole-organ or whole-body fluorescence imaging or 2D microscopy,^{1,7-9,17,18,23} leading to an incomplete understanding of cellular localization and molecular interactions within 3D tissue niches. Serial lung block-face imaging of cryomicrotome slices has been employed to generate 3D lung airway meshes and visualize local particle deposition.²⁴ However, it involves time-intensive data acquisition, intricate pre- and post-imaging processes, and laborious manual corrections.²⁴ Computation mathematical models are also applied to project site-specific aerosol deposition in 3D lung airway models.²⁵ Our prior work has combined X-ray phase contrast imaging with tissue-cleared light sheet fluorescence microscopy (LSFM) for real-time in vivo monitoring and 3D cellular-resolution visualization of NP distribution in entire murine lungs.^{20,26} This study further advances the visualization and quantification capability of spatial NP profiling in the lung through development of LungVis 1.0, a technological ecosystem integrating tissue-cleared LSFM with artificial intelligence (AI) and deep learning-driven imaging analysis (convolutional neural networks, CNNs)²⁷ in a precise and automatic manner.

2. In the manuscript, all quantifications are based on fluorescent dyes. However, it is crucial to ascertain the stability of these dyes and ensure that they have not detached from the nanoparticles.

We thank the reviewer for highlighting this important issue. We agree with the reviewer that stability of the fluorescent dyes is critical for NP dosimetry under the chemically harsh conditions of the tissue clearing process and biodegradation in the lung for up to 14 days. The former and the latter requires the fluorescent NPs to withstand treatment with organic solvents such as tetrahydrofuran (THF), dichloromethane (DCM), and dibenzyl ether (DBE) and the latter low pH values and harsh metabolic degradation in the lysosomal compartment of the cells. For the study presented here we selected high size-monodisperse (CV<5%), spherical NPs with high thermal and chemical stability, and strong fluorescence signal relative to tissue autofluorescence.

Since 2015, we have screened various NP types and selected melamine resin

fluorescent particles (MF, Microparticles Inc.) as our NP model for inhalation studies (Yang et al. 2019a²⁰, Yang et al. 2019b²⁶). According to manufacturer's information (https://www.microparticles-shop.de/Fluorescent-Particles/Melamine-resin-fluorescent-particles/Melamine-resin-particles-Red-fluorescent-MF-FluoRed-Ex-Em-636-nm-686-nm:::7_42_20.html), these MF particles are stable **in common organic solvents and oils, without dye leaching, swelling, or shrinking**. We have gathered several pieces of experimental evidence demonstrating the exceptional stability of MF particles under various chemical, mechanical, and biological treatments.

First, in one of our previous studies we observed that MF particles remained stable and exhibited no significant bleaching or leaching even after 7 days of optical clearing treatment with organic solvents (THF, DCM, and BABB/DBE). This is depicted in the following figure, which was taken from Yang et al. 2019b²⁶.

Figure legend: Analysis of MF NP stability in murine lungs at various stages during the tissue clearing process using an In Vivo Imaging System (IVIS). Fluorescence intensity after each step of the clearing process (c1 - c8) was normalized to the intensity after DCM treatment, starting post-lung perfusion, through water and lipid removal (panel a), to refractive index matching with BABB or DBE (panel b). (c) Representative *ex vivo* lung images (c1–c8) from mice receiving 25 μ g of MF via ITLI, measured by IVIS, show that MF fluorescence remains stable within statistical uncertainty during the tissue clearing process and is preserved for up to 7 days, despite some lung shrinkage (1.27-fold, 1D). These images and data are retrieved from our previous publication Yang et al. 2019b²⁶.

Second, using LSFM, we observed single-dot-like distribution patterns of NPs in the lungs at 14 days post ITLI or VAAD delivery (Fig. 3d and 3i). Additionally, we visualized MF particle signals with cellular resolution using confocal laser microscopy (Fig. 4b and 4c, Supplementary Fig. 12c and 12f). These results indicate that the MF NPs remain stable enough for fluorescence imaging after at least 14 days in the lung, despite a significant reduction in the total NP dose due to NP clearance from the lung due to e.g. the mucociliary escalator. The issue of absolute fluorescence stability of NP in the lung is explored further with in vitro experiments as described in the next paragraph.

Lastly, considering that MF particles are predominantly ingested by lung macrophages where they experience a harsh biological environment in the phagolysosome (e.g. pH

= 4.5), we incubated MF particles with murine alveolar macrophages (MH-S cells) cultured *in vitro* for up to 4 days - the maximum duration for culturing the cells without medium change. We measured the fluorescence intensities in the supernatant and cell pellets at 0h, 1d, and 4d to detect any loss of MF signal in the cells (pellet) or bleaching of MF signal into the supernatant. The MF fluorescence intensities in cell pellets were normalized to the fluorescence intensities of the nominally expected NP concentration in culture medium (added Supplementary Fig. 4a - see below) These values remained stable near unity and showed no significant difference among various time points. Moreover, the ratio of MF fluorescence in cell supernatant was constant below 3% (Supplementary Fig. 4b) of the total NP signal from cell pellets and supernatant for all time points. Taken together, this indicates that the fluorescence signal associated with MF particles was resilient to cellular uptake into phagolysosomes resulting neither in bleaching nor leaching of the fluorescence signal.

Moreover, we have demonstrated in one of our previous studies, that the mechanical tissue homogenization process, which was conducted in PBS, did not affect the MF fluorescence intensity (Fig. 5 in Yang et al. 2019b²⁶). Hence, fluorescence-based NP dosimetry in lung homogenates can be used as a reference method for determination of the total NP dose in murine lungs (see Fig. 2h in present manuscript).

Collectively, these results underscore the suitability of MF fluorescent particles for this study, providing robust and reliable data for pulmonary drug delivery research.

The following changes were made to the manuscript:

Results, Lines 191-199

Melamine resin fluorescent (MF) NPs exhibit uniform particle size (monodisperse) and spherical morphology (Fig. 2b). As reported in our previous studies,^{20,26} MF NPs demonstrate exceptional fluorescence stability under various chemical (organic solvents during tissue clearing) and mechanical treatments (tissue homogenization). Furthermore, *in vitro* incubation of MF NPs with MH-S cells (macrophages) for up to four days shows that the fluorescence signal remains resilient to phagolysosomal degradation by macrophages, exhibiting neither bleaching nor leaching (Supplementary Fig. 4). These results underscore the optical stability of MF NPs for longitudinal co-imaging of NP distributions and lung architecture for biokinetics studies from macroscale to cellular resolution in tissue-cleared optically transparent lungs (Fig. 2c).

Added Supplementary Fig. 4

Supplementary Fig. 4: Quantitative measurement of fluorescence stability of melamine

resin (MF) particles in MH-S cells (murine macrophage cell line) for up to four days. MF particles at the concentration of 10 $\mu\text{g}/\text{mL}$ were incubated with MH-S cells in 500 μl culture medium. At 0h, 24h and 4d the culture medium and cells were collected and centrifuged at 10,000 rpm to separate and detect the NP signals in cell pellets and supernatant. a: MF NP fluorescence intensity in cell pellets normalized to the fluorescence intensities of the nominally expected NP concentration in cell culture medium at each time point. b: Ratio of NP intensity in the supernatant to the total NP intensity from cell pellets and supernatant. n=2. Statistical comparisons were performed with one-way ANOVA followed by Holm-Sidak multiple comparisons tests.

Experimental details regarding “Fluorescence stability measurement of melamine resin particles in cell culture” were added to the *Supplementary file_Materials and Methods* section.

3. (Fig2-h) From the quantitative perspective, the delivery efficiency of NOAI is the lowest. Why does j III have a higher NP fluorescence intensity compared to f III?

We assume that the question of the reviewer refers to Fig 2g III (not 2j III) and 2f III depicting the spatial NP distribution directly after NOAI and VAAD application, respectively. Ideally, the pulmonary NP fluorescence intensity is directly proportional to the pulmonary NP dose not to the delivery efficiency. However, total NP signals in the LSFM lung images can be influenced by other factors, such tissue transparency, residual blood, lung volume, and more. To achieve best imaging quality for ca. 80 murine lungs in this manuscript, LSFM instrumental settings needed to be adjusted at least for each delivery method to account for the up to 15-fold different deposited NP dose (ITLA: $14.9 \pm 3.7 \mu\text{g}$, INLA: 9.7 ± 5.76 , VAAD: 4.5 ± 1.6 NOAI: $0.91 \pm 0.12 \mu\text{g}$). For the lowest NP dose (NOAI: $0.91 \mu\text{g}/\text{mouse}$ the highest fluorescence yield (fluorescence intensity per NP dose) was selected. Thus, the NP fluorescence intensity in the LSFM images is not necessarily indicative of the pulmonary NP dose.

It is important to note that optimal LSFM imaging is essential for accurate spatial profiling of the NPs in the lung (e.g. C/P and B/A dose ratio). As this was the primary objective of the present imaging study, the “optimal” LSFM instrumental settings were chosen for each lung. The total lung deposited NP dose was obtained from spectrofluorimetric analysis of lung homogenates. This information is now more clearly provided in Fig 2a_(3) as shown below.

Fig. 2a: The schematic illustration of the experimental and analytical approach. In section (3), the total NP dose in different lung compartments was qualified in lung/trachea homogenates as well as in BAL cells (3.1) and AI-driven NP spatial deposition analysis was done with LSFM

images (3.2).

Fig. 2 shows LSFM images of four lungs representing the four different routes of application. To avoid confusion (as noted by the reviewer) we have now re-scaled the NP intensity in each of these images such that the dose-ranking of the pulmonary delivery routes (ITLA: $14.1 \pm 3.7 \mu\text{g}$, INLA: 9.7 ± 5.76 , VAAD: 4.5 ± 1.6 NOAI: $0.91 \pm 0.12 \mu\text{g}$) is represented as trend in the total NP intensity of the LSFM images. Thus, there is no extremely obvious mismatch anymore between NP dose and NP intensity in the LSFM images. It is important to note that lung segmentation and determination of C/P, B/A or any other spatial dose ratios were performed with the “optimal” LSFM images.

In summary, raw LSFM images were primarily optimized to visualize and quantify relative differences in the spatial NP distribution within each lung for four delivery routes and might not accurately reflect the delivered NP dose. However, to prevent misinterpretation, we have 1) clarified that absolute dosimetry is obtained from lung homogenates and spatially resolved NP distribution is obtained from LSFM images (see Fig. 2a_(3) and 2) we adjusted the intensity scales in Fig. 2d-2g and Supplementary Fig. 3a-3b such that the NP dose- and intensity-ranking is consistent.

In the manuscript, we have modified Fig. 2d-2g as described above and we have made the following changes:

Materials and Methods, Lines 891-895

Revised: To achieve the best imaging quality for the approximately 80 murine lungs presented here, LSFM instrumental settings were adjusted for each delivery method and longitudinal measurement to account for the up to 15-fold variation in deposited NP dose. It is also important to note that optimal and individualized LSFM imaging is essential for accurate spatial profiling of NPs in the lung, such as the C/P and B/A dose ratios.

Subsection of “Visualization of NP delivery features”, Lines 958-966.

Revised: As mentioned above, optimal LSFM imaging with appropriate instrument settings is essential for each lung to profile the spatial patterns of NP deposition qualitatively and quantitatively (*e.g.*, C/P and B/A dose ratios) after four delivery routes and at various time points. It is important to note that NP total intensity in LSFM images (Fig. 2, Fig.3, Supplementary Fig. 3a-3b and Fig. 7) does not necessarily reflect the deposited NP dose, which is accurately determined from spectrofluorimetric analysis of lung homogenates (described below). This information is shown in Fig. 2a_(3). To prevent misinterpretation of NP dose in LSFM images, the NP intensity was therefore scaled to relatively match the lung homogenate-based dosage ranking in the whole lung or left lung images shown in Fig. 2d-2g, Fig. 3a-3d, Fig. 3f-3i, and Supplementary Fig. 3a-3b and Fig. 7.

Fig. 2d-g caption, Line 299-300

Added: NP intensity optimized for spatial visualization, was scaled to match lung homogenate dose trends but may not reflect the actual dose.

4. (Fig3-b,c) Why does the 24-hour NP fluorescence signal exhibit a greater intensity compared

to that of the 2-hour signal? The authors should explain.

We agree with the reviewer that the fluorescence signal (retained NP dose) should decrease with time due to lung clearance mechanisms. NP clearance from the lung is biphasic with rapid mucociliary clearance from the bronchial region and a much slower macrophage clearance rate from the acinar region. For murine lungs, the fast bronchial clearance ends between 2h and 24h post-delivery - often just determined by the first measured data point in the respective studies (e.g., Rijt, et al. 2016)⁷⁵. Subsequently, “slow” clearance from the acinar region occurs due to migration of NP-laden macrophages from the acinar into the distal bronchial region from where they are cleared by the mucociliary escalator from the lung. According to Rijt et al., 2016⁷⁵ one would expect ca. 15% clearance occurring within 24h and then another 50% (absolute) clearance at 14d yielding a retained NP fraction of 35% (of the delivered dose).

We agree with the reviewer that in the original version of Figure 3 the total NP fluorescence in the LSFM images depicted in Fig. 3b and 3c indicate that the NP intensity is lower at 2h than at 24h. In this case this is not due to large differences in instrumental settings during LSFM image acquisition (they were identical), but it is a result of measurement uncertainty in the LSFM images. This can be seen from the lung homogenate analysis in Fig. 3m which reveals a constant NP dose within measurement uncertainties during the first 24h after ITLI. Considering the combined measurement uncertainties in lung homogenate, trachea homogenate and BAL, the expected reduction of NP dose by 15% cannot be resolved within the experimental uncertainties lung/trachea homogenate analysis. Yet, at 14d the measured retained dose of 27% is in agreement with the 35% reported by Rijt et al., 2016⁷⁵ (within the combined experimental uncertainties from the present study and from Rijt et al., 2016⁷⁵). To prevent misinterpretations, we have again adjusted the intensity scale according to the delivered dose measured in lung homogenates (Fig. 3a-d and 3f-3i, and Supplementary Fig. 7).

See revisions for the last comment and the following revisions in Fig. 3 caption.

Fig. 3a-d and 3f-3i legend, Line 433-434

Added: NP intensity optimized for spatial visualization, was scaled to match lung homogenate dose trends but may not reflect the actual dose.

5. In the manuscript, the authors argue that “there is no inter-alveolar exchange of nanoparticles”. The rationale is based on “those acini with no/low initial NP deposition are not gaining more NPs over time in VAAD lungs”. The phenomenon is intriguing, but the signal intensity comparison should be quantified rather than visual observation.

On a qualitative level (visual examinations) we observed for ITLI 24h lungs (Fig. 3C_II), that NPs remained concentrated in the central acini, with few NPs in the peripheral acini, indicating no NP movement between these regions (or between the central and peripheral acini). This phenomenon was also observed at 14d where the NP signal is much weaker in the lower part of the lung (Fig. 3D_II). The same feature was also found after aerosol application in a VAAD 24h lung (where the NP signal is much weaker in the upper left part of the lung, Fig. 3H_II).

Inter-acinar exchange of NPs is expected to be most reliably demonstrated and

quantified in those regions of the lung, where acini with extremely high NP dose are in close proximity to acini with extremely low NP dose. This is due to the fact that the clearance rate from high-dose acini is high and potential transport of those NPs into an adjacent low-dose acinus is most likely and most feasible detectable via imaging. To confirm it, we quantified the NP fluorescence signals in equivalently placed and sized 2D regions of adjacent acini with high NP dose (NP+) and no/low NP dose (NP-) for both VAAD and ITLI lungs at 24 hours and 14 days post-delivery (see added Supplementary Fig. 8 - attached below). The regions of interest were placed at the entrance of the acini along the centerline of the alveolar duct directly connected to the terminal bronchioles, since there the NP dose is highest. Histogram profiling of pixel intensity shows a substantial NP signal difference between NP+ and NP- acinar regions (Supplementary Fig. 8_II), with the latter receiving only about 2-5% of the particles (Supplementary Fig. 8_III, i.e. 20-50-fold dose enrichment in the NP+ acinus) for both 24 h and 14 d time points. This enormous disparities in NP signals between adjacent acini especially in the 14 d time point suggests that no active inter-acinar exchange of NPs occurs, i.e. no mitigation of these large disparities occurs over time.

We acknowledge that it is difficult to perform this analysis quantitatively and statistically robust, since one would have to assess the NP dose in the 3D acinar region and perform a statistical analysis for equivalently loaded low/high NP acinus pairs. This is technically difficult/impossible due to the lack of automatic segmentation of individual acini. However, we offer Supplementary Fig. 8 as circumstantial quantitative evidence supporting our claim of no inter-acinar NP transport.

Additionally, it is well-known that particles located in the airway epithelium can be removed from the lung via the mucociliary escalator. An effective lung mucociliary escalator will transport mucus, with any captured foreign particles, in an upstream direction towards the upper airway away from the alveoli. Specifically, this mechanism clears particles from the terminal bronchioles (the smallest and last airways connecting to the acinar/alveolar regions) to the larger bronchi, trachea, and larynx. However, to the best of our knowledge, there is no literature indicating that acinar-located NPs, once returned to the terminal bronchiole, can translocate to other acinar regions. Our image observations and quantifications further support this common understanding of lung biology.

The manuscript was changed accordingly:

Results, Lines 327-331 and 351-354.

Revised: Observations at later stages (2h, 24h, 14d) indicate NPs continue to concentrate in the central acini without apparent spreading throughout the lung (Fig. 3b-d_(II and III), Supplementary Fig. 7a-7d and Supplementary Movies 6_2-4), suggesting the absence of transport from central/upper to peripheral/lower acini. The substantial NP signal difference in adjacent acini at both 24h and 14d post-ITLI delivery (ca. 40-50-fold dose ratio, Supplementary Fig. 8a-8b) further suggests an absence of inter-acinar NP exchange, as such exchange would likely alleviate these large NP dose differences between closely situated acini.

Revised: Also, histogram profiling of pixel intensity shows a substantial NP signal difference between high and no/low NP acinar regions, with the latter receiving only

about 4-5% of the particles at both the 24h and 14d time points (Supplementary Fig. 8c-8d). This indicates no active NP exchange among acini (inter-acinarly).

Added Supplementary Fig. 8

Supplementary Fig. 8: Quantitative analysis of NP fluorescence signal in adjacent acini in the ITLI and VAAD lungs at 24h and 14d post-delivery. Representative acinar regions were selected to demonstrate the substantial difference of NP signals in adjacent acini (20-50-fold difference). Regions of interest were placed at the entrance of the acini along the centerline of the alveolar duct directly connected to the terminal bronchioles, where the NP dose is highest. The yellow ovals or rectangles (I) indicate the selected acinar regions where the NP fluorescence signals in adjacent high NP-dose regions (NP+) and no/low NP-dose regions (NP-) were quantified. The data is quantitatively presented using histogram profiles (II) and pie charts (III). a.u. indicate arbitrary units.

Experimental details regarding “Quantitative analysis of NP signals in adjacent acini” were added to the *Supplementary file_Materials and Methods* section.

6. Line 649-652 “NP accumulation was observed in the tracheobronchial (TB) lymph nodes isolated from blank and NP-exposed lungs at 24h and 14d post ITLI administrations (Fig. 6g and Figure S14d)”. The authors mentioned, “this could at least partially be due to phagocytosis and translocation of TRMs to the TB lymph nodes as suggested by previous studies^{31,40}”. Please provide relevant experimental verification.

To confirm the presence of NP-laden tissue-resident macrophages (TRMs) in tracheobronchial (TB) lymph nodes, we performed immunofluorescence staining with several TRM markers on formalin-fixed paraffin-embedded sections of TB lymph nodes from VAAD mice. We used antibodies targeting the common immune cell marker CD45, the TRM marker CD68, the alveolar macrophage marker CD11c, and the interstitial macrophage marker CD11b (all antibodies were from rabbit hosts). Our observations revealed NP⁺CD45⁺, NP⁺CD68⁺, NP⁺Cd11b⁺, and NP⁺Cd11c⁺ macrophages in the TB lymph nodes of VAAD mice at both 24 hours and 14 days post-VAAD delivery (added Fig. 6h). These findings suggest that NP accumulation in the lymph nodes can at least partially be attributed to phagocytosis and subsequent translocation activities of TRMs.

The manuscript was changed accordingly:

Results, Lines 659-663

Revised: As expected, NP accumulation was observed in the tracheobronchial (TB) lymph nodes isolated from 24h and 14d ITLI mice (Fig. 6g and Supplementary Fig. 16d). This could be partially attributed to phagocytosis and translocation of TRMs to the TB lymph nodes,^{32,42} as evidenced with NP⁺CD45⁺, NP⁺CD68⁺, NP⁺Cd11b⁺, and NP⁺Cd11c⁺ macrophages in the TB lymph nodes of 24h and 14d VAAD mice (Fig. 6h).

Added Fig. 6h

Fig. 6h: Immunostaining of 2D sections of TB lymph nodes with several TRM markers at 24h and 14 d post-VAAD exposure, as well as in untreated controls. Arrowheads indicate the NP-laden macrophages in TB lymph nodes. Scale bars:10 μ m. n=3.

Experimental details regarding “Immunofluorescence staining and microscopy of tracheobronchial lymph nodes” were added to the *Materials and Methods* section.

Minor concerns:

1. Line 351-352: The author argues those acini with no/low initial NP deposition are not gaining more NPs over time in VAAD lungs (Fig. 2f-i and Video S7_1-4). However, it seems the graph has been mislabeled. The error persists in line 355-356. The authors should check and correct it.

Yes, indeed, we got confused with the numbering scheme. We thank the reviewer for pointing out these errors.

- it has been changed to “Fig. 3f-i and Supplementary Movie 7_1-4” in line 351 in the revised manuscript.
- also changed to “Fig. 3f_(III-IV) and Supplementary Fig. 7e” in line 357 in the revised manuscript.

2. In Fig6-g, LSMF imaging of tracheobronchial (TB) lymph nodes at 24h and 14d after ITLI

exposure. Scale bars: 1000 μm . In the manuscript, the scale of the lymph node images is equivalent to the panoramic view of the lungs. The accuracy of the scale bar is questionable. The scale bars in Fig. 6g and Supplementary Fig. 16d are 100 μm . It has been corrected in the revised manuscript.

3. The entire manuscript should also be re-examined by authors to check for grammatical errors. We have re-examined the manuscript thoroughly and corrected any grammatical errors and typos.

4. The manuscript contains 23 references to research endeavors conducted before 2014. Suggest more recent references should be included in the manuscript. It is important to ensure that the information provided is accurate.

Research on lung morphology, pulmonary application of substances and macrophage-nanoparticle interaction has a long-standing history. In light of this, we argue that 23 out of 82 references does not seem unjustified. However, we have replaced 7 references (refs. 33, 36, 41, 44, 46, 58, 61, and 65 in the previous manuscript) with more recent and relevant ones (refs. 25, 36, 37, 43, 46, 47, 61, and 74 in the revised manuscript).

5. In the reference section, the references are not presented in Nature Communication style. Moreover, there are no page numbers in many references, including references 9,18,70.

The superscript position of the reference citation format is inconsistent.

We have updated our reference list to comply with the Nature Communications style, including page numbers for all cited papers. Additionally, we have standardized the superscript position of cited references throughout the manuscript.

Point-by-point response to the reviewers' comments

Reviewer #1 (Remarks to the Author):

Reviewer #1 (Remarks to the Author):

The authors performed considerable work to improve the manuscript during revision. The name has been changed from LungView to Lung Vis, and the manuscript has been improved to make it significantly clearer that it is an imaging ecosystem. Sections have been added throughout the manuscript to clarify which parameters were extracted from the data and how these were assessed. In addition, Supplementary Fig. 6 has been added to clarify data. Particularly the "Materials and Method" section is significantly more clear. Information has been added to the "Materials and Method" section to provide more detail on code and parameters. The authors have done an excellent job to address all scientific questions, and included a large amount of information to provide depth across the biology and data science side of the manuscript. Additionally, major improvements have been made to readability (e.g. flow and shorter sentences) and accessibility (e.g. magenta-green rather than red-green images). In particular the improved "Introduction" helps to set the stage and the "Materials and Method" aids to supply depth and information.

We are pleased that the substantial yet essential revisions, including renaming to LungVis, additional figures, and updates to the Methods, have enhanced clarity and accessibility. We appreciate your recognition of the improvements in scientific depth and readability throughout the manuscript.

Reviewer #1 (Remarks on code availability):

I did not run the code, but reviewed the repository and some of the code. The authors added a clear "Readme" file to instruct users on how to run and install LungVis. Additionally, test data are provided, so running the code shouldn't be a problem. The code itself is very succinct and follows standard code structures. The only remark I would have here is that the code could benefit from some more commenting (but that is almost always the case).

We have updated our code by adding additional comments and explanations to improve accessibility for potential users.

Reviewer #3 (Remarks to the Author):

The author has addressed the issues I raised and I recommend acceptance.

After revision, the manuscript was significantly improved. It is good to see that quantitative measurement of fluorescence stability has been added. Additional experiments were

performed to demonstrate NP-laden tissue-resident macrophages (TRMs) in tracheobronchial (TB) lymph nodes. The author has addressed the issues I raised and I recommend acceptance.

Thank you for your positive feedback and recommendation for acceptance. We are glad that the additional quantitative measurements and experiments addressing fluorescence stability and NP-laden TRMs in TB lymph nodes have strengthened the manuscript.